

**A single-point modeling approach for the intercomparison and**
**evaluation of ozone dry deposition across chemical transport models**
**(Activity 2 of AQMEII4)**
Olivia E. Clifton[1], Donna Schwede[2], Christian Hogrefe[2], Jesse O. Bash[2], Sam Bland[3], Philip Cheung[4],
Mhairi Coyle[5], Lisa Emberson[6], Johannes Flemming[7], Erick Fredj[8], Stefano Galmarini[9], Laurens
Ganzeveld[10], Orestis Gazetas[9,11], Ignacio Goded[9], Christopher D. Holmes[12], László Horváth[13], Vincent
Huijnen[14], Qian Li[15], Paul A. Makar[4], Ivan Mammarella[16], Giovanni Manca[9], J. William Munger[17], Juan L.
Pérez-Camanyo[18], Jonathan Pleim[19], Limei Ran[20], Roberto San Jose[18], Sam J. Silva[21], Ralf Staebler[22],
Shihan Sun[23], Amos P. K. Tai[23,24], Eran Tas[15], Timo Vesala[16,25], Tamás Weidinger[26], Zhiyong Wu[27],
Leiming Zhang[4]
[1]NASA Goddard Institute for Space Studies, New York, NY, 10025 USA
[2]United States Environmental Protection Agency, Office of Research and Development, Research Triangle Park, NC, 27711
USA
[3]Stockholm Environment Institute, Environment and Geography Department, University of York, York, YO10 5DD UK
[4]Air Quality Research Division, Atmospheric Science and Technology Directorate, Environment and Climate Change Canada,
Toronto, M3H 5T4, Canada
[5]United Kingdom Centre for Ecology and Hydrology, Bush Estate, Penicuik, Midlothian, EH26 0QB UK, and The James Hutton
Institute, Craigiebuckler, Aberdeen, AB15 8QH UK
[6]Environment and Geography Department, University of York, York, YO10 5DD UK
[7]European Centre for Medium-Range Weather Forecasts, Reading, RG2 9AX UK
[8]Department of Computer Science, The Jerusalem College of Technology, Jerusalem, Israel
[9] European Commission, Joint Research Centre (JRC), Ispra, Italy
[10]Wageningen University, Meteorology and Air Quality Section, Wageningen, the Netherlands
[11]Now at: Scottish Universities Environmental Research Centre (SUERC), East Kilbride G75 0QF, UK
[12]Earth, Ocean and Atmospheric Science, Florida State University, Tallahassee, FL, 32306 USA
[13]Department of Optics and Quantum Electronics, ELKH-SZTE Photoacoustic Research Group, University of Szeged, Szeged,
Hungary
[14]Royal Netherlands Meteorological Institute, De Bilt, Netherlands
[15]The Institute of Environmental Sciences, The Robert H. Smith Faculty of Agriculture, Food and Environment, The Hebrew
University of Jerusalem, Rehovot 76100, Israel
[16]Institute for Atmospheric and Earth System Research/Physics, University of Helsinki, Helsinki, Finland
[17]School of Engineering and Applied Sciences and Department of Earth and Planetary Sciences, Harvard University, Cambridge,
MA, USA
[18]Computer Science School, Technical University of Madrid (UPM), Madrid, Spain
[19]Center for Environmental Measurement & Modeling, U.S. Environmental Protection Agency, Research Triangle Park, NC,
USA
[20]Natural Resources Conservation Service, US Department of Agriculture, Greensboro, NC, USA
[21]Department of Earth Sciences, University of Southern California, Los Angeles, CA
[22]Air Quality Processes Section, Environment and Climate Change Canada, Toronto, M3H 5T4, Canada
[23]Earth and Environmental Sciences Programme, Faculty of Science, The Chinese University of Hong Kong, Hong Kong, China
[24]State Key Laboratory of Agrobiotechnology and Institute of Environment, Energy and Sustainability, The Chinese University
of Hong Kong, Hong Kong, China
[25]Institute for Atmospheric and Earth System Research/Forest Sciences, University of Helsinki, Helsinki, Finland
[26]Department of Meteorology, Institute of Geography and Earth Sciences, Eötvös Loránd University, Pázmány Péter sétány 1/A,
Budapest 1117, Hungary



[27]ORISE Fellow at Center for Environmental Measurement and Modeling, US Environmental Protection Agency, Research
Triangle Park, NC, 27711 USA
*Correspondence to*: Olivia E. Clifton (olivia.e.clifton@nasa.gov)
**Abstract.** A primary sink of air pollutants and their precursors is dry deposition. Dry deposition estimates differ across chemical
transport models yet an understanding of the model spread is incomplete. Here we introduce Activity 2 of the Air Quality Model
Evaluation International Initiative Phase 4 (AQMEII4). We examine dry deposition schemes from regional and global chemical
transport models as well as standalone models used for impacts assessments or process understanding. We configure eighteen
schemes as single-point models at eight northern hemisphere locations with observed ozone fluxes. Single-point models are
driven by a common set of site-specific meteorological and environmental conditions. Five of eight sites have at least three years
and up to twelve years of ozone fluxes. The spread across models that de-emphasizes outliers in multiyear mean ozone
deposition velocities ranges from a factor of 1.2 to 1.9 annually across sites and tends to be highest during winter compared to
summer. No model is within 50% of observed multiyear averages across all sites and seasons, but some models perform well for
some sites and seasons. For the first time, we demonstrate how contributions from depositional pathways vary across models.
Models can disagree in relative contributions from the pathways, even when they predict similar deposition velocities, or agree in
the relative contributions but predict different deposition velocities. Both stomatal and nonstomatal uptake contribute to the large
model spread across sites. Our findings are the beginning of results from AQMEII4 Activity 2, which brings scientists who
model air quality and dry deposition together with scientists who measure ozone fluxes to evaluate and improve dry deposition
schemes in chemical transport models used for research, planning, and regulatory purposes.
**Short summary.** A primary sink of air pollutants is dry deposition. Dry deposition estimates differ across models used to
simulate atmospheric chemistry on regional to global scales. Here we introduce an effort to examine dry deposition schemes
from atmospheric chemistry models. We provide our approach's rationale, document the schemes, and describe datasets used to
drive and evaluate the schemes. We also launch the analysis of results by evaluating against observations and identifying the
processes leading to model-model differences.
**1 Introduction**
Dry deposition is a sink of air pollutants and their precursors, removing compounds from the atmosphere after turbulence
transports them to the surface and the compounds stick to or react with surfaces. Dry deposition may be a key influence on air
pollution levels including high episodes (Vautard et al., 2005; Solberg et al., 2008; Emberson et al., 2013; Huang et al., 2016;
Anav et al., 2018; Baublitz et al., 2020; Clifton et al., 2020b; Lin et al., 2020; Gong et al., 2021). Dry deposition can also harm
plants when gases diffuse through stomata (Krupa, 2003; Ainsworth et al., 2012; Lombardozzi et al., 2013; Grulke and Heath,
2019; Emberson, 2020). In particular, stomatal uptake of ozone adversely impacts crop yields (Mauzerall and Wang, 2001; Tai et



al., 2014; McGrath et al., 2015; Guarin et al., 2019; Hong et al., 2020; U.S. EPA 2020a,b), carbon storage (Ren et al., 2007; Sitch
et al., 2007; Lombardozzi et al., 2015; Oliver et al., 2018), and ecosystem services (Paoletti et al., 2010; Manes et al., 2012).
Chemical transport models are key tools for research, planning, and regulatory purposes, including quantifying the influence of
meteorology and emissions on air pollution. Accurate estimates of sinks like dry deposition are needed for source attribution.
Simulated tropospheric and near surface abundances of air pollutants are highly sensitive to dry deposition (Wild, 2007; Tang et
al., 2011; Walker, 2014; Bela et al., 2015; Beddows et al., 2017; Hogrefe et al., 2018; Baublitz et al., 2020; Sharma et al., 2020;
Ryan and Wild, 2021; Liu et al., 2022). However, chemical transport models do not always reproduce observed variability in dry
deposition or in near-surface abundances of air pollutants expected to be influenced strongly by dry deposition (Hardacre et al.,
2015; Clifton et al., 2017; Kavassalis and Murphy, 2017; Silva and Heald, 2018; Travis and Jacob, 2019; Visser et al., 2021;
Wong et al., 2022; Ye et al., 2022).
Dry deposition rates differ across chemical transport models (Dentener et al., 2006; Flechard et al., 2011; Hardacre et al., 2015;
Li et al., 2016; Vivanco et al., 2018). Differences can stem from dry deposition scheme (Le Morvan-Quéméner et al., 2018; Wu
et al., 2018; Wong et al., 2019; Otu-Larbi et al., 2021; Sun et al., 2022) as well as near-surface concentrations of the air pollutant
and model-specific forcing related to meteorology and land use/land cover (LULC) (Hardacre et al., 2015; Tan et al., 2018, Zhao
et al., 2018; Huang et al., 2022). Even with the same forcing, deposition velocities, or the strength of the dry deposition
independent from near-surface concentrations, can vary by 2- to 3-fold across models (Flechard et al., 2011; Schwede et al.,
2011; Wu et al., 2018; Wong et al., 2019; Cao et al., 2022; Sun et al., 2022), highlighting roles for process representation and
parameter choice. Minimizing process, parametric, and structural uncertainties in dry deposition schemes is not only important
for chemical transport models used for forecasting and regulatory applications, but also for improved understanding of long-term
trends and variability in air pollution and impacts on humans, ecosystems, and resources, and building predictive ability using
global Earth system and chemistry-climate models (Archibald et al., 2020; Clifton et al., 2020a).
In addition to dry deposition occurring after diffusion through stomata, dry deposition occurs via nonstomatal pathways,
including soil and leaf cuticles, as well as snow and water (Wesely and Hicks, 2000; Helmig et al., 2007; Fowler et al., 2009;
Hardacre et al., 2015; Clifton et al., 2020a). A recent review estimates that nonstomatal uptake is 45% on average of ozone dry
deposition over physiologically active vegetation (Clifton et al., 2020a). For highly soluble gases, nonstomatal uptake may
dominate dry deposition (e.g., Karl et al., 2010; Nguyen et al., 2015; Clifton et al., 2022). Observations show strong unexpected
spatiotemporal variations in nonstomatal uptake (Lenschow et al., 1981; Godowitch, 1990; Fuentes et al., 1992; Rondón et al.,
1993; Coe et al., 1995; Mahrt et al., 1995; Fowler et al., 2001; Coyle et al., 2009; Helmig et al., 2009; Stella et al., 2011; Rannik
et al., 2012; Potier et al., 2015; Wolfe et al., 2015; Fumagalli et al., 2016; Clifton et al., 2017; Clifton et al., 2019; Stella et al.,
2019). A dearth of common process-oriented diagnostics has prevented a clear picture of the deposition pathways driving
differences in past model intercomparisons.



Measured turbulent fluxes are the best existing observational constraints on dry deposition but are limited in informing relative
roles of individual deposition pathways (Fares et al., 2017; Clifton et al., 2020a; He et al., 2021). While we can build mechanistic
understanding of individual processes with laboratory and field chamber measurements (Fuentes and Gillespie, 1992; Cape et al.,
2009; Fares et al., 2014; Fumagalli et al., 2016; Sun et al., 2016a,b; Potier et al., 2017; Finco et al., 2018), the models that are
used to scale processes to the ecosystem scale, often the same models used in dry deposition schemes in chemical transport
models, are highly empirical and poorly constrained. For example, a recent synthesis finds that while we have basic knowledge
of processes controlling ozone dry deposition, the relative importance of various processes remains uncertain and we lack ability
to predict spatiotemporal changes (Clifton et al., 2020a).
Launched in 2009, the Air Quality Model Evaluation International Initiative (AQMEII) has organized several activities (Rao et
al., 2011). The fourth phase of AQMEII emphasizes process-oriented investigation of deposition in a common framework
(Galmarini et al., 2021). AQMEII4 has two main activities. Activity 1 evaluates both wet and dry deposition across regional air
quality models (Galmarini et al., 2021). Here we introduce Activity 2, which examines dry deposition schemes as standalone
single-point models at eight sites with ozone flux observations. Importantly, single-point models are forced with the same, site-
specific observational datasets of meteorology and ecosystem characteristics, and thus the intercomparison and evaluation can
focus on deposition processes and parameters, as recommended by a recent review (Clifton et al., 2020a).
The four aims of Activity 2 are:
1.  To quantify the performance of a variety of dry deposition schemes under identical conditions
2.  To understand how different deposition pathways contribute to the intermodel spread
3.  To probe the sensitivity of schemes to environmental factors, and variability in the sensitivities across schemes
4.  To understand differences in dry deposition simulated in regional models in Activity 1
Our effort builds on recent work using observation-driven single-point modeling of dry deposition schemes at Borden Forest
(Wu et al., 2018), Ispra and Hyytiälä (Visser et al., 2021), and two sites in China (Cao et al., 2022), but is designed to test more
sites and schemes as well as gain better understanding of intermodel differences. For example, sites examined represent a range
of ecosystems in North America, Europe, and Israel, and single-point models are required to archive process-level diagnostics to
facilitate understanding of simulated variations. Although our fourth aim is to contextualize differences among regional air
quality models in Activity 1, we also include schemes from global chemical transport models and used always as standalone
models to allow for a more comprehensive range of intermodel variation.
Below we describe single-point models (Sect. 2), as well as the northern hemisphere locations and site-specific meteorological
and environmental datasets used to drive and evaluate the models (Sect. 3) and post-processing of observed and simulated values
(Sect. 4). Our focus on ozone reflects availability of long-term ozone flux measurements. With five datasets with more than three
years of observations, model evaluation can not only examine seasonality and diel cycles, but also interannual and day-to-day



variability (unique to this intercomparison). In the results (Sect. 5), we present how models differ in capturing observed
seasonality in ozone deposition velocities, including the contribution of different deposition pathways and how some
environmental factors drive changes. We focus on multiyear averages and thus climatological evaluation but examine some
aspects of interannual variability for sites with ozone flux records with three or more years. We then present a summary of our
findings (Sect. 6). To our knowledge, this is the first model intercomparison demonstrating how the contribution of different
pathways varies across dry deposition schemes and contributes to the model spread in ozone deposition velocities.
**2 Single-point models**
Single-point models used here are standalone dry deposition schemes driven by meteorological and environmental inputs from
observations at sites with ozone fluxes. The single-point models were extracted from regional models used in AQMEII4 Activity
1 as well as other chemical transport models, or are always configured as single-point models. Dry deposition schemes vary in
structure and level of detail in terms of the processes represented. Because there is limited documentation in the peer-reviewed
literature of dry deposition schemes (especially as the schemes are configured in chemical transport models), and complete and
consistent model descriptions aid our effort, we fully describe the participating schemes here. Due to our focus on ozone, we
limit our description to dry deposition of ozone. For brevity, we limit our description to the implementation of the schemes in the
single-point models at the eight sites examined, as opposed to how the schemes work at larger scales as embedded within the
chemical transport models (hereinafter, 'host models').

We note that surface- and soil-dependent parameter choices in the host model implementation of the schemes have likely been
optimized for generalized LULC and soil classification schemes as well as environmental conditions and meteorology generated
or used by the host model. Thus, our prescription of common site-specific drivers across the single-point models in this study
may create potential inconsistencies with performance inside host models. However, this separation and unification of drivers is
key for realistic estimates of the model spread due to parameter choice and process representation.

Table 1 gives measured and inferred variables or parameters used to force single-point models as well as other common variables
used in the models. The meaning and units of variables listed in Table 1 are consistent throughout the manuscript. If a variable is
not listed in Table 1 then that variable's meaning and units cannot be assumed to be consistent across models or the manuscript.
The first time that we mention variables included in Table 1, we refer to Table 1.

The forcing variables provide inputs to drive models with detailed dependencies on biophysics, such as coupled photosynthesis-
stomatal conductance models, as well as models that depend mainly on atmospheric conditions. Not every model uses every
forcing variable. In general, input variables used by each single-point model should reflect the operation of the dry deposition
scheme. For example, if the scheme in the host model ingests precipitation to calculate canopy wetness, rather than ingesting
canopy wetness, then the single-point model should ingest precipitation to calculate canopy wetness.




We note that dry deposition schemes in many chemical transport models use methods derived from classic schemes like Wesely
(1989). Implementations of classic schemes may deviate from original parameterization description papers in ways that can
affect simulated rates (e.g., Hardacre et al., 2015) but may not be well documented. For example, there may be changes to
LULC-specific parameters or the use of different LULC categories. In addition, implementations may tie processes to variables
like leaf area index to capture seasonal changes rather than relying on season-specific parameters. To foster understanding of
how adaptations from original schemes influence simulated dry deposition rates, we encouraged participation in Activity 2 from
models using schemes based on classic parameterizations, in addition to models with different approaches.
**Table 1: Variables related to forcing datasets for single-point models.**

| Variables in forcing data | Other common model variables |
|---|---|
| $B$ parameter related to soil moisture [unitless] | $D_{O_3}$ diffusivity of ozone in air [m$^2$ s$^{-1}$] |
| $[CO_2]$ ambient carbon dioxide mixing ratio [ppmv] | $D_w$ diffusivity in air of water vapor [m$^2$ s$^{-1}$] |
| $d$ displacement height [m] | $D_{CO_2}$ diffusivity in air of carbon dioxide [m$^2$ s$^{-1}$] |
| $f_{wet}$ fraction of the canopy that is wet [fractional] | $e_{sat}$ saturation vapor pressure [Pa] |
| $G$ incoming shortwave radiation [W m$^{-2}$] | $f_0$ reactivity factor for ozone [unitless] |
| $h$ canopy height [m] | $H$ Henry's Law constant [M atm$^{-1}$] |
| $LAI$ leaf area index [m$^2$ m$^{-2}$] | $\kappa$ thermal diffusivity of air [m$^2$ s$^{-1}$] |
| $[O_3]$ ambient ozone mixing ratio [ppbv] | $L$ Obukhov length [m] |
| $P$ precipitation rate [mm hr$^{-1}$] | $M_{air}$ molar mass of air [g mol$^{-1}$] |
| $p_a$ air pressure [Pa] | $Pr$ Prandtl number [unitless] |
| $PAR$ photosynthetically active radiation [$\mu$mol m$^{-2}$ s$^{-1}$] | $\rho$ air density [kg m$^{-3}$] |
| $RH$ relative humidity [fractional] | $Sc$ Schmidt number [unitless] |
| $r_0$ roughness length [m] | $v_d$ ozone deposition velocity [m s$^{-1}$] |
| $SD$ snow depth [cm] | $VPD$ vapor pressure deficit [kPa] |
| $SH$ sensible heat flux [W m$^{-2}$] | $\psi_{leaf}$ leaf water potential [MPa] |
| $T_a$ air temperature [ºC] | $\psi_{soil}$ soil matric potential [kPa] |
| $T_g$ ground temperature near surface [ºC] | |
| $u$ wind speed [m s$^{-1}$] | |
| $u^*$ friction velocity [m s$^{-1}$] | |
| $w_g$ volumetric soil water content near surface [m$^3$ m$^{-3}$] | |
| $w_2$ volumetric soil water content at root zone [m$^3$ m$^{-3}$] | |
| $w_{fc}$ volumetric soil water content at field capacity [m$^3$ m$^{-3}$] | |
| $w_{sat}$ volumetric soil water content at saturation [m$^3$ m$^{-3}$] | |
| $w_{wlt}$ volumetric soil water content at wilting point [m$^3$ m$^{-3}$] | |
| $z_0$ roughness length [m] | |
| $z_r$ reference height [m] | |
| $\theta$ solar zenith angle [º] | |


Like many model intercomparisons, our effort is an 'ensemble of opportunity' (e.g., Galmarini et al., 2004; Tebaldi and Knutti,
2007; Potempsky and Galmarini, 2009; Solazzo and Galmarini, 2014; Young et al., 2018) and may underestimate uncertainty
due to process, structural, and parametric differences across models. Nonetheless, the design of our effort, with emphasis on
processes, parameters, and sensitivities, is designed to explore uncertainty more systematically than past attempts.



The first set of Activity 2 simulations is driven by inputs from observations, and those simulations are examined here. Future
work will examine sensitivity tests in which dry deposition is calculated with perturbed values of input variables (e.g., air
temperature, leaf area index). We will also design tests that isolate the influence of input parameters (e.g., initial resistance to
stomatal uptake, field capacity of soil).

Diagnostic outputs required from single-point models follow requirements of Activity 1 (see Table 4 in Galmarini et al. (2021)).
Among required outputs are effective conductances (Paulot et al., 2018; Clifton et al., 2020b) for dry deposition to plant stomata,
leaf cuticles, the lower canopy, and soil. Not all single-point models simulate deposition to the lower canopy. As explained and
defined in Galmarini et al. (2021), an effective conductance [m s$^{-1}$] represents the portion of $v_d$ that occurs via a single pathway.
The sum of the effective conductances is $v_d$. Archiving effective conductances facilitates comparison of the contribution of each
pathway across dry deposition schemes with varying resistance frameworks (i.e., structures) and resistances to transport.
Previous model comparisons examine different absolute conductances, suggesting that differences in processes lead to
differences in $v_d$ (Wu et al., 2018; Huang et al., 2022); our approach with effective conductances offers an apples-to-apples
comparison across models.

The classic resistance network for ozone deposition velocity ($v_d$) [m s$^{-1}$] (Table 1) is based on three resistances, which are added
in series, following:
$$v_d = (r_a + r_b + r_c)^{-1} \quad (1)$$
The variable $r_a$ is aerodynamic resistance; $r_b$ is quasi-laminar boundary layer resistance around the bulk surface; $r_c$ is surface
resistance. All resistances (denoted by $r$) are in s m$^{-1}$ throughout the manuscript. Models examined here employ Eq. (1).
Exceptions are MLC-CHEM, which is a multilayer canopy model that simulates the ozone concentration gradient within the
canopy, and CMAQ STAGE, which uses surface-specific quasi-laminar resistances. Thus, MLC-CHEM and CMAQ STAGE
deviate from Eq. (1); we present $v_d$ equations for these models in the individual model subsections below. Otherwise, in this
section, we describe methods for $r_a$ and $r_b$ across models (Tables S1, S2, S3), and ozone-specific dry deposition parameters as
related to all three main resistances (Table S4). Equations for $r_c$ are in individual model subsections.

All models except one use $r_a$ equations based on Monin-Obukhov Similarity Theory (Table S1). However, the exact forms of the
equations vary across models. Obukhov length ($L$) [m] (Table 1) is often used in $r_a$ equations but is not observed. Most model $L$
equations are similar, apart from whether models use virtual or ambient temperature and whether they include bounds on $L$ (and
what the bounds are) (Table S2).

Models are configured to accept inputs and return predicted values at the specified ozone flux measurement height at the given
site (i.e., reference height $z_r$ [m] (Table 1)). Roughness length ($z_0$) [m] (Table 1) and displacement height ($d$) [m] (Table 1) are
also often used in $r_a$ equations yet are not observed, and are especially important in estimating fluxes at $z_r$ rather than the lowest





atmospheric level of the host model. Thus, we supply consistent estimates of these variables across the models that employ them.
Estimates follow Meyers et al. (1998):
$z_0 = h\left(0.23 - \frac{LAI^{0.25}}{10} - \frac{a-1}{10}\right)$ (2)
$d = h\left(0.05 + \frac{LAI^{0.2}}{2} + \frac{a-1}{20}\right)$ (3)
The variable $h$ [m] is canopy height (Table 1); $LAI$ [m$^2$ m$^{-2}$] is leaf area index (Table 1); $a$ [unitless] is a parameter based on
LULC (Meyers et al., 1998). Meyers et al. (1998) suggest a correction for $z_0$ if $LAI < 1$ but we do not employ this correction
given that it creates discontinuities in the time series.

For models employing quasi-laminar boundary layer resistance around the bulk surface (i.e., $r_b$ in Eq. (1)), most use $r_b$ from
Wesely and Hicks (1977) (Table S3). A key part of $r_b$ parameterizations is the ratio scaling the quasi-laminar boundary layer
resistance for heat to ozone ($R_{diff,b}$) (Table S4). Fundamentally, $R_{diff,b} = Sc/Pr$ , where $Sc$ [unitless] is the Schmidt number
(Table 1) and $Pr$ [unitless] is the Prandtl number (Table 1). All but one employ $R_{diff,b} = Sc/Pr = \kappa/D_{O_3}$ where $\kappa$ [m$^2$ s$^{-1}$] is
thermal diffusivity of air (Table 1), and $D_{O_3}$ [m$^2$ s$^{-1}$] is ozone diffusivity in air (Table 1); however, values of $\kappa$ and $D_{O_3}$ vary
across models (Table S4).

Table S4 also presents model prescriptions for the ratio that scales stomatal resistance from water vapor to ozone ($R_{diff,st}$),
reactivity factor for ozone ($f_0$) [unitless] (Table 1), and Henry's Law constant for ozone ($H$) [M atm$^{-1}$] (Table 1). Where used,
values of $f_0$ and $H$ are very similar across models. Some models employ temperature dependencies on $H$. Notably, values of
$R_{diff,st}$ vary from 1.2 to 1.7 across models. The current estimate of this ratio is 1.61 (Massman, 1998). GEM-MACH Zhang and
models based on GEOS-Chem prescribe lower $R_{diff,st}$ values.
**2.1 Documentation of single-point models**
**2.1.1 WRF-Chem Wesely**
WRF-Chem uses a scheme based on Wesely (1989). Parameters in Table S5 are site- and season-specific. WRF-Chem has two
seasons: midsummer with lush vegetation [day of year between 90 and 270] and autumn with unharvested croplands [day of year
less than 90 or greater than 270]. If we reference Table S5, then the parameter's value is in Table S5.
Surface resistance ($r_c$) follows:
$r_c = \left(\frac{1}{r_{st}+r_m} + \frac{1}{r_{cut}} + \frac{1}{r_{dc}+(r_{cl}+r_T)} + \frac{1}{r_{ac}+(r_g+r_T)}\right)^{-1}$ (4)
Stomatal resistance ($r_{st}$) follows:
$r_{st} = R_{diff,st}\frac{r_i}{f(T_a)f(G)}$ (5)
The parameter $r_i$ is initial resistance for stomatal uptake (Table S5).
Effects of air temperature ($T_a$) [ºC] (Table 1) follow:

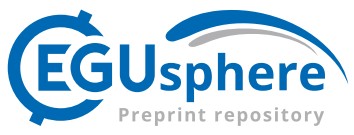

$f(T_a) = T_a \frac{(40 - T_a)}{400}$ (6)
Effects of incoming shortwave radiation ($G$) [W m$^{-2}$] (Table 1) follow:
$f(G) = \left( 1 + \left( \frac{200}{G + 0.1} \right)^2 \right)^{-1}$ (7)
Mesophyll resistance ($r_m$) follows:
$r_m = \left( \frac{H}{3000} + 100 \, f_0 \right)^{-1}$ (8)
Cuticular resistance ($r_{cut}$) follows:
$r_{cut} = \frac{r_{lu} + r_T}{\frac{H}{10^5} + f_0}$ (9)
The parameter $r_{lu}$ is initial resistance for cuticular uptake (Table S5). If relative humidity ($RH$) [fractional] (Table 1) is greater
than 0.95 or precipitation rate ($P$) [mm hr$^{-1}$] (Table 1) is greater than zero then:
$r_{cut} = \left( \frac{1}{W} + \frac{3}{r_{lu} + r_T} \right)^{-1}$ (10)
The parameter $W$ equals 3000 if $P$ equals zero whereas $W$ equals 1000 if $P$ is greater than zero.
The resistance associated with within-canopy convection ($r_{dc}$) follows:
$r_{dc} = 100 \left( 1 + \frac{1000}{G} \right)$ (11)
Resistances to the lower canopy ($r_{cl}$), in-canopy turbulence ($r_{ac}$), and soil ($r_g$) are prescribed (Table S5). To consider effects of
$T_a$, resistance $r_T$ (Walmsley and Wesely, 1996) follows:
$r_T = 1000 \, e^{-T_a - 4}$ (12)
**2.1.2 GEOS-Chem Wesely**
GEOS-Chem is based on Wesely (1989). Wang et al. (1998) describe the initial implementation. We examine the scheme from
GEOS-Chem v13.3. Parameters in Table S6 are site-specific. If there is snow, then $r_c$ is calculated with the snow parameters in
Table S6. If we reference Table S6, then the parameter's value in Table S6.
Surface resistance ($r_c$) follows:
$r_c = \left( \frac{1}{r_{st} + r_m} + \frac{1}{r_{cut}} + \frac{1}{r_{dc} + r_{cl}} + \frac{1}{r_{ac} + r_g} \right)^{-1}$ (13)
Stomatal resistance ($r_{st}$) follows:
$r_{st} = R_{diff,st} \frac{r_i}{LAI_{eff} \, f(T_a)}$ (14)
The parameter $r_i$ is initial resistance to stomatal uptake (Table S6); $LAI_{eff}$ [m$^2$ m$^{-2}$] is effective $LAI$ of actively transpiring
leaves. The variable $LAI_{eff}$ is calculated using function of $LAI$, solar zenith angle ($\theta$) [°] (Table 1), and cloud fraction. In GEOS-
Chem, if $G$ is zero then $LAI_{eff}$ equals 0.01. For the single-point model, we set $G$ to be zero when $\theta$ is greater than 95° so that
nighttime $r_{st}$ values in the single-point model more similar to GEOS-Chem. GEOS-Chem almost never has non-zero $G$ at night
but measured values are frequently small and non-zero. Here cloud fraction is assumed to be zero.



Effects of $T_a$ follows:
$f\left(T_a\right) = \begin{cases} 0.01, & T_a \leq 0 \\ T_a \frac{(40 - T_a)}{400}, & 0 < T_a < 40 \\ 0.01, & 40 \leq T_a \end{cases}$ (15)
Mesophyll resistance ($r_m$) follows:
$r_m = \left(\frac{H}{3000} + 100\, f_0\right)^{-1}$ (16)
Cuticular resistance ($r_{cut}$) follows:
$r_{cut} = \begin{cases} \frac{r_{lu} + \min\{r_T, r_{lu}\}}{LAI} \left(\frac{H}{10^5} + f_0\right)^{-1}, & \frac{r_{lu} + \min\{r_T, r_{lu}\}}{LAI} < 9999 \\ 10^{12}, & \frac{r_{lu} + \min\{r_T, r_{lu}\}}{LAI} \geq 9999 \end{cases}$ (17)
The parameter $r_{lu}$ is initial resistance for cuticular uptake (Table S6).
To consider effects of $T_a$, resistance $r_T$ follows:
$r_T = 1000\, e^{-T_a - 4}$ (18)
The resistance associated with in-canopy convection ($r_{dc}$) follows:
$r_{dc} = 100 \left(1 + \frac{1000}{G + 10}\right)$ (19)
The resistance to surfaces in the lower canopy ($r_{cl}$) follows:
$r_{cl} = \left(\frac{H}{10^5\,(r_{cl,S} + \min\{r_T, r_{cl,S}\})} + \frac{f_0}{r_{cl,O} + \min\{r_T, r_{cl,O}\}}\right)^{-1}$ (20)
Parameters $r_{cl,S}$ and $r_{cl,O}$ are initial resistances to the lower canopy (Table S6).
The resistance to turbulent transport to the soil ($r_{ac}$) is constant (Table S6). Resistance to soil ($r_g$) follows:
$r_g = \left(\frac{H}{10^5\,(r_{g,S} + \min\{r_T, r_{g,S}\})} + \frac{f_0}{r_{g,O} + \min\{r_T, r_{g,O}\}}\right)^{-1}$ (21)
Parameters $r_{g,S}$ and $r_{g,O}$ are initial resistances to uptake on soil (Table S6).
**2.1.3 IFS**
ECMWF IFS uses two schemes based on Wesely (1989): Meteo-France's SUMO (Michou et al., 2004) ("IFS SUMO Wesely")
and GEOS-Chem 12.7.2 ("IFS GEOS-Chem Wesely"). First, we describe components that are the same between schemes.
Second, we detail components specific to IFS SUMO Wesely and then to IFS GEOS-Chem Wesely. IFS SUMO Wesely
parameters in Table S7 are site- and season-specific. Seasons are defined as: 'transitional spring' [March, April, May], 'mid-
summer' [June, July, August], 'autumn' [September, October, November] and 'late autumn' [December, January, February].
Otherwise, if there is snow then the model employs the 'winter, snow' parameter values. IFS GEOS-Chem Wesely parameters
in Table S8 are site-specific. If there is snow, then the model employs the snow type. For snow type, only $r_{cl}$ is defined [1000 s
m$^{-1}$]. If we reference one of the tables, then the parameter's value is in the table.
Surface resistance ($r_c$) follows:





$\quad r_c = \left( \frac{1}{r_{st}+r_m} + \frac{1}{r_{cut}} + \frac{1}{r_{dc}+r_{cl}} + \frac{1}{r_{ac}+(r_g+r_T)} \right)^{-1}$ (22)
Mesophyll resistance ($r_m$) follows:
$\quad r_m = \left( \frac{H}{3000} + 100\, f_0 \right)^{-1}$ (23)
The resistance associated with in-canopy convection ($r_{dc}$) follows:
$\quad r_{dc} = 100 \left( 1 + \frac{1000}{G} \right)$ (24)
Resistances to surfaces in the lower canopy ($r_{cl}$), in-canopy turbulence ($r_{ac}$), and soil ($r_g$) are prescribed (Tables S7 and S8). To
consider effects of $T_a$, resistance $r_T$ follows:
$\quad r_T = 1000\, e^{-T_a - 4}$ (25)
For IFS SUMO Wesely, stomatal resistance ($r_{st}$) follows:
$\quad r_{st} = R_{diff,st} \, \frac{r_i}{LAI\, f(G)\, f(VPD)\, f(w_2)}$ (26)
The parameter $r_i$ is initial resistance to stomatal uptake (Table S7).
Effects of $G$ follow:
$\quad f(G) = \min \left\{ \frac{0.004\, G + 0.5}{0.81\, (0.004\, G + 1)}, 1 \right\}$ (27)
Effects of vapor pressure deficit ($VPD$) [kPa] (Table 1) follow:
$\quad f(VPD) = e^{0.3\, VPD}$ (28)
Equation (28) is only employed for forests, otherwise $f(VPD)$ equals 1.
Effects of root-zone soil water content ($w_2$) [m$^3$ m$^{-3}$] (Table 1) follow:
$\quad f(w_2) = \begin{cases} 0, w_2 < w_{wlt} \\ \frac{w_2 - w_{wlt}}{w_{fc} - w_{wlt}}, w_{wlt} < w_2 < w_{fc} \\ 1,\ w_2 > w_{fc} \end{cases}$ (29)
Cuticular resistance ($r_{cut}$) follows:
$\quad r_{cut} = (r_{lu} + r_T) \left( \frac{H}{10^5} + f_0 \right)^{-1}$ (30)
The parameter $r_{lu}$ is initial resistance for cuticular uptake (Table S7).
For IFS GEOS-Chem Wesely, stomatal resistance ($r_{st}$) follows Wang et al. (1998):
$\quad r_{st} = R_{diff,st} \, \frac{r_i}{LAI_{eff}\, f(T_a)}$ (40)
The parameter $r_i$ is initial resistance to stomatal uptake (Table S8); $LAI_{eff}$ [m$^2$ m$^{-2}$] is effective $LAI$ of actively transpiring
leaves. The variable $LAI_{eff}$ is calculated as a function of $LAI$, $\theta$, and cloud fraction. In GEOS-Chem, if $G$ is zero then $LAI_{eff}$ is
equal to 0.01. For the single-point model, we set $G$ to be zero when $\theta$ is greater than 95°. GEOS-Chem almost never has non-
zero $G$ at night but measured values are frequently small and non-zero. This change makes nighttime $r_{st}$ values in the single-
point model more similar GEOS-Chem. Here cloud fraction is assumed to be zero.
Effects of $T_a$ follow:





$f(T_a) = T_a \frac{40 - T_a}{400}$ (41)
Cuticular resistance ($r_{cut}$) follows:
$r_{cut} = \frac{(r_{lu} + r_T)}{LAI} \left( \frac{H}{10^5} + f_0 \right)^{-1}$ (42)
The parameter $r_{lu}$ is initial resistance to cuticular uptake (Table S8).
**2.1.4 GEM-MACH Wesely**
Operationally, GEM-MACH uses a dry deposition scheme based on Wesely (1989) (Makar et al., 2018). Parameters defined in
Table S9 are site- and sometimes season-specific. Table S10 describes how seasons are distributed as a function of month and
latitude. If we reference Table S9, then the parameter's value is in Table S9.
Surface resistance ($r_c$) follows:
$r_c = \left( \frac{1 - W_{st}}{r_{st} + r_m} + \frac{1}{r_{cut}} + \frac{1}{r_{dc} + r_{cl}} + \frac{1}{r_{ac} + r_g} \right)^{-1}$ (43)
The variable $W_{st}$ [fractional] is used to account for leaf wetness; $W_{st}$ is 0.5 if $P$ is greater than 1 mm hr[-1] or $RH$ is greater than
0.95 and zero otherwise.
Stomatal resistance ($r_{st}$) is based on Jarvis (1976), Zhang et al. (2002a, 2003) and Baldocchi et al. (1987):
$r_{st} = R_{diff,st} \frac{r_i}{LAI \max\{f(G)\, f(VPD)\, f(T_a)\, f(c_a),\ 0.0001\}}$ (44)
The parameter $r_i$ is initial resistance to stomatal uptake (Table S9).
Curve-fitting of data from Jarvis (1976) and Ellsworth and Reich (1993) was used to infer the following:
$f(G) = \max\{0.206 \ln(G) - 0.605, 0\}$ (45)
Effects of $VPD$ follow:
$f(VPD) = \max\left\{0.0, \max\left\{1.0, \left(1.0 - 0.03 \left(1 - RH\right) 10^{\frac{0.7859 + 0.03477\, T_a}{1 + 0.00412\, T_a}}\right)\right\}\right\}$ (46)
Effects of $T_a$ follow:
$f(T_a) = \left( \frac{(T_a - T_{min})(T_{max} - T_a)}{(T_{opt} - T_{min})(T_{max} - T_{opt})} \right)^{0.62}$ (47)
Parameters $T_{min}$, $T_{max}$, and $T_{opt}$ [°C] are minimum, maximum, and optimum temperature, respectively (Table S9).
Effects of ambient carbon dioxide mixing ratio ($[CO_2]$) [ppmv] (Table 1) follow:
$f(c_a) = \begin{cases} 1, & [CO_2] \leq 100 \\ 1 - \left(7.35\ x\ 10^{-4}\ \ln\bigl(\ln(G)\bigr) - 8.75\ x\ 10^{-4}\right) [CO_2], & 100 < [CO_2] < 1000 \\ 0, & [CO_2] \geq 1000 \end{cases}$ (48)
Mesophyll resistance ($r_m$) follows:
$r_m = \left( LAI \left( \frac{H}{3000} + 100\, f_0 \right) \right)^{-1}$ (49)
Cuticular resistance ($r_{cut}$) follows:
$r_{cut} = \frac{r_{lu}}{LAI} \left( \frac{H}{10^5} + f_0 \right)^{-1}$ (50)



The parameter $r_{lu}$ is initial resistance to cuticular uptake (Table S9).
The resistance associated with in-canopy convection ($r_{dc}$) follows:
$r_{dc} = 100 + \left( 1 + \frac{1000}{G + 10} \right)$ (51)
The resistance posed by uptake to the lower canopy ($r_{cl}$) follows:
$r_{cl} = \left( \frac{H}{10^5 r_{cl,S}} + \frac{f_0}{r_{cl,O}} \right)^{-1}$ (52)
Parameters $r_{cl,S}$ and $r_{cl,O}$ are initial resistances to uptake by surfaces in the lower canopy (Table S9).
The parameter $r_{ac}$ is resistance to in-canopy turbulence and $r_g$ is resistance to soil; both are prescribed (Table S9).
**2.1.5 GEM-MACH Zhang**
GEM-MACH also has an implementation of Zhang et al. (2002b). Parameters in Table S11 are site-specific. If we reference
Table S11, then the parameter's value is in Table S11.
Surface resistance ($r_c$) follows:
$r_c = \min \left\{ 10, \left( \frac{1 - W_{st}}{r_{st}} + \frac{1}{r_{cut}} + \frac{1}{r_{ac} + r_g} \right)^{-1} \right\}$ (53)
The variable $W_{st}$ [fractional] is used to account for leaf wetness; $W_{st}$ is zero unless precipitation or dew is occurring using the
below thresholds, and $G$ is greater than 200 W m$^{-2}$. If this is the case,
$W_{st} = \min \left\{ 0.5, \frac{G - 200}{800} \right\}$ (54)
Precipitation is assumed to occur if $T_a$ is greater than -1ºC and $P$ is greater than 0.20 mm hr$^{-1}$. Dew is assumed to occur if $T_a$ is
greater than -1ºC and $P$ is less than 0.20 mm hr$^{-1}$ and
$u^* < c_{dew} \frac{1.5}{\max\left\{1 \, x \, 10^{-4}, \frac{0.622 \, e_{sat} \, (1 - RH)}{p_a}\right\}}$ (55)
The variable $e_{sat}$ [Pa] is saturation vapor pressure (Table 1); $p_a$ [Pa] is air pressure (Table 1); $c_{dew}$ is the dew coefficient [0.3].
Stomatal resistance ($r_{st}$) follows:
$r_{st} = R_{diff,st} \frac{r_i(LAI, PAR)}{f(T_a) \, f(VPD) \, f(\psi_{leaf})}$ (56)
The variable $r_i(LAI, PAR)$ is initial resistance to stomatal uptake that varies with $LAI$ and $PAR$, based on Norman (1982) and
Zhang et al. (2001):
$r_i(LAI, PAR) = \left( \frac{LAI_{sun}}{r_i \left(1 + \frac{b_{rs}}{PAR_{sun}}\right)} + \frac{LAI_{shd}}{r_i \left(1 + \frac{b_{rs}}{PAR_{shd}}\right)} \right)^{-1}$ (57)
The parameter $r_i$ is initial resistance to stomatal uptake (Table S11); $b_{rs}$ [W m$^{-2}$] is empirical (Table S11); $LAI_{sun}$ and $LAI_{shd}$
[m$^2$ m$^{-2}$] are sunlit and shaded LAI:
$LAI_{sun} = \frac{1 - e^{-K_b \, LAI}}{K_b}$ (58)
$LAI_{shd} = LAI - LAI_{sun}$ (59)



The variable $K_b$ is canopy light extinction coefficient [unitless]:
$K_b = \frac{0.5}{\cos\left(\frac{\pi}{180}\theta\right)}$ (60)
Variables $PAR_{sun}$ and $PAR_{shd}$ [W m$^{-2}$] are photosynthetically active radiation reaching sunlit and shaded leaves:
$PAR_{shd} = PAR_{diff}\, e^{-0.5\, LAI^a} + 0.07\, PAR_{dir}\, (1 - 0.1\, LAI)e^{-\cos\left(\frac{\pi}{180}\theta\right)}$ (61)
$PAR_{sun} = PAR_{shd} + \frac{0.5\, PAR_{dir}^b}{\cos\left(\frac{\pi}{180}\theta\right)}$ (62)
If $LAI$ is greater than 2.5 m$^2$ m$^{-2}$ and $G$ is less than 200 W m$^{-2}$, then empirical parameters $a$ equals 0.8 and $b$ equals 0.8.
Otherwise, $a$ equals 0.07 and $b$ equals 1. Calculation of direct and diffuse components of $PAR$ ($PAR_{dir}$ and $PAR_{diff}$) has been
updated from Zhang et al. (2001) to follow Iqbal (1983):
$PAR_{dir} = G\, FRAD_V\, FD_V$ (63)
$PAR_{diff} = G\, FRAD_V\, (1 - FD_V)$ (64)
The variable $FRAD_v$ follows:
$FRAD_V = \frac{R_V}{R_V + R_N}$ (65)
Variables $R_v$ and $R_N$ follow:
$R_N = RD_M + RD_N$ (66)
$R_V = RD_U + RD_V$ (67)
The variable $RD_U$ follows:
$RD_U = 600 \cos\left(\frac{\pi}{180}\theta\right) e^{\frac{-0.185\, p_a}{p_{std}\cos\left(\frac{\pi}{180}\theta\right)}}$ (68)
The variable $p_{std}$ is standard air pressure [1.0132 x 10$^5$ Pa].
The variable $RD_V$ follows:
$RD_V = 0.42\,(600 - RD_U)\cos\left(\frac{\pi}{180}\theta\right)$ (69)
The variable $RD_M$ follows:
$RD_M = \cos\left(\frac{\pi}{180}\theta\right)\left(720\, e^{\left(-\frac{0.06\, p_a}{p_{std}\cos\left(\frac{\pi}{180}\theta\right)}\right)} - \left(1320 * 0.077\left(\frac{2\, p_a}{p_{std}\cos\left(\frac{\pi}{180}\theta\right)}\right)^{0.3}\right)\right)$ (70)
The variable $RD_N$ follows:
$RD_N = 0.65\cos\left(\frac{\pi}{180}\theta\right)\left(720 - RD_M - \left(1320 * 0.077\left(\frac{2\, p_a}{p_{std}\cos\left(\frac{\pi}{180}\theta\right)}\right)^{0.3}\right)\right)$ (71)
The variable $FD_v$ follows:



$$FD_V = \begin{cases} 0.941124\, RD_U/R_V & \frac{G}{R_V+R_N} \geq 0.89 \\ \left(1 - \left(\frac{\left(0.9 - \frac{G}{R_V+R_N}\right)}{0.7}\right)^{\frac{2}{3}}\right) RD_U/R_V & 0.21 \geq \frac{G}{R_V+R_N} < 0.89 \\ 0.00955\, RD_U/R_V & \frac{G}{R_V+R_N} < 0.21 \end{cases} \text{(72)}$$

Effects of $T_a$ follow:
$f(T_a) = \left(\frac{T_a - T_{min}}{T_{opt} - T_{min}}\right)\left(\frac{T_{max} - T_a}{T_{max} - T_{opt}}\right)^{\frac{T_{max} - T_{opt}}{T_{max} - T_{min}}}$ (73)
Parameters $T_{min}$, $T_{max}$, and $T_{opt}$ [ºC] are minimum, maximum, and optimum temperature, respectively (Table S11).
Effects of $VPD$ follow:
$f(VPD) = \min\{\max\{1 - b_{vpd}\, VPD, 0\}, 1\}$ (74)
The parameter $b_{vpd}$ [kPa$^{-1}$] is empirical (Table S11).
Effects of leaf water potential ($\psi_{leaf}$) [MPa] (Table 1) follow:
$f\left(\psi_{leaf}\right) = \min\left\{\max\left\{\frac{\psi_{leaf} - \psi_{leaf,2}}{\psi_{leaf,1} - \psi_{leaf,2}}, 0\right\}, 1\right\}$ (75)
The variable $\psi_{leaf}$ is approximated as:
$\psi_{leaf} = -0.72 - 0.0013\, G$ (76)
Parameters $\psi_{leaf,1}$ and $\psi_{leaf,1}$ [MPa] are empirical (Table S11).
If $T_a$ is greater than or equal to -1 ºC and there is neither precipitation nor dew then cuticular resistance ($r_{cut}$) follows:
$r_{cut} = \max\left\{100, \frac{c_{cut,dry}}{u^*\, LAI^{0.25}\, e^{3\,RH}}\right\}$ (77)
The variable $u^*$ [m s$^{-1}$] is friction velocity (Table 1); $c_{cut,dry}$ [unitless] is a coefficient related to dry cuticular uptake (Table S11).
If $T_a$ is less than -1ºC and there is neither precipitation nor dew then:
$r_{cut} = \max\left\{100, \frac{r_{cut,dry}}{u^*\, LAI^{0.25}\, e^{3\,RH}}\, \min\{2, e^{0.2\,(-1-T_a)}\}\right\}$ (78)
If there is precipitation or dew and $T_a$ is greater than or equal to -1ºC then:
$r_{cut} = \frac{c_{cut,wet}}{u^*\, \sqrt{LAI}}$ (79)
The parameter $c_{cut,wet}$ [unitless] is a coefficient related to dry cuticular uptake (Table S11).
If the fraction of snow coverage ($f_{snow}$) is greater than $10^{-4}$ then a correction is applied:
$r_{cut} = \left(\frac{1 - f_{snow}}{r_{cut}} + \frac{f_{snow}}{2000}\right)^{-1}$ (80)
If $LAI$ is less than 2 x 10$^{-6}$ m$^2$ m$^{-2}$ then $r_{cut}$ is very large.
The resistance to in-canopy turbulence ($r_{ac}$) follows:
$r_{ac} = r_{ac0}\, \frac{LAI^{0.25}}{(u^*)^2}$ (81)





The variable $r_{ac0}$ follows:
$r_{ac0} = r_{ac0,min} + \frac{LAI - LAI_{min}}{LAI_{max} - LAI_{min}} \left( r_{ac0,max} - r_{ac0,min} \right)$ (82)
Parameters $LAI_{min}$ and $LAI_{max}$ [m$^2$ m$^{-2}$] are minimum and maximum $LAI$ across the site's observational record; $r_{ac0,min}$ and
$r_{ac0,max}$ are initial resistances (Table S11).
Soil resistance ($r_g$) is prescribed but modified under certain conditions. If $T_s$ is less than -1ºC then:
$r_g = r_g \min\{2, e^{-0.2\,(T_s + 1)}\}$ (83)
The near-surface air temperature ($T_s$) is approximated from a linear interpolation between $T_a$ and $T_g$ to a height of 1.5 m. If $f_{snow}$
is greater than or equal to $10^{-4}$ then:
$r_g = \left( \frac{1 - \min\{1,\ 2f_{snow}\}}{r_g} + \frac{\min\{1,\ 2f_{snow}\}}{2000} \right)^{-1}$ (84)
The fraction of snow coverage ($f_{snow}$) follows:
$f_{snow} = \min\left\{1, \frac{SD}{SD_{max}}\right\}$ (85)
The variable $SD$ [cm] is snow depth (Table 1); $SD_{max}$ [cm] is maximum snow depth (Table S11).
**2.1.6 CMAQ M3Dry**
M3Dry (Pleim and Ran, 2011) is designed to couple with the Pleim-Xiu land surface model (PX LSM; Pleim and Xiu, 1995) in
the Weather Research and Forecasting (WRF) model and is used operationally in CMAQ. There is also M3Dry-psn, which
follows M3Dry but uses a coupled photosynthesis-stomatal conductance model. M3DRY-psn was developed and evaluated with
the intention to supplement PX LSM and M3Dry in CMAQ (Ran et al., 2017). To date, however, M3DRY-psn has not been
implemented in CMAQ. We first describe M3Dry, and then M3Dry-psn. Parameters in Table S12 are site-specific. If we
reference Table S12, then the parameter's value is in Table S12.
Surface resistance ($r_c$) follows:
$r_c = \left( f_{veg} \left( \frac{1}{r_{st} + r_m} + \frac{(1 - f_{wet})\,LAI}{r_{cut,dry}} + \frac{f_{wet}\,LAI}{r_{cut,wet}} + \frac{1}{r_{ac} + r_g} \right) + \frac{1 - f_{veg}}{r_g} \right)^{-1}$ (86)
The parameter $f_{veg}$ is the fraction of the site covered by the vegetation canopy (Table S12); $f_{wet}$ is the fraction of canopy that is
wet (Table 1).
Mesophyll resistance ($r_m$) follows:
$r_m = \frac{0.01}{LAI}$ (87)
Stomatal resistance ($r_{st}$) follows Xiu and Pleim (2001):
$r_{st} = R_{diff,st} \frac{r_i}{LAI\,f(PAR)\,f(w_2)\,f(RH_l)\,f(T_a)}$ (88)
The parameter $r_i$ is initial resistance to stomatal uptake (Table S12).
Effects of photosynthetically active radiation ($PAR$) [$\mu$mol m$^{-2}$ s$^{-1}$] (Table 1) follow Echer and Rosolem (2015):



$f(PAR) = (1 - a\,LAI)(1 - e^{-0.0017\,PAR})$ (89)
The parameter $a$ [unitless] is empirical (Table S12).
Effects of $w_2$ follow Xiu and Pleim (2001):
$f(w_2) = \left(1 + e^{-5\left(\frac{w_2 - w_{wlt}}{w_{fc} - w_{wlt}} - \left(\frac{w_{fc} - w_{wlt}}{3} + w_{wlt}\right)\right)}\right)^{-1}$ (90)
Effects of leaf-level $RH$ ($RH_l$) [fractional] follow:
$f(RH_l) = RH_l = \frac{q_a\,(r_a + r_{b,v})^{-1} + q_s\,r_{st,v}^{-1}}{\left(r_{st,v}^{-1} + (r_a + r_{b,v})^{-1}\right)q_s}$ (91)
The variable $q_a$ is ambient air humidity mixing ratio, $q_s$ is saturation mixing ratio at leaf temperature ($T_{leaf}$), $r_{b,v}$ is quasi-
laminar boundary layer resistance for water vapor and $r_{st,v}$ is stomatal resistance for water vapor. M3Dry assumes: when
sensible heat flux ($SH$) [W m$^{-2}$] (Table 1) is greater than 0, then $T_{leaf}$ equals $T_a - \frac{SH}{(r_a + r_{b,h})\,\rho\,c_p}$ where $r_{b,h}$ is quasi-laminar
boundary layer resistance for heat. Otherwise, $T_{leaf}$ equals $T_a$. Equation (91) is computed using an implicit quadratic solution as
described by Xiu and Pleim (2001).
Effects of $T_a$ follow:
$f(T_a) = \begin{cases} \left(1 + e^{-0.41\,(T_a - 8.9)}\right)^{-1}, & T_a \le 29 \\ \left(1 + e^{0.5\,(T_a - 40.85)}\right)^{-1}, & T_a > 29 \end{cases}$ (92)
The variable $r_{cut,wet}$ is the resistance to wet cuticles:
$r_{cut,wet} = \begin{cases} 1250, & T_g > 0 \\ 6667, & T_g < 0 \end{cases}$ (93)
The variable $T_g$ [°C] is ground temperature near surface (Table 1).
The variable $r_{cut,dry}$ is resistance to dry cuticles:
$r_{cut,dry} = r_{cut,dry,0}(1 - f(RH)) + r_{cut,wet}\,f(RH)$ (94)
The parameter $r_{cut,dry,0}$ equals 2000 s m$^{-1}$. Effects of $RH$ follow:
$f(RH) = \max\left\{100 * \frac{RH - 0.7}{0.3}, 0\right\}$ (95)
The resistance to in-canopy turbulence ($r_{ac}$) follows Erisman et al. (1994):
$r_{ac} = 14\frac{h\,LAI}{u_*}$ (96)
Soil resistance ($r_g$) follows:
$r_g = \begin{cases} \left(\frac{1 - f_{wet}}{r_{g,dry}} + \frac{f_{wet}}{r_{g,wet}}\right)^{-1}, & no\ snow \\ \left(\frac{1 - X_m}{r_{snow}} + \frac{X_m}{r_{sndiff} + r_{g,wet}}\right)^{-1}, & snow \end{cases}$ (97)



$r_{g,wet} = \begin{cases} 500, T_g > 0 \\ 6667, T_g < 0 \end{cases}$ (98)
The variable $r_{g,dry}$ follows (Massman, 2004; Mészáros et al., 2009):
$r_{g,dry} = 200 + \left(r_{g,wet} - 200\right) \frac{w_g}{w_{fc}}$ (99)
If near-surface soil water content ($w_g$) [m$^3$ m$^{-3}$] (Table 1) is greater than soil water content at field capacity ($w_{fc}$) [m$^3$ m$^{-3}$] (Table
1) then soil is wet (i.e., $r_{g,dry}$ equals $r_{g,wet}$). The parameter $r_{snow}$ is resistance to snow or ice [6667 s m$^{-1}$]; $r_{sndiff}$ is resistance to
diffusion through snowpack [10 s m$^{-1}$]. Parallel pathways to frozen snow/ice and diffusion through snowpack to liquid water
follow Bales et al. (1987). Snow liquid water mass ($X_m$) follows:
$X_m = \begin{cases} \max\{0.02(T_a + 1)^2, \ 0.5\}, \ T_a > -1 \\ 0, T_a < -1 \end{cases}$ (100)
M3Dry-psn simulates $r_{st}$ at leaf level using the Ball-Woodrow-Berry approach (Ball et al., 1987) as described by Collatz et al.
(1991, 1992) and Bonan et al. (2011):
$r_{st} = \left( g_0 + g_1 \frac{A_n}{\frac{p_{CO_2,l}}{p_a}} RH_l \right)^{-1} \frac{D_{CO_2}}{D_{O_3}} \frac{1000.0 \, \rho}{M_{air}}$ (101)
The parameter $g_0$ equals 0.01 mol CO$_2$ m$^{-2}$ s$^{-1}$ for C$_3$ plants; $g_1$ equals 9 [unitless]; $A_n$ is leaf-level net photosynthesis [mol CO$_2$
m$^{-2}$ s$^{-1}$]; $p_{CO_2,l}$ is carbon dioxide partial pressure at the leaf surface [Pa]; $RH_l$ is leaf-level $RH$ [fractional], which follows Eq. (91)
as described for M3Dry; $D_{CO_2}$ [m$^2$ s$^{-1}$] is carbon dioxide diffusivity in air (Table 1); $\rho$ [kg m$^{-3}$] is air density (Table 1); $M_{air}$ [g
mol$^{-1}$] is molar mass of air (Table 1). Leaf-level $A_n$ is estimated based on Farquhar et al. (1980) as described by Ran et al.
(2017), based on co-limitation among three potential assimilation rates, limited by Rubisco, light, and transport of photosynthetic
products. The maximum rate of carboxylation of Rubisco ($V_{cmax}$) [μmol m$^2$ s$^{-1}$] is key for $A_n$ and thus we include values at 25°C
in Table S12.
Leaf-level $A_n$ and $r_{st}$ are calculated separately for sunlit vs. shaded leaves in M3Dry-psn. Sunlit and shaded portions of *LAI*
(*LAI$_{sun}$* and *LAI$_{shd}$*, respectively) follow Campbell and Norman (1998) and Song et al. (2009). Canopy scale $r_{st}$ follows:
$r_{st} = \left( \left( \frac{LAI_{sun}}{r_{st,sun}} + \frac{LAI_{shd}}{r_{st,shd}} \right) f(w_2) \right)^{-1}$ (102)
Variables $r_{st,sun}$ and $r_{st,shd}$ are leaf-level stomatal resistances for sunlit and shaded leaves, respectively, calculated via Eq. (101).
The function $f(w_2)$ follows Eq. (90).
**2.1.7 CMAQ STAGE**
The Surface Tiled Aerosol and Gaseous Exchange (STAGE) parameterization is an option in CMAQ. Parameters in Table S13
are site-specific. If we reference Table S13, then the parameter's value is in Table S13.



$$v_d = f_{veg}\left(r_a + \cfrac{1}{r_{b,v} + \cfrac{1}{\frac{1}{r_{st}+r_m} + \frac{1}{r_{cut}}} + \cfrac{1}{r_{ac}+r_{b,g}+r_g}}\right)^{-1} + \left(1-f_{veg}\right)\left(r_a + r_{b,g} + r_g\right)^{-1} \text{(103)}$$
CMAQ STAGE considers separate quasi-laminar boundary layer resistances around vegetation vs. the ground ($r_{b,v}$ and $r_{b,g}$,
respectively) (Table S3). The parameter $f_{veg}$ is the vegetated fraction of the site; the M3Dry value is used (Table S12). Stomatal
resistance ($r_{st}$) follows Pleim and Ran (2011):
$$r_{st} = R_{diff,st} \frac{r_i}{LAI\, f(PAR)\, f(w_2)\, f(RH_l)\, f(T_a)} \text{ (104)}$$
The parameter $r_i$ is initial resistance to stomatal uptake (Table S13). The functions follow M3Dry (Eqs. (89)-(92)).
Mesophyll resistance ($r_m$) follows Wesely (1989):
$$r_m = \left(\frac{H}{3000} + 100\, f_0\right)^{-1} \text{(105)}$$
Cuticular resistance ($r_{cut}$) follows:
$$r_{cut} = \left(LAI\left(\frac{f_{wet}}{1250} + \frac{1-f_{wet}}{2000}\right)\right)^{-1} \text{(106)}$$
The resistance to in-canopy turbulence ($r_{ac}$) is similar to Shuttleworth and Wallace (1985):
$$r_{ac} = \int_0^h \frac{dz}{K_t} \text{(107)}$$
The variable $K_t$ is in-canopy eddy diffusivity [m$^2$ s$^{-1}$]. By applying the drag coeffiecient ($C_d = \frac{u_*^2}{u^2}$), assuming a uniform vertical
distribution of leaves, and using an in-canopy attenuation coefficient of momentum following Yi (2008) [$\frac{LAI}{2}$]:
$$r_{ac} = Pr\frac{u}{u_*^2}\left(e^{\frac{LAI}{2}} - 1\right) = r_a\left(e^{\frac{LAI}{2}} - 1\right)\text{(108)}$$
The variable $u$ [m s$^{-1}$] is wind speed (Table 1).
The resistance to soil ($r_g$) changes whether soil is snow covered, dry or wet (wet is $w_g$ greater than or equal to $w_{sat}$ where $w_{sat}$
[m$^3$ m$^{-3}$] is soil water content at saturation (Table 1)). For dry ground, $r_g$ follows Fares et al. (2004) and Fumagalli et al. (2016).
An asymptotic function bounds the resistance, following observations reported in Fumagalli et al. (2016):
$$r_g = \begin{cases} 250 + 2000\, \text{atan}\left(\frac{\left(\frac{w_g - w_{wlt}}{w_{fc}}\right)^B}{\pi}\right), w < w_{sat} \\ \frac{62500}{H\,R\,(T_g+273.15)}, \ w \geq w_{sat} \\ \frac{1-X_m}{r_{snow}} + \frac{X_m}{r_{sndiff} + \frac{62500}{H\,R\,(T_g+273.15)}}, \ snow \end{cases} \text{(109)}$$
The parameter $R$ [L atm K$^{-1}$ mol$^{-1}$] is the universal gas constant; $B$ [unitless] is an empirical parameter related to soil moisture
(Table 1); $r_{snow}$ is resistance to snow or ice [6667 s m$^{-1}$]; $r_{sndiff}$ is resistance to diffusion through snowpack [10 s m$^{-1}$]. The
liquid fraction of the quasi-liquid layer in snow ($X_m$) is modeled as a system dominated by van der Walls forces using the



temperature parameterization following Huthwelker et al. (2006), and assuming a maximum of 20% to match gas-liquid
partitioning findings in Conklin et al. (1993):
$$X_m = \begin{cases} \frac{0.025}{(273.15-T_g)^{1/3}}, & 0.002 < 273.15 - T_g < 10 \\ 0.2, & 273.15 - T_g < 0.002 \end{cases} \quad (110)$$

### 2.1.8 TEMIR

The Terrestrial Ecosystem Model in R (TEMIR) provides two dry deposition schemes (Sun et al., 2022): Wesely and Zhang.
Wesely in TEMIR largely follows GEOS-Chem version 12.0.0, while Zhang follows Zhang et al. (2003). In both schemes, the
default stomatal resistance is highly empirical. TEMIR can also use two photosynthesis-based stomatal conductance models: the
Farquhar-Ball-Berry model (hereinafter, BB; Farquhar et al., 1980; Ball et al., 1987) and the Medlyn et al. (2011) model
(hereinafter, Medlyn). Thus, for TEMIR Wesely and Zhang, three stomatal conductance models are used each. We first describe
Wesely, then Zhang, and then photosynthesis-based approaches (hereinafter, psn). TEMIR Zhang parameters in Table S14 and
TEMIR psn parameters in Table S15 are site-specific. If we reference one of the tables, then the parameter's value is in the table.
For Wesely, surface resistance ($r_c$) follows:
$$r_c = \left( \frac{1}{r_{st}} + \frac{1}{r_{cut}} + \frac{1}{r_{dc} + r_{cl}} + \frac{1}{r_{ac} + r_g} \right)^{-1} \quad (111)$$

Stomatal resistance ($r_{st}$) follows Wang et al. (1998):
$$r_{st} = R_{diff,st} \frac{r_i}{LAI_{eff} \, f(T_a)} \quad (112)$$

The parameter $r_i$ is initial resistance to stomatal uptake (same for GEOS-Chem Wesely; Table S6); $LAI_{eff}$ [m$^2$ m$^{-2}$] is effective
$LAI$ of actively transpiring leaves. The variable $LAI_{eff}$ is calculated using function of $LAI$, $\theta$, and cloud fraction. In GEOS-
Chem, if $G$ is zero then $LAI_{eff}$ equals 0.01. For the single-point model, we set $G$ to be zero when $\theta$ is greater than 95° so that
nighttime $r_{st}$ values in the single-point model more similar GEOS-Chem. GEOS-Chem almost never has non-zero $G$ at night but
measured values are frequently small and non-zero. Here cloud fraction is assumed to be zero.
Effects of $T_a$ follow:
$$f(T_a) = \begin{cases} 0.01, & T_a \leq 0 \\ T_a \frac{(40-T_a)}{400}, & 0 < T_a < 40 \\ 0.01, & 40 \leq T_a \end{cases} \quad (113)$$

Cuticular resistance ($r_{cut}$) follows:
$$r_{cut} = \begin{cases} r_{lu} \min\{2, e^{0.2(-1-T_a)}\} \left( \frac{H}{10^5} + f_0 \right)^{-1}, & T_a < -1 \\ \left( \frac{r_{lu}}{LAI} + 1000 \, e^{-T_a-4} \right) \left( \frac{H}{10^5} + f_0 \right)^{-1}, & T_a \geq -1 \end{cases} \quad (114)$$

The parameter $r_{lu}$ is initial resistance for cuticular uptake. Values follow GEOS-Chem Wesely (Table S6).
The resistance associated with in-canopy convection ($r_{dc}$) follows:
$$r_{dc} = 100 \left( 1 + \frac{1000}{G+10} \right) \quad (115)$$

The resistance to the lower canopy ($r_{cl}$) follows:

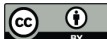


$r_{cl} = \left(\frac{H}{10^5 r_{cl,S}} + \frac{f_0}{r_{cl,O}}\right)^{-1}$ (116)
Parameters $r_{cl,S}$ and $r_{cl,O}$ are initial resistances to uptake to the lower canopy and follow GEOS-Chem Wesely (Table S6).
Resistance to soil $(r_g)$ follows:
$r_g = \left(\frac{H}{10^5 r_{g,S}} + \frac{f_0}{r_{g,O}}\right)^{-1}$ (117)
Parameters $r_{g,S}$ and $r_{g,O}$ are initial resistances to soil and follow GEOS-Chem Wesely (Table S6). The resistance to turbulent
transport to the ground $(r_{ac})$ follows GEOS-Chem Wesely (Table S6).
The changes in resistances when there is snow follow GEOS-Chem Wesely (Table S6).
For Zhang, surface resistance $(r_c)$ follows:
$r_c = \left(\frac{1-W_{st}}{r_{st}} + \frac{1}{r_{cut}} + \frac{1}{r_{ac}+r_g}\right)^{-1}$ (118)
The variable $W_{st}$ [fractional] is used to account for leaf wetness. If $P$ is greater than 0.2 mm hr$^{-1}$ then:
$W_{st} = \begin{cases} 0, & G \leq 200 \\ \frac{G-200}{800}, & 200 \leq G \leq 600 \\ 0.5, & G > 600 \end{cases}$ (119)
Stomatal resistance $(r_{st})$ follows:
$r_{st} = R_{diff,st}\frac{r_i(LAI,PAR)}{f(T_a)\,f(VPD)\,f(\psi_{leaf})}$ (120)
Dependencies on $T_a$, $VPD$, and $\psi_{leaf}$ are as described in Brook et al. (1999).
The variable $r_i(LAI,PAR)$ follows:
$r_i(LAI,PAR) = \left(\frac{LAI_{sun}}{r_i\left(1+\frac{b_{rs}}{PAR_{sun}}\right)} + \frac{LAI_{shd}}{r_i\left(1+\frac{b_{rs}}{PAR_{shd}}\right)}\right)^{-1}$ (121)
The parameter $r_i$ is initial resistance to stomatal uptake (Table S14); $b_{rs}$ [W m$^{-2}$] is empirical (Table S14); $LAI_{sun}$ and $LAI_{shd}$
[m$^2$ m$^{-2}$] are sunlit and shaded LAI:
$LAI_{sun} = \frac{1-e^{-K_b\,LAI}}{K_b}$ (122)
$LAI_{shd} = LAI - LAI_{sun}$ (123)
The variable $K_b$ is canopy light extinction coefficient [unitless]:
$K_b = \frac{0.5}{\cos\left(\frac{\pi}{180}\theta\right)}$ (124)
The variables $PAR_{sun}$ and $PAR_{shd}$ [W m$^{-2}$] are $PAR$ reaching sunlit and shaded leaves:
$PAR_{shd} = R_{diff}\,e^{-0.5\,LAI^a} + 0.07\,R_{dir}\,(1.1 - 0.1\,LAI)\,e^{-\cos\left(\frac{\pi}{180}\theta\right)}$ (125)
$PAR_{sun} = PAR_{shd} + \frac{R_{dir}^b\,\cos\left(\frac{\pi}{180}\alpha\right)}{\cos\left(\frac{\pi}{180}\theta\right)}$ (126)



The parameter $\alpha$ is the angle between the leaf and the sun [60º]; $R_{diff}$ and $R_{dir}$ are downward visible radiation fluxes from
diffuse and direct-beam radiation above the canopy. Here we use diffuse fraction from the reanalysis product Modern-Era
Retrospective analysis for Research and Applications, Version 2 (MERRA-2) (GMAO, 2015) to separate $R_{diff}$ and $R_{dir}$ from
observed $PAR$. If $LAI$ is less than 2.5 m² m⁻² or $G$ is less than 200 W m⁻² then $a$ equals 0.7 and $b$ equals 1. Otherwise, $a$ equals
0.8 and $b$ equals 0.8.
Effects of $T_a$ follow:
$f(T_a) = \left(\frac{T_a - T_{min}}{T_{opt} - T_{min}}\right) \left(\frac{T_{max} - T_a}{T_{max} - T_{opt}}\right)^{\frac{T_{max} - T_{opt}}{T_{opt} - T_{min}}}$ (127)
Parameters $T_{min}$, $T_{max}$, and $T_{opt}$ [ºC] are minimum, maximum, and optimum temperature, respectively (Table S14).
Effects of $VPD$ follow:
$f(VPD) = 1 - b_{VPD} VPD$ (128)
The parameter $b_{VPD}$ [kPa⁻¹] is empirical (Table S14).
Effects of $\psi_{leaf}$ follow:
$f(\psi_{leaf}) = \frac{\psi_{leaf} - \psi_{leaf,2}}{\psi_{leaf,1} - \psi_{leaf,2}}$ (129)
Parameters $\psi_{leaf,1}$ and $\psi_{leaf,2}$ [MPa] are empirical (Table S14); $\psi_{leaf}$ is parameterized as:
$\psi_{leaf} = -0.72 - 0.0013 G$ (130)
Cuticular resistance ($r_{cut}$) follows:
$r_{cut} = \begin{cases} \frac{c_{cut,dry}}{u^* LAI^{0.25} e^{3 RH}}, dry \\ \frac{c_{cut,wet}}{u^* LAI^{0.5}}, wet \end{cases}$ (131)
Parameters $c_{cut,dry}$ and $c_{cut,wet}$ [unitless] are empirical coefficients related to dry and wet cuticular uptake (Table S14). If $P$ is
greater than 0.2 mm hr⁻¹ then cuticles are wet; otherwise, cuticles are dry.
The variable $r_{cut}$ is adjusted for snow:
$r_{cut} = \left(\frac{1 - f_{snow}}{r_{cut}} + \frac{2 f_{snow}}{2000}\right)^{-1}$ (132)
In-canopy aerodynamic resistance ($r_{ac}$) follows:
$r_{ac} = r_{ac0} \frac{LAI^{0.25}}{(u^*)^2}$ (133)
The variable $r_{ac0}$ follows:
$r_{ac0} = r_{ac0,min} + \frac{LAI - LAI_{min}}{LAI_{max} - LAI_{min}} \left(r_{ac0,max} - r_{ac0,min}\right)$ (134)
Variables $LAI_{min}$ and $LAI_{max}$ [m² m⁻²] are minimum and maximum observed $LAI$ during a specific year; $r_{ac0,min}$ and $r_{ac0,max}$
are initial resistances (Table S14).
Resistance to soil ($r_g$) follows:
$r_g = \left(\frac{1 - min\{1,2 f_{snow}\}}{200} + \frac{min\{1,2 f_{snow}\}}{2000}\right)^{-1}$ (135)



The variable $f_{snow}$ is the fraction of the surface covered by snow [unitless]:
$f_{snow} = \min\left\{1, \frac{SD}{SD_{max}}\right\}$ (136)
The parameter $SD_{max}$ is maximum snow depth [cm] (Table S14).

We now discuss psn options for TEMIR Wesely and TEMIR Zhang. For BB (Ball et al., 1987; Farquhar et al., 1980; von
Caemmerer and Farquhar, 1981; Collatz et al., 1991, 1992),
$r_{st} = \left( \beta_t\, g_0 + g_1\, \frac{A_n\, RH}{\frac{p_{CO_2,l}}{p_a}} \right)^{-1} \frac{p_a}{R\, \theta_a}$ (137)
The parameter $g_0$ equals 0.01 mol m$^{-2}$ s$^{-1}$; $g_1$ equals 9; $A_n$ is net photosynthesis [mol m$^{-2}$ s$^{-1}$]; $\beta_t$ is a soil water stress factor
[unitless]; $p_{CO_2,l}$ is carbon dioxide partial pressure at leaf surface [Pa]; $R$ is the universal gas constant [J mol$^{-1}$ K$^{-1}$]; $\theta_a$ is
potential air temperature [K].
For Medlyn (Medlyn et al., 2011),
$r_{st} = \left( \beta_t\, g_0 + \frac{D_w}{D_{CO_2}} \left(1 + \frac{g_{1M}}{\sqrt{VPD}}\right) \frac{A_n}{\frac{p_{CO_2,l}}{p_a}} \right)^{-1} \frac{p_a}{R\, \theta_a}$ (138)
The parameter $g_{1M}$ [kPa$^{0.5}$] is empirical (Table S15); $g_0$ equals 0.0001 mol m$^{-2}$ s$^{-1}$; $D_w$ [m$^2$ s$^{-1}$] is the diffusivity of water vapor
in air (Table 1); the ratio of diffusivities is 1.6.
A single-layer bulk soil formulation considering the root zone (0-100 cm) is used to calculate $\beta_t$:
$\beta_t = \begin{cases} 1, & \psi_{soil} > \psi_{soil,fc} \\ \frac{\psi_{soil,wlt} - \psi_{soil}}{\psi_{soil,wlt} - \psi_{soil,fc}}, & \psi_{soil,wlt} \leq \psi_{soil} \leq \psi_{soil,fc} \\ 0, & \psi_{soil} < \psi_{soil,fc} \end{cases}$ (139)
The variable $\psi_{soil}$ [kPa] is soil matric potential (Table 1):
$\psi_{soil} = \psi_{soil,sat}\, w_2^{-B}$ (140)

For both Medlyn and BB, leaf-level $r_{st}$ is calculated individually for sunlit and shaded leaves, and then scaled up:
$r_{st} = R_{diff,st} \left( \frac{LAI_{sun}}{r_{b,leaf} + r_{st,sun}} + \frac{LAI_{shd}}{r_{b,leaf} + r_{st,shd}} \right)^{-1}$ (141)
Variables $r_{st,sun}$ and $r_{st,shd}$ are leaf-level stomatal resistances for sunlit and shaded leaves, respectively; $LAI_{sun}$ and $LAI_{shd}$ are
sunlit and shaded $LAI$, respectively; $r_{b,leaf}$ is leaf boundary layer resistance:
$r_{b,leaf} = \frac{1}{c_v} \sqrt{\frac{u_*}{l}}$ (142)
The parameter $c_v$ [0.01 m s$^{-0.5}$] is the turbulent transfer coefficient; $l$ [0.04 m] is the characteristic dimension of leaves.
Variables $LAI_{sun}$ and $LAI_{shd}$ follow:
$LAI_{sun} = PAI_{sun} \frac{LAI}{LAI + SAI}$ (143)



$LAI_{shd} = PAI_{shd} \frac{LAI}{LAI+SAI}$ (144)
The variable $SAI$ [m$^2$ m$^{-2}$] is stem area index; $PAI_{sun}$ and $PAI_{shd}$ [m$^2$ m$^{-2}$] are sunlit and shaded plant area index, respectively:
$PAI_{sun} = \frac{1-e^{-K_b(LAI+SAI)}}{K_b}$ (145)
$PAI_{shd} = LAI + SAI - PAI_{sun}$ (146)
The variable SAI follows Zeng et al. (2002):
$SAI_n = \max\{0.5\,SAI_{n-1} + \max\{LAI_{n-1} - LAI_n, 0\}, 1\}$ (147)
The parameter $n$ is n$^{th}$ month of the year.
Leaf-level photosynthesis of C$_3$ plants is represented by the formulation that relates to Michaelis–Menten enzyme kinetics and
photosynthetic biochemical pathways, as in Community Land Model 4.5 (CLM4.5) (Oleson et al., 2013) and following Collatz et
al. (1992):
$A_n = \min\{A_c, A_j, A_p\} - R_d$ (148)
The Rubisco-limited photosynthetic rate ($A_c$) [mol m$^{-2}$ s$^{-1}$] follows:
$A_c = V_{cmax} \frac{c_i - \Gamma_*}{c_i + K_c\left(1 + \frac{o_i}{K_o}\right)}$ (149)
The variable $c_i$ is intercellular carbon dioxide partial pressure [Pa]; $K_c$ and $K_o$ are Michaelis–Menten constants for carboxylation
and oxygenation [Pa]; $o_i$ is intercellular oxygen partial pressure [0.029 $p_a$ Pa]; $\Gamma_*$ is carbon dioxide compensation point [Pa];
$V_{cmax}$ is maximum rate of carboxylation [mol m$^{-2}$ s$^{-1}$] adjusted for leaf temperature:
$V_{cmax} = V_{cmax,25}\, f(T_l)\, f_H(T_l)\, \beta_t$ (150)
The parameter $V_{cmax,25}$ is the value of $V_{cmax}$ at 25ºC (Table S15).
The function of leaf temperature ($T_l$) [K] follows:
$f(T_l) = e^{\frac{\Delta H_a}{298.15 * 0.001R}\left(1 - \frac{298.15}{T_l}\right)}$ (151)
The parameter $R$ is the universal gas constant [J kg$^{-1}$ K$^{-1}$]. The high temperature function of $T_l$ follows:
$f_H(T_l) = \frac{1 + e^{\frac{298.15\,\Delta S - \Delta H_d}{298.15 * 0.001\,R}}}{1 + e^{\frac{\Delta S T_v - \Delta H_d}{0.001\,R\,T_l}}}$ (152)
The variables $\Delta H_a$ [J mol$^{-1}$], $\Delta S$ [J mol$^{-1}$ K$^{-1}$], and $\Delta H_d$ [J mol$^{-1}$] are temperature dependent and follow definitions in CLM4.5
(see Table S15 for the CLM4.5 PFTs for each site).
The ribulose-1,5-bisphosphate (RuBP)-limited photosynthetic rate ($A_j$) [mol m$^{-2}$ s$^{-1}$] follows:
$A_j = \frac{J}{4} \frac{c_i - \Gamma_*}{c_i + 2\Gamma_*}$ (153)
The parameter $J$ is the electron transport rate [mol m$^{-2}$ s$^{-1}$], taken as the smaller of the two roots of the equation below:
$\theta_{PSII} J^2 - (I_{PSII} + J_{max}) J + I_{PSII} J_{max} = 0$ (154)
$J_{max} = 1.97\, V_{cmax,25}\, f(T_l)\, f_H(T_l)$ (155)
$I_{PSII} = 0.5\, \Phi_{PSII}\, 4.6\, x\, 10^{-6}\, \phi$ (156)



The parameter $\theta_{PSII}$ [unitless] represents curvature; $I_{PSII}$ [mol m$^{-2}$ s$^{-1}$] is light utilization in electron transport by photosystem II;
$J_{max}$ [mol m$^{-2}$ s$^{-1}$] is potential maximum electron transport rate; $\Phi_{PSII}$ [unitless] is quantum yield of photosystem II; $\phi$ [W m$^{-2}$]
is photosynthetically active radiation absorbed by leaves, converted to photosynthetic photon flux density with 4.6 x 10$^{-6}$ mol J$^{-1}$.
The product-limited photosynthetic rate ($A_p$) [mol m$^{-2}$ s$^{-1}$] follows:
$A_p = 3\,T_p$ (157)
The parameter $T_p$ is the triose phosphate utilization rate [mol m$^{-2}$ s$^{-1}$].
$T_p = 0.167\,V_{cmax,25}\,f(T_l)\,f_H(T_l)$ (158)
Dark respiration ($R_d$) [mol m$^{-2}$ s$^{-1}$] follows:
$R_d = 0.015\,V_{cmax,25}\,f(T_l)\,f_H(T_l)\,\beta_t$ (159)
Calculation for $A_n$ and $r_{st}$ involves a coupled set of equations that are solved iteratively at each time step until $c_i$ converges (see
Sect. 8.5 of Oleson et al., 2013):
$A_n = \dfrac{p_{CO_2,a} - p_{CO_2,i}}{\left(1.4\,r_{b,leaf} + \frac{D_w}{D_{CO_2}}r_{st}\right)p_a} = \dfrac{p_{CO_2,a} - p_{CO_2,l}}{1.4\,r_{b,leaf}\,p_a} = \dfrac{p_{CO_2,l} - p_{CO_2,i}}{\frac{D_w}{D_{CO_2}}r_{st}\,p_a}$ (160)
Variables $p_{CO_2,a}$ and $p_{CO_2,i}$ are carbon dioxide partial pressure [Pa] in air and intercellular space, respectively.
**2.1.9 DO₃SE**
DO₃SE as described below is consistent with the parameterization in the EMEP model (Simpson et al., 2012). DO₃SE uses two
methods to estimate $r_{st}$: the multiplicative method based on Jarvis (1976) ("DO₃SE multi") and the coupled photosynthesis-
stomatal conductance method based on Leuning (1995) ("DO₃SE psn"). First, we describe components that are the same between
DO₃SE multi and DO₃SE psn. Second, we describe the components unique to DO₃SE multi and then to DO₃SE psn. Parameters
in Table S16 are site-specific. If we reference Table S16, then the parameter's value is in the table.
Surface resistance ($r_c$) follows:
$r_c = \left(\dfrac{LAI}{r_{st}} + \dfrac{StAI}{r_{cut}} + \dfrac{1}{r_{ac}+r_g}\right)^{-1}$ (161)
The parameter $r_{cut}$ is resistance to cuticular uptake [2500 s m$^{-1}$]; $StAI$ is the stand area index [m$^2$ m$^{-2}$].
For forests,
$StAI = LAI + 1$ (162)
For the other LULC types examined here,
$StAI = LAI$ (163)
The resistance to in-canopy turbulence ($r_{ac}$) follows Erisman et al. (1994):
$r_{ac} = 14\dfrac{h\,StAI}{u_*}$ (164)
Resistance to soil ($r_g$) follows:
$r_g = 200 + 1000\,e^{-T_a - 4} + 2000\,\delta_{snow}$ (165)
The parameter $\delta_{snow}$ equals 1 when snow is present and 0 when snow is absent.




For DO₃SE multi, according to Simpson et al. (2012), stomatal resistance ($r_{st}$) follows:
$r_{st} = \left(g_{max} \max\{f_{min}, f(T_a)\, f(VPD)\, f(w_2)\}\, a_{phen}\, a_{light}\right)^{-1}$ (166)
The parameter $g_{max}$ is maximum stomatal conductance [m s⁻¹] (Table S16); $f_{min}$ is the minimum factor [unitless] (Table S16).
Effects of $T_a$ follow:
$f(T_a) = \frac{T_a - T_{min}}{T_{opt} - T_{min}} \left(\frac{T_{max} - T_a}{T_{max} - T_{opt}}\right)^{\frac{T_{max} - T_{opt}}{T_{opt} - T_{min}}}$ (167)
The function $f(T_a)$ equals 0.01 when $T_a$ is outside $T_{min}$ to $T_{max}$; $T_{min}$, $T_{max}$, and $T_{opt}$ [°C] are minimum, maximum, and
optimum temperature, respectively (Table S16).
Effects of $VPD$ follow:
$f(VPD) = \min\{1, \max\{f_{min}, f_{min} + (1 - f_{min})\frac{VPD_{min} - VPD}{VPD_{min} - VPD_{max}}\}$ (168)
Parameters $VPD_{min}$ and $VPD_{max}$ [kPa] are minimum and maximum $VPD$, respectively (Table S16).
Effects of $w_2$ follow:
$f(w_2) = \min\{1, \max\{f_{min}, f_{min} + (1 - f_{min})\frac{w_{wlt} - w_2}{w_{max} - 0.5\,(w_{fc} - w_{wlt})}\}$ (169)
The variable $a_{phen}$ follows:
$a_{phen} = \begin{cases} 0, d_y \leq d_{SGS}\ or\ d_y > d_{EGS} \\ \emptyset_a + \left(\frac{d_y - d_{SGS}}{(d_{SGS} + \emptyset_d) - d_{SGS}}\right)(\emptyset_b - \emptyset_a), d_{SGS} \leq d_y < d_{SGS} + \emptyset_d \\ \emptyset_b, d_{SGS} + \emptyset_d < d_y \leq d_{EGS} - \emptyset_e \\ \emptyset_b - \left(\frac{d_y - (d_{EGS} - \emptyset_e)}{d_{EGS} - \emptyset_e}\right)(\emptyset_b - \emptyset_c), d_{EGS} - \emptyset_e < d_y \leq d_{EGS} \end{cases}$ (170)
The variable $d_y$ is the day of the year; $d_{SGS}$ is day of the year that corresponds to the start of the growing season; $d_{EGS}$ is the day
of the year that corresponds to the end of the growing season. For forests, $d_{SGS}$ and $d_{EGS}$ are estimated whereby $d_{SGS}$ equals 105
at 50°N and alters by 1.5 day per degree latitude earlier on moving south and later on moving north, and $d_{EGS}$ equals 297 at 50°N
and alters by 2 days per degree latitude earlier on moving north and later on moving south. The values of $\emptyset_a$, $\emptyset_b$, $\emptyset_c$, $\emptyset_d$, and $\emptyset_e$
are given in Table S16. For other LULC, we assume a year-long growing season.
The variable $a_{light}$ follows:
$a_{light} = \frac{LAI_{sun}}{LAI}\left(1 - e^{-\alpha\, I_{PAR}^{sun}}\right) + \frac{LAI_{shd}}{LAI}\left(1 - e^{-\alpha\, I_{PAR}^{shd}}\right)$ (171)
The parameter $\alpha$ is empirical (Table S16); sunlit and shaded portions of $LAI$ ($LAI_{sun}$ and $LAI_{shd}$, respectively) follow Norman

746 (1979, 1982):

$LAI_{sun} = \left(1 - e^{-0.5\frac{LAI}{\cos\theta}}\right) 2\cos\theta$ (172)
$LAI_{shd} = LAI - LAI_{sun}$ (173)
The variables $I_{PAR}^{sun}$ and $I_{PAR}^{shade}$ [W m⁻²] follow:
$I_{PAR}^{shd} = I_{diff}e^{-0.5\,LAI^{0.7}} + 0.07\, I_{dir}(1.1 - 0.1\,LAI)\, e^{-\cos\theta}$ (174)



$I_{PAR}^{sun} = \frac{I_{dir}\cos\alpha_1}{\cos\theta} + I_{PAR}^{shd}$ (175)
The parameter $\alpha_1$ is the average inclination of leaves [°60]; $I_{diff}$ and $I_{dir}$ are diffuse and direct radiation [W m$^{-2}$] estimated as a
function of the potential to actual PAR. Potential PAR is estimated using standard solar geometry methods assuming no cloud
cover and a sky transmissivity of 0.9.

For DO3SE psn (Leuning, 1990; 1995), which requires an estimate of net photosynthesis ($A_n$) [mol CO$_2$ m$^{-2}$ s$^{-1}$] (Farquhar et al.,
1980), stomatal resistance ($r_{st}$) follows:
$r_{st} = \left( g_0 + g_1 \frac{A_n}{([CO_2]_l - \Gamma_*)\left(1+\left(\frac{VPD}{D_0}\right)^8\right)} \right)^{-1} \frac{D_{CO_2}}{D_{O_3}} \frac{1000.0\,\rho}{M_{air}}$ (176)
The parameter $g_0$ is minimum conductance [mol air m$^{-2}$ s$^{-1}$] (Leuning, 1990); $g_1$ is empirical [unitless]; $D_0$ is a parameter related
to $VPD$ [kPa] (Leuning et al., 1998) (Table S16); $[CO_2]_l$ is the leaf surface carbon dioxide mixing ratio [mol CO$_2$ mol air$^{-1}$]; $\Gamma_*$ is
carbon dioxide compensation point [mol CO$_2$ mol air$^{-1}$]. We assume the diffusivity ratio is 0.96. The variable $[CO_2]_l$ is
calculated from $[CO_2]$ and leaf boundary layer resistance ($r_{b,leaf}$):
$r_{b,leaf} = 186\sqrt{\frac{u}{l}}$ (177)
The parameter $l$ is the characteristic dimension of leaves [m].
The variable $A_n$ follows Sharkey et al. (2007):
$A_n = \min\{A_c, A_j, A_p\} - R_d$ (178)
The parameter $R_d$ is dark respiration [0.015 x 10$^{-6}$ mol m$^{-2}$ s$^{-1}$].
The Rubisco-limited rate ($A_c$) [mol m$^{-2}$ s$^{-1}$] follows:
$A_c = a_{phen}\, f(w_2)\, V_{cmax,25} \frac{[CO_2]_i - \Gamma_*}{[CO_2]_i + K_c\left(1+\frac{o_i}{K_o}\right)}$ (179)
The variable $c_i$ is intercellular carbon dioxide partial pressure [Pa]; $K_c$ and $K_o$ are Michaelis–Menten constants for carboxylation
and oxygenation [Pa]; $o_i$ is intercellular oxygen partial pressure [Pa]; $\Gamma_*$ is CO$_2$ compensation point [Pa]; $V_{cmax,25}$ is maximum
rate of carboxylation at 25°C [mol m$^{-2}$ s$^{-1}$] (Table S16); $a_{phen}$ follows Eq. (170); $f(w_2)$ follows Eq. (169).
The ribulose-1,5-bisphosphate (RuBP)-limited rate ($A_j$) [mol m$^{-2}$ s$^{-1}$] follows:
$A_j = J\frac{c_i - \Gamma_*}{a\,c_i + b\,\Gamma_*}$ (180)
The variable $J$ is electron transport rate [mol m$^{-2}$ s$^{-1}$]; $a$ and $b$ denote electron requirements for formation of NADPH and ATP,
respectively. We use $a$ equals 4 and $b$ equals 8 (Sharkey et al., 2007).
The product-limited photosynthetic rate ($A_p$) [mol m$^{-2}$ s$^{-1}$] follows:
$A_p = 0.5\, V_{cmax,25}$ (181)



**2.1.10 MLC-CHEM**
The Multi-layer Canopy and Chemistry Exchange Model (MLC-CHEM) has been applied to evaluate the role of in-canopy
interactions on atmosphere-biosphere exchanges and atmospheric composition at field sites (e.g., Visser et al., 2021) and the
global scale (e.g., Ganzeveld et al., 2010). MLC-CHEM requires a minimum $h$ of 0.5 m so has not been configured for all sites.
The canopy environment is represented by an understory and crown layer. However, radiation dependent processes such as
biogenic emissions, photolysis, and stomatal conductance are estimated at four canopy layers to consider observed large gradients in in-
canopy radiation as a function of the vertical distribution of biomass. For the single-point model, ~75% and ~25% of the total $LAI$ is
present in the crown layer and understory, respectively. These canopy structure settings are used to calculate in-canopy profiles of direct
and diffusive radiation as well as the fraction of sunlit leaves from the surface incoming solar radiation (Norman, 1979). Simulated
radiation-dependent processes for the four layers are then scaled-up to two layers for in-canopy and canopy-top fluxes and
concentrations using the vertical $LAI$ distribution.
MLC-CHEM diagnoses canopy-scale $v_d$ from simulated canopy-top ozone fluxes divided by $[O_3]$, which is ambient ozone
mixing ratio at $z_r$ [ppbv] (Table 1). Turbulent exchanges of ozone between the crown layer and understory and between the
surface and crown layer are calculated from assumed linear $[O_3]$ gradients between heights, and eddy diffusivities. The eddy
diffusivity ($K_{sl \to cl}$) [m² s⁻¹] follows (Ganzeveld and Lelieveld, 1995):
$$K_{sl \to cl} = \frac{(z_{sl} - z_{cl})}{r_a} \quad (182)$$
The eddy diffusivity between the crown layer and understory ($K_{cl \to us}$) [m² s⁻¹] follows:
$$K_{cl \to us} = K_{sl \to cl} \frac{u_{cl \to us}}{u} \quad (183)$$
The variable $u_{cl \to us}$ is wind speed at the crown layer-understory interface [m s⁻¹] calculated as a function of $u$ and canopy
structure (Cionco, 1978).
Resistance to leaf-level uptake per layer ($r_{l,layer}$) follows:
$$r_{l,layer} = \frac{r_{b,leaf} + \left(\frac{1}{r_{st}} + \frac{1}{r_{cut}}\right)^{-1}}{\max\{LAI_{layer}, 10^{-5}\}} \quad (184)$$
Leaf-level stomatal resistance ($r_{st}$) is calculated using a photosynthesis-stomatal conductance model (Ronda et al., 2001):
$$r_{st} = f(w_2) R_{diff,st} \left( \frac{D_w}{D_{CO_2}} \left( g_0 + g_1 \frac{A_n}{([CO_2] - \Gamma_*)\left(1 + 8.09\frac{VPD}{D_0}\right)} \frac{M_{air}}{1000\,\rho} \right) \right)^{-1} \quad (185)$$
The ratio of diffusivities of water vapor to carbon dioxide is 1.6; $g_0$ is set to 0.025 x 10⁻³ m s⁻¹ (Leuning, 1990); $g_1$ is set to 9.09;
$A_n$ is net photosynthesis [$\mu$mol CO₂ m⁻² s⁻¹], calculated as a function of $G$, leaf temperature, $[CO_2]$, and soil moisture (Ronda et
al., 2001); $\Gamma_*$ is CO₂ compensation point [45 ppmv]; $D_0$ [kPa] is $VPD$ at which stomata close (this term is calculated each
timestep from vegetation-specific constants; Ronda et al., 2001). The soil moisture effect follows:
$$f(w_2) = 2\max\left\{\min\left\{10^{-3}, \frac{w_s - w_{wlt}}{0.75 w_{fc} - w_{wlt}}\right\}, 1\right\} - \left(\max\left\{\min\left\{10^{-3}, \frac{w_s - w_{wlt}}{0.75\, w_{fc} - w_{wlt}}\right\}, 1\right\}\right)^2 \quad (186)$$
Leaf-level cuticular resistance ($r_{cut}$) follows (Wesely, 1989; Ganzeveld and Lelieveld, 1995; Ganzeveld et al., 1998):



$r_{cut} = \left( \frac{1 - f_{wet}}{5 \, x \, 10^5} + \frac{f_{wet}}{1000} \right)^{-1}$ (187)
In-canopy aerodynamic resistance ($r_{ac}$) considers turbulent transport through the understory to the ground:
$r_{ac} = 14 \, \frac{0.25 \, h \, LAI}{u^*}$ (188)
To estimate dry deposition to the ground, $r_{ac}$ is added in series with $r_g$, resistance to soil [400 s m$^{-1}$] (Wesely, 1989; Ganzeveld and
Lelieveld, 1995; Ganzeveld et al., 1998). If there is snow, then $r_g$ is 2000 s m$^{-1}$. Resistances are combined with the lower most
understory leaf resistance ($r_{l,layer,1}$) to create a lower most understory canopy resistance ($r_{c,layer,1}$):
$r_{c,layer,1} = \left( \frac{1}{r_{l,layer,1}} + \frac{1}{r_{ac} + r_g} \right)^{-1}$ (189)
In contrast to big-leaf schemes, effective conductances for MLC-CHEM do not add up exactly to $v_d$ because there is an in-
canopy [$O_3$] gradient due to sources and sinks and transport.

### 3 Measurements for driving and evaluating single-point models

### 3.1 Turbulent fluxes of ozone

Our best observational constraints on dry deposition are vertical turbulent fluxes, but fluxes integrate the influence of many
processes and are not necessarily only reflective of dry deposition. For example, ambient chemical loss of ozone can influence
ozone fluxes when the chemistry occurs on the timescale of turbulence. Relevant reactions for ozone fluxes are ozone reacting
with highly reactive biogenic volatile organic compounds (BVOCs) or nitrogen oxide (NO). When there are no other sources and
sinks aside from dry deposition below the measurement height, dividing the observed turbulent flux by ambient concentration at
the same height can give a measure of efficiency of dry deposition ('the deposition velocity'). While fluxes provide key
constraints on the amount of gas removed by the surface, deposition velocities aid in building predictive ability given that they
indicate how the strength of the removal changes with meteorology and environmental conditions. Turbulent fluxes are mostly
measured at individual sites, representing the 'ecosystem' scale where the measurement footprint typically extends from the
order of 100 m to 1 km. Turbulent fluxes can also be measured from airplanes (e.g., Lenschow et al., 1981; Godowitch, 1990;
Mahrt et al., 1995; Wolfe et al., 2015). Turbulent flux observations typically record changes on hourly or half hourly timescales,
which is important because there is strong sub-daily variability in dry deposition.

Here we leverage existing long-term and short-term ozone flux datasets over a variety of LULC types to develop current
understanding of model performance and the spread across current dry deposition parameterizations. Strong observed interannual
variability in ozone deposition velocities (Rannik et al., 2012; Clifton et al., 2017; Gerosa et al., 2022), as well as development of
dry deposition schemes based on short-term data (e.g., days to months), motivates our multiyear evaluation approach. Although
our evaluation effort would ideally include fluxes of many reactive gases (as well as aerosols), there are not long-term flux
measurements of most compounds for which the fluxes primarily represent dry deposition. Generally, flux observations of dry
depositing air pollutants and their precursors are oftentimes few and far between and/or challenging to access (Guenther et al.,



2011; Fares et al., 2017; Clifton et al., 2020a; Farmer et al., 2021; He et al., 2021). A key reason is that obtaining high-frequency
concentration measurements of some compounds can be challenging. Ozone fluxes are the most measured fluxes of any dry
depositing reactive gas, and they can be measured over seasonal to multiyear timescales. While the model evaluation component
of Activity 2 is only for ozone, the model comparison can be performed for other gases.

Ozone fluxes are measured either via eddy covariance or the gradient method. Eddy covariance is the most fundamental and
direct method for measuring turbulent exchange (e.g., Hicks et al., 1989; Dabberdt et al., 1993). Eddy covariance fluxes require
concentration analyzers with high measurement frequency to capture the transport of material via turbulent eddies. While fast
analyzers are available for ozone, they are resource intensive to operate. Gradient techniques are more practical because slow
analyzers can be used. However, gradient techniques assume transport only occurs down the local mean concentration gradient
while in reality organized turbulent motions can transport material up-gradient (e.g., Raupach, 1979; Gao et al., 1989; Collineau
and Brunet, 1993; Thomas and Foken, 2007; Steiner et al., 2011; Patton and Finnigan, 2013). We use some gradient ozone flux
datasets, but caution that they may be particularly uncertain, especially for tall vegetation.
**3.2 Site-specific datasets**
We simulate ozone deposition velocities by driving single-point models with site-level meteorological and environmental
variables measured or inferred from measurements at eight sites with ozone flux measurements. Table 2 summarizes site
locations, LULC types, vegetation composition, and soil types. The set of sites represents a variety of LULC types and climates.
The sites include deciduous, evergreen, and mixed forests, shrubs, grasses, and a peat bog. Climate types include Mediterranean,
temperate, and boreal, as well as maritime and continental. Dry deposition parameterizations strongly rely on the concept that
key processes and parameters are specific to LULC type. While we examine several LULC types here, we note that our
measurement testbed is likely insufficient to generalize the results of our study to specific LULC types, and thus we focus our
discussion on individual sites.

Table S17 summarizes details about ozone flux measurements, time periods examined, and post-processing of data. Five of eight
sites selected have at least three and up to twelve years of ozone flux data. The rest have fewer than three years of ozone flux
data (Auchencorth Moss, Bugacpuszta, Ramat Hanadiv) but were included to diversify climate and LULC types examined. The
eddy covariance technique is used for Auchencorth Moss, Bugacpuszta, Harvard Forest, Hyytiälä, Ispra, and Ramat Hanadiv.
The gradient technique is used for Borden Forest and Easter Bush.

The gradient technique used at Borden Forest is described in Wu et al. (2015, 2016) and was developed for Harvard Forest by
comparing gradient and eddy covariance fluxes. Wu et al. (2015) shows that the gradient technique used at Borden Forest
strongly overestimates ozone deposition velocities at night and during winter at Harvard Forest, as compared to eddy covariance.
Wu et al. (2015) also show that parameter choice can strongly influence deposition velocities inferred from the gradient



technique. Thus, seasonal and diel cycle amplitudes as well as the magnitude of observed ozone deposition velocities at Borden
Forest are uncertain.
**Table 2: Summary of ozone flux tower sites.**

| Site | Location | Land use/land cover Type | More complete description of vegetation | Soil properties |
|---|---|---|---|---|
| **Auchencorth Moss, Scotland** | 55.79ºN, 3.24ºW | Peat bog | Covered with heather, moss, and grass; vegetation primarily *Calluna vulgaris*, *Juncus effusus*, grassy hummocks, and hollows; drained and cut over 100 years ago but rewetted over many decades (Leith et al., 2014); low intensity grazing by sheep | 85% Histosols |
| **Borden Forest, Canada** | 44.32ºN, 79.93ºW | Temperate mixed forest | Boreal-temperate transition forest with mostly *Acer rubrum L.* but also *Pinus strobes L.*, *Populus grandidentata Michx.*, *Fraxinus americana L.,* and *Fagus grandifolia*; regrowing on farmland abandoned about a century ago (Froelich et al., 2015; Wu et al., 2016) | Tioga sand/sandy loam |
| **Bugacpuszta, Hungary** | 46.69ºN, 19.60ºE | Grass | Semi-natural and semi-arid; primarily *Festuca pseudovina*, *Carex stenophylla*, and *Cynodon dactylon* (Koncz et al., 2014); grazing during most of the year (Machon et al., 2015) | Chernozem with 79% sand and 13% clay in upper soil layer (10 cm) (Horváth et al., 2018) |
| **Easter Bush, Scotland** | 55.87ºN, 03.03ºW | Grass | On the boundary between two fields that have been managed for silage harvest and intensive grazing by sheep and cattle (Coyle, 2006); greater than 90% *Lolium perenne* (Coyle, 2006; Jones et al., 2017) | Imperfectly drained Macmerry with Rowanhill soil association (Eutric Cambisol) and with 20-26% clay (Jones et al., 2017) |
| **Ispra, Italy** | 45.81°N, 8.63°E | Deciduous broadleaf forest | Grassland and meadowland prior to 1960s but has since regrown undisturbed; mainly *Quercus robur, Robinia pseudoacacia, Alnus glutinosa*, and *Pinus rigida* (Ferréa et al., 2012; Putaud et al., 2014); *Q. robur* (~80%) dominates except to the southeast of the flux tower | Mostly umbrisols with sandy-loam or loamy-sand texture for top 50 cm below which soil is mainly sandy (Ferréa et al., 2012) |





| | | | where *A. glutinosa* dominates due to a higher water table | |
|---|---|---|---|---|
| **Harvard Forest, USA** | 42.54ºN, 72.17ºW | Temperate mixed forest | Regrowing on farmland abandoned over 100 years ago; dominated by *Quercus rubra* and *Acer rubrum*, with scattered individual and patches of *Tsuga canadensis*, *Pinus resinosa*, and *Pinus strobus* particularly to the northwest of the tower where *T. canadensis* are most common (Munger and Wofsy, 2021) | Canton fine sandy loam, Scituate fine sandy loam, and hardwood peat swamp (Savage and Davidson, 2001) |
| **Hyytiälä, Finland** | 61.85°N, 24.29°E | Evergreen needleleaf forest | Boreal forest; predominately *Pinus sylvestris*; shrubs underneath the canopy are *Vaccinium vitis-idaea* and *Vaccinium myrtillus*, and dense moss covers forest floor (Launiainen et al., 2013); *P. sylvestris* stand established in 1962 and thinned by 25% between January and March 2002 (Vesala et al., 2005) | Haplic podzol formed on glacial kill with 5-cm average organic layer thickness (Kolari et al., 2006) |
| **Ramat Hanadiv, Israel** | 32.55°N, 34.93°E | Shrub | Near eastern Mediterranean coast; mostly *Quercus calliprinos* and *Pistacia lentiscus*, but also include *Phillyrea latifolia*, *Cupressus*, *Sarcopoterium spinosum*, *Rhamnus lycioides*, and *Calicotome villosa*; west of the measurement tower are scattered *Pinus halepensis* (~5%) (Li et al., 2018) | Xerochrept (Li et al., 2018) and clay to silty clay (Kaplan, 1989) |


For this effort, we selected sites without known influences of highly reactive BVOCs on ozone fluxes. However, there may be
unknown influences, especially at coniferous or mixed forests (Kurpius and Goldstein, 2003; Goldstein et al., 2004; Clifton et al.,
2019; Vermeuel et al., 2021), and generally the magnitude of the contribution and how it changes with time are uncertain (Wolfe
et al., 2011; Vermeuel et al., 2022). Most sites are expected to have very low NO. There may be some influences of NO on ozone
fluxes at Ramat Hanadiv (Li et al., 2018) and Ispra, but the magnitude and timing of the contribution is uncertain. Constraining
contributions of highly reactive BVOCs and NO to ozone fluxes is beyond the scope of our work here.

Removal of observed hourly or half-hourly ozone deposition velocity outliers for all sites leverages a univariate adjusted boxplot
approach following Hubert and Vandervieren (2008), which explicitly accounts for skewness in distributions and identifies the



most extreme ozone deposition velocities at each site. Non-Gaussian univariate distributions, or skewness, are present to some
degree in each observational dataset used here. This method designates the most extreme 0.7% of a normal unimodal distribution
as outliers, but the exact percentage depends on the degree of skewness. For datasets used here, which can be highly skewed, we
filter 1–6% of ozone deposition velocities across sites. Table S17 describes any antecedent post-processing of ozone deposition
velocities performed for this effort.

Many dry deposition schemes include adjustments for snow. Table S18 identifies sites with snow depth ($SD$) measurements.
Unless the single-point model directly takes $SD$ input to infer fractional snow coverage of the surface, we define the presence of
snow as $SD$ greater than 1 cm. Models assume no snow if $SD$ less than or equal to 1 cm or missing.

Canopy wetness is an input to several single-point models. Others do not ingest canopy wetness explicitly as an input variable,
but rather indicate canopy wetness using a precipitation and/or dew indicator. For the latter type, the fraction of canopy wetness
($f_{wet}$) from datasets is not used, and models' indicators are used. Table S18 details canopy wetness measurements at each site.
For sites where $f_{wet}$ data are not available, $f_{wet}$ values are approximated using an approach used in CMAQ (Table S18).

Soil moisture and soil properties and hydraulic variables are important for stomatal conductance as well as soil deposition
processes (Fares et al., 2014; Fumagalli et al., 2016; Stella et al., 2011, 2019). Site-specific details of variables used for near-
surface and root-zone volumetric soil water content are described in Table S19. A set of soil hydraulic properties (Table S20) are
estimated for each site from soil texture and used across models employing these parameters.

Overall, the core description for each site includes key information needed to drive the single-point models: LULC type,
vegetation composition, soil type, and measurement height for ozone fluxes (Tables 2 and S17). We also describe inputs for
snow, canopy wetness, $h$, and $LAI$ (Table S18). Outside of the core description, other meteorological variables are measured with
standard techniques, which are not discussed here. When an input variable is inferred, we detail assumptions involved in the
inference because variability in inferred input variables may not be accurately represented and this may need to be accounted for
in comparing simulated vs. observed ozone deposition velocities (Tables S17 and S19).

We note that in addition to data screening conducted by data providers, driving datasets were visually inspected and clearly
erroneous values were set to missing (e.g., in one case $T_a$ less than -50°C). Driving datasets are not gap-filled (unless explicitly
stated otherwise) so simulated ozone deposition velocities have gaps whenever one or more of a model's input variables is missing.
Single-point models require different sets of input variables. Thus, output from different models may have different data gaps at
a given site. Additionally, because data capture for observed deposition velocities is based on availability of ozone flux
measurements, and data gaps in input variables may be different from data gaps in the ozone flux measurements, simulated
deposition velocities can have different data gaps from observed deposition velocities. We address data coverage discrepancies



across models and observed deposition velocities in two ways. First, we identify time-averaged observed and simulated
deposition velocities with suboptimal coverage in our results (e.g., see Figure 1). Second, we account for diel imbalances in our
analysis. Both approaches are described more fully in Section 4.
**4 Creation of monthly and seasonal average observed and simulated quantities**
We examine averages across 24 hours, except for Ramat Hanadiv. For Ramat Hanadiv, many months have missing values during
night and morning and thus we limit our analysis to 11am–5pm. Across sites and analyses, we use a weighted averaging
approach for daily averages that considers the number of observations for a given hour to avoid over-representation of any given
hour due to sampling imbalances across the diel cycle (e.g., more valid observations during daylit hours).

There are sometimes periods of missing ozone fluxes in the datasets. We indicate year-specific monthly averages with low data
capture for observed ozone deposition velocities ($v_d$) on Figure 1. Low data capture is defined as less than or equal to 25% data
capture averaged across 24 hours (or 11am–5pm for Ramat Hanadiv). In other words, we first compute data capture for each
hour of a given month (or season), and then average across hour-specific data capture rates to compare against the 25%
threshold. We indicate multiyear monthly averages with low data capture for observations and models on Figures 2 and 3. Note
that the number of data points used in constructing monthly averages differs between models and observations, and across
models. Data capture for each model depends on availability of the specific measured input data required for driving that model.
Data capture for observed $v_d$ is based on availability of ozone flux measurements

When we examine multiyear averages, we do not consider sampling biases across years (e.g., more valid observations in one
year over the other). Thus, more data in one year may skew multiyear averages towards values for that year (Fig. 1). However,
results are generally similar if we include weighting by years, except when there are only a few years contributing to multiyear
averages, and one or some of those years have low data coverage. For seasonal averages, months are not given equal weight
unless stated otherwise. For example, all non-missing data for a given hour across months of the season are considered equally
(e.g., that there may be more data at noon in July than August is not considered in a summertime average).
**5 Results**
Figure 1 shows monthly mean observed ozone deposition velocities ($v_d$) across years, as well as multiyear averages, at all sites.
There are a variety of seasonal patterns and magnitudes of observed $v_d$ across sites. Interannual variability is strong in terms of
the standard deviation across yearly annual averages normalized by the multiyear average (range of 10% to 60% across sites). In
some cases, periods with low data coverage contribute to apparent interannual variability and/or seasonality. However, more
complete ozone flux records also show strong variability from year to year and month to month. The following focuses on
multiyear averages, but we briefly examine summertime (June-August) interannual variability at sites with three or more years of
data to establish whether models capture the range of interannual variability and/or ranking among different summers.




Figure 2 shows multiyear monthly mean $v_d$ from observations and the spread across models, whereas Figure 3 shows multiyear
monthly mean values from each model and observations. We first consider model ensembles. Across models, minimum and
maximum averages bracket observations across sites except Auchencorth Moss (all months except July), Borden Forest (October-
November only), and Ispra (October-February only). In some cases, model outliers allow the full set of models to bracket observations
(Fig. 3). If we instead consider the interquartile range across models (hereinafter, 'the central models'), then there are at least a few
months at every site when observations fall out of range. At the same time, at every site except Auchencorth Moss, there are also at least a
few months when the observations are within the range, indicating that failure of central models to capture observations consistently
across the seasonal cycle does not suggest a complete lack of skill from the model ensemble that de-emphasizes outliers. Further, central
models are very close to bracketing observations across months at Easter Bush, Hyytiälä, and Harvard Forest.


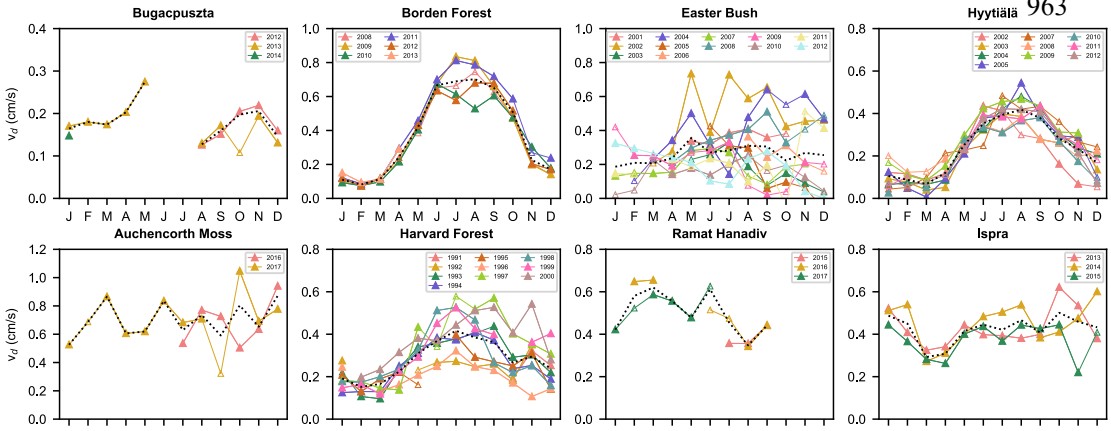

Figure 1 Monthly mean ozone deposition velocities ($v_d$) from the ozone flux observations. Multiyear average is in black. Open
symbols indicate months for a given year with low data capture. Note different y-axis ranges among panels.
The model spread in multiyear mean $v_d$ across months and sites is large (Fig. 2). The spread in terms of the model with the highest
annual average divided by the model with the lowest ranges from 1.8 to 2.3 except Hyytiälä (2.7) and Auchencorth Moss (5). The spread
in wintertime (December-February) averages is very high at some sites: Borden (10), Hyytiälä (21), Auchencorth Moss (9.1), and
Harvard Forest (6.3). The spread in wintertime averages is 2 to 3.3 at other sites. The spread is typically lower during summer (June-
August) than winter, on par with annual values. We also use the 75[th] percentile divided by the 25[th] percentile as a metric of the spread.
This metric for the annual average is 1.2–1.8. For winter, the metric is also lower for sites with high spreads based on all models: 3 for
Borden Forest, 2.4 for Hyytiälä, 3 for Auchencorth Moss, and 2.7 for Harvard Forest.

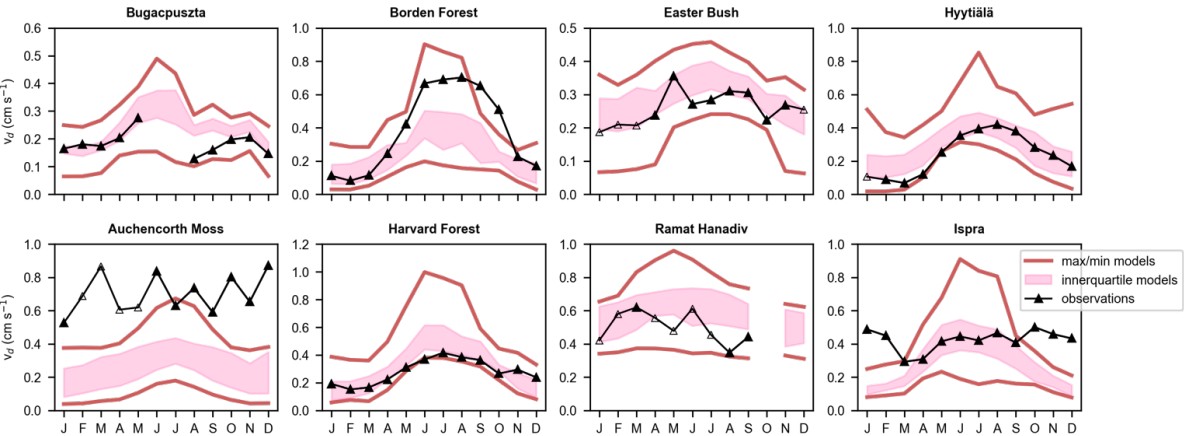

Figure 2 Multiyear monthly mean ozone deposition velocities ($v_d$) from ozone flux observations and the spread across the single-point models. Pink shading denotes the interquartile range across models. Red lines denote the minimum and maximum across monthly simulated values. Open symbols on observations indicate months with low data capture. Note different y-axis ranges among panels.

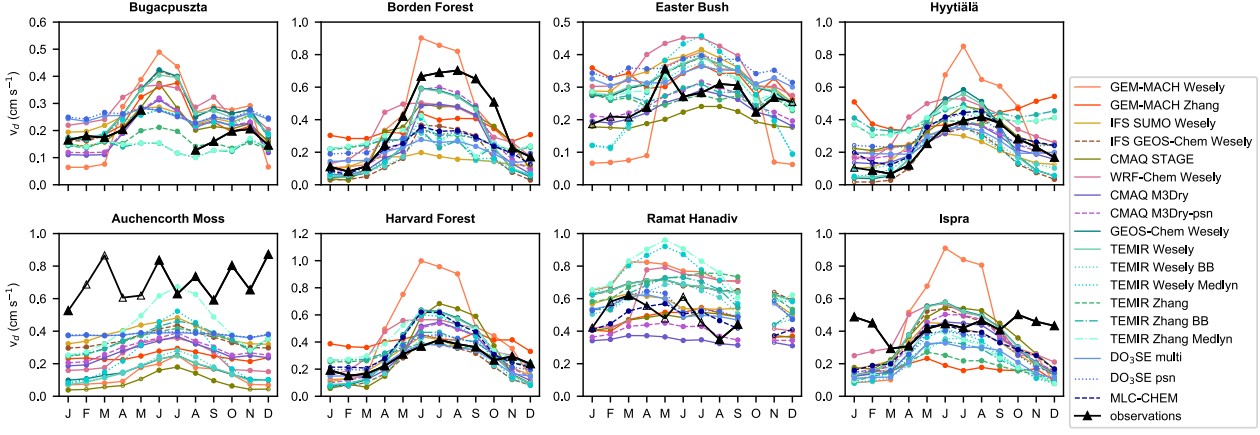

Figure 3 Multiyear monthly mean ozone deposition velocities ($v_d$) from ozone flux observations and individual single-point models. Open symbols indicate months with low data capture. Note different y-axis ranges among panels.

If we consider individual model performance, then we find that no model is always within 50% of observed multiyear averages across sites and seasons (Fig. 4). Models are very low against observations at Auchencorth Moss, but the previous statement holds even excluding this site. In general, a key finding is that model performance varies strongly by model, season, and site. Below, we first discuss mean absolute biases across sites, and then drivers of seasonality across models and sites. Then, in subsections, we discuss each site, starting with short vegetation, and then forests.




The mean absolute bias (simulated minus observed) across multiyear seasonal averages and sites is highest for GEM-MACH
Wesely (0.22 cm s$^{-1}$) and lowest for CMAQ M3Dry-psn (0.12 cm s$^{-1}$). GEM-MACH Zhang, WRF-Chem Wesely, GEOS-Chem
Wesely, TEMIR Wesely, TEMIR Wesely BB, and TEMIR Wesely Medlyn are on the higher end of the spread in mean absolute
bias across seasons and sites (0.17–0.18 cm s$^{-1}$), while DO$_3$SE multi, DO$_3$SE psn, and IFS SUMO Wesely (0.13 cm s$^{-1}$) and
CMAQ M3Dry (0.14 cm s$^{-1}$) are on the lower end, with the rest in between (0.15–0.16 cm s$^{-1}$). (MLC-CHEM does not simulate
three sites so we exclude it here).

Annual mean absolute biases may overemphasize model performance when $v_d$ are high. Given that wintertime $v_d$ tends to be lower in
magnitude than during other seasons, we also examine wintertime mean absolute biases across sites. Values are highest for GEM-
MACH Zhang (0.22 cm s$^{-1}$), GEM-MACH Wesely (0.20 cm s$^{-1}$), TEMIR Wesely (0.20 cm s$^{-1}$), and TEMIR Wesely Medlyn
(0.19 cm s$^{-1}$). Otherwise, model biases are below 0.16 cm s$^{-1}$.

Figure 5 shows simulated multiyear wintertime and summertime mean effective conductances, as well as the observed multiyear seasonal
average $v_d$ (recall that simulated effective conductances sum to simulated $v_d$). The three main pathways are stomata, cuticles, and soil;
even when models simulate lower canopy uptake, uptake via this pathway tends to be low. We thus focus on stomatal, cuticular, and soil
pathways. There are three important takeaways from Figure 5. First, models can disagree in terms of relative contributions from
pathways, even when they predict similar $v_d$. Conversely, models can agree in terms of relative contributions of pathways but
predict different $v_d$. Second, both stomatal and nonstomatal pathways are important for $v_d$ across models, as well as key drivers
of variability across models. Third, models tend to disagree on cuticular vs. soil contributions to nonstomatal uptake at some sites, while
agreeing at others.

Figure 6 shows how multiyear mean seasonality of effective conductances contributes to the multiyear mean seasonality of simulated $v_d$
across models. Specifically, the variance in each pathway across months is shown, as well as twice the covariance between individual
pathways. Negative covariances imply offsetting seasonality between the two pathways (i.e., an anticorrelation in seasonal cycles of two
pathways, and this acts to dampen the total seasonality). Positive covariances mean that a positive correlation in seasonal cycles of the
two pathways acts to amplify total seasonality. Values are normalized by the absolute sum of the variance and twice the covariances so
that Figure 6 does not emphasize differences in the seasonal amplitude, rather what pathways control the seasonality.

The key finding from Figure 6 is that stomatal uptake is the most important driver of multiyear mean $v_d$ seasonality for most models and
sites. For some models and sites, cuticular uptake also plays a role, albeit mostly just via correlations with stomatal uptake. Correlations
between stomatal and cuticular pathways are mostly positive, and thus tend to amplify $v_d$ seasonality. Exceptions are Hyytiälä and
Easter Bush where some models show anticorrelations between stomatal and cuticular uptake seasonal cycles. With a few exceptions
(e.g., at Easter Bush and for GEM-MACH Wesely and DO$_3$SE models), soil uptake tends to play a more minor role.




In general, parameters and dependencies driving simulated $v_d$ seasonality are model dependent. Expected dominant influences include
changes in initial resistances with season, cuticular and stomatal dependencies on $LAI$, stomatal dependencies on soil moisture,
temperature response functions (used in Wesely (1989) to decrease nonstomatal deposition pathways at cold temperatures), and
changes with snow. Multiyear monthly mean observed and simulated $v_d$ generally increases with $LAI$ across sites during at least some
time periods of plant growth (Fig. 7). In general, however, the relationship between $v_d$ and $LAI$ on monthly timescales is nonlinear for
both observations and models, distinct between observations vs. models, and distinct across models. Many models show a strong
sensitivity to $LAI$, which has been pointed out in previous work (Cooter and Schwede, 2000; Charusombat et al., 2010; Schwede
et al., 2011; Silva and Heald, 2018). Our analysis here, combined with past work, suggests that predictive ability hinges on better
understanding of observed $v_d$-$LAI$ relationships in terms of seasonality and site-to-site differences.

Figure 8 shows snow's impact on multiyear mean $v_d$ at sites with snow depth records and sufficient snowy periods. Observations suggest
modest reductions with snow at Bugacpuszta and Hyytiälä, but not much change at Borden Forest. At Borden Forest, some models show
decreases, while others show little change. At Hyytiälä and Bugacpuszta, some models capture decreases with snow despite biases
whereas other models understate or exaggerate decreases. Observed reductions with snow are larger at Bugacpuszta than Hyytiälä, and
many models capture this. Findings with respect to Borden Forest may reflect that snow is not measured there, rather 15 km away, and
thus this not reflect local conditions exactly. Even though some models do not capture the magnitude of observed $v_d$ decreases with
snow, Figure 8 shows that models' inability to capture the magnitude of wintertime values (snow or snow-free) at a given site is a much
larger problem than models' inability to capturing responses to snow, at least at these three sites. The relative model spread (based on the
standard deviation across models divided by the average) does not change substantially under snowy vs. all conditions, except at
Bugacpuszta (27% vs. 70%), further underscoring the need to better understand wintertime $v_d$ in a more general sense.

The relatively low magnitude of snow-induced observed $v_d$ changes indicates that snow-induced changes are not the main driver of
observed $v_d$ seasonality (Fig. 8). For example, observed changes with snow are a small fraction of the observed absolute seasonal
amplitude of multiyear monthly averages at these sites, at least for Hyytiälä and Borden Forest. We also note that models simulate $v_d$
reductions with snow at Hyytiälä and Bugacpuszta even when snow is not model input, suggesting that other model dependencies (e.g.,
temperature response functions) may lead to changes coincident with snow. Recent papers suggest that better snow cover representation
may be key for $v_d$ spatial variability at regional scales and seasonal cycles as well as changes with climate change (Helmig et al., 2007;
Andersson and Engardt, 2010; Matichuk et al., 2017; Clifton et al., 2020b). Despite insufficient data to examine spatial variability or
responses to climate change, our analysis suggests drivers of wintertime $v_d$ other than snow are important to understand.





Figure 2 Relative biases (simulated minus observed divided by observed) across models, sites, and seasons for ozone deposition
velocities ($v_d$), expressed in fractions. Numbers next to model names in the subpanel titles are mean absolute biases across
seasons and sites in cm s$^{-1}$.

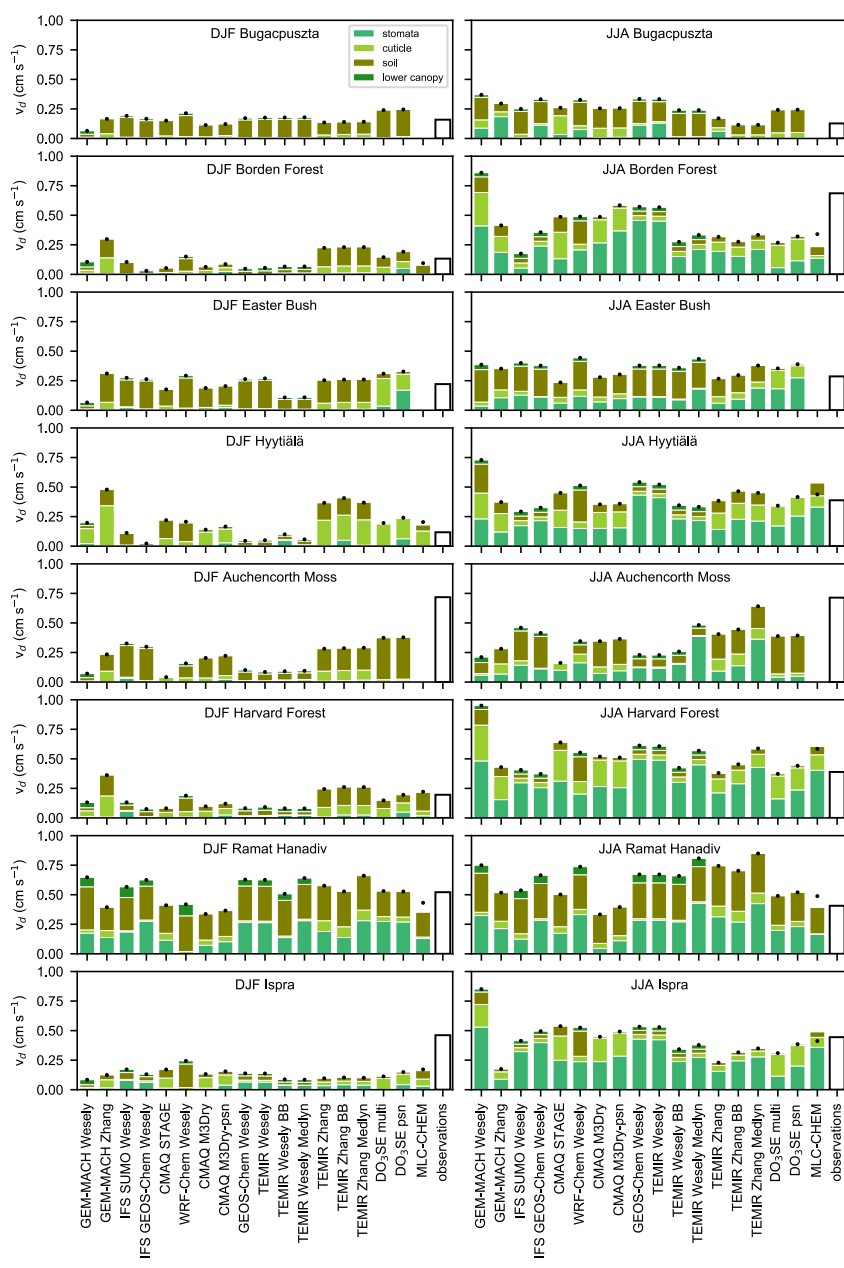

Figure 3 Multiyear seasonal mean simulated effective conductances and observed ozone deposition velocities ($v_d$). Black dots are simulated $v_d$ (black dots should equal the top of the bars). DJF is December, January, and February. JJA is June, July, and August.

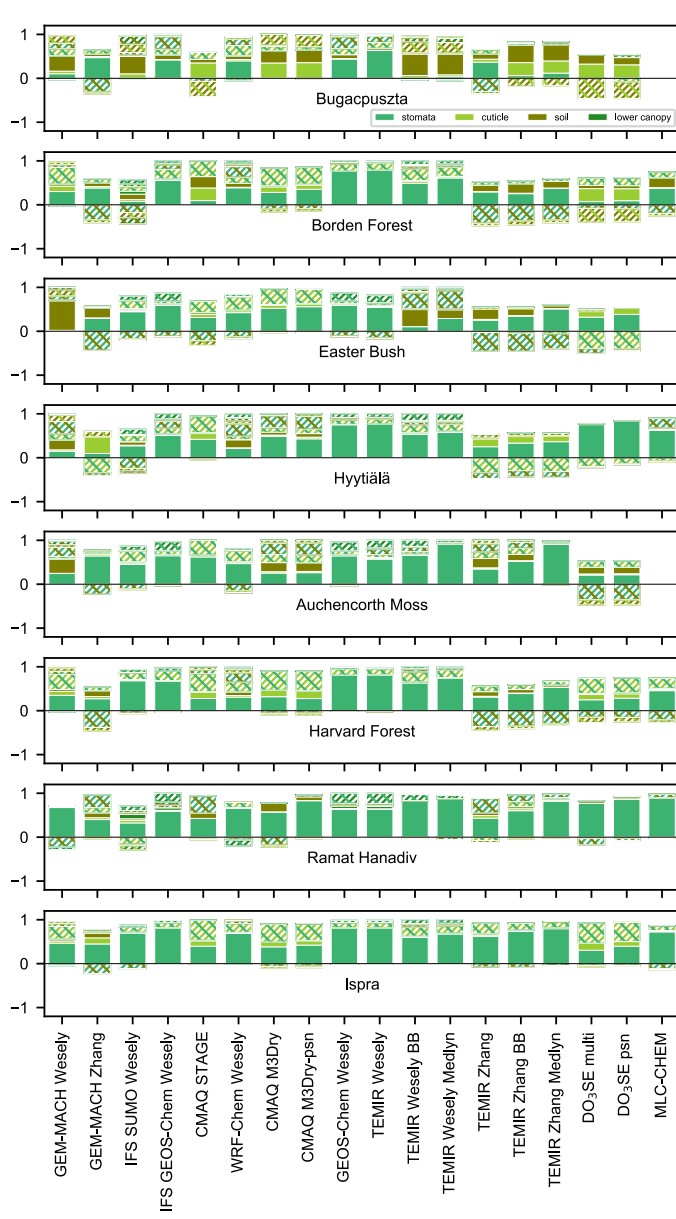

Figure 4 Pathways contributing to variability across simulated multiyear monthly mean ozone deposition velocities. The variance for each effective conductance is solid. Twice the covariance between effective conductances is hatched (the colors of hatch correspond to pathways examined). Each value is normalized by the absolute value of the sum of the variances and twice the covariances so that we are comparing the pathways that drive seasonality across models in a relative sense (rather than the seasonal amplitude as well).

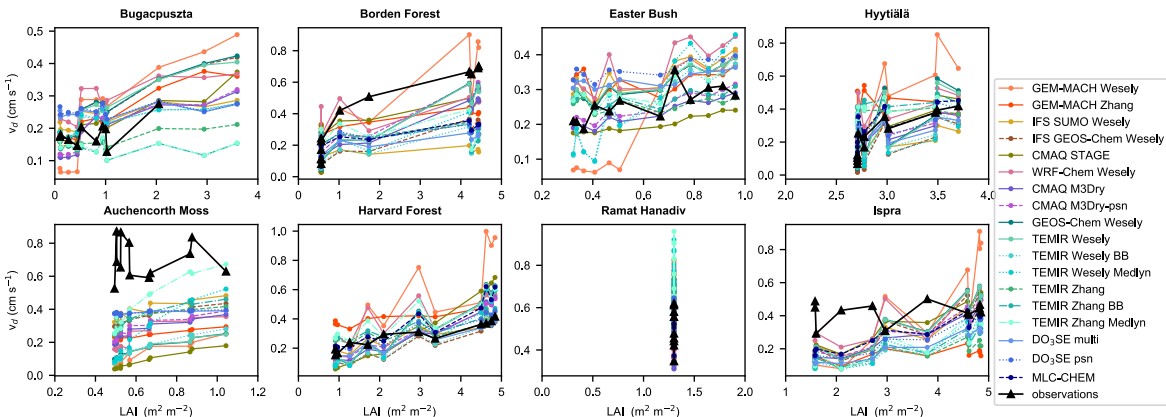


Figure 5 Multiyear monthly mean ozone deposition velocities ($v_d$) versus leaf area index (*LAI*).
**5.1 Bugacpuszta**
Bugacpuszta is a semi-arid and semi-natural grassland in Hungary. In terms of variability across models, the model spread based
on the model with the highest annual average $v_d$ divided by the model with the lowest is 2.1 (2.8 during summer and 2.2 during
winter) but based on the interquartile range is 1.3 (1.2 during summer and 1.3 during winter). The model spread at Bugacpuszta
is on the lower end of the estimates across sites examined.

A longer ozone flux record data is needed to assess interannual variability at Bugacpuszta. This site has only a single year of data
during February–May (2013), two years of data during August–December (2012 and 2013), and two years of data during January
(2013 and 2014) (Fig. 1). Data is always missing during June and July. For time periods with two years of data, observed
monthly mean $v_d$ are very close in magnitude between years. The exception is October when 2013 values are half of the 2012
values. However, October 2013 has very low data coverage (only ~2–3 days of coverage), and hourly values show high
uncertainty compared to other months (not shown). We thus focus below on 'multiyear averages', acknowledging that there are
only two years of data during six months of the year (with ten months total with data).

Without June and July observations, we cannot fully assess seasonality at Bugacpuszta. Instead, we evaluate seasonality across
other months. Thus, the observed seasonal cycle is: $v_d$ maximizes during May, following an increase from March, and minimizes
during August, after which $v_d$ increases to November and levels off from December–February (Fig. 1). Seasonal patterns are
similar across many models, with mid-summer peaks after slow increases from winter and similar values from August–
November (Fig. 3). Despite similar seasonal patterns across models as well as fair agreement in the relative seasonal amplitude
(Fig. 9), models disagree with respect to pathways dominating the seasonal cycle (Fig. 6). Notably, models disagree most in
terms of pathway(s) driving seasonality at Bugacpuszta relative to other sites, suggesting that changes in individual pathways on
seasonal timescales at this location may be a key uncertainty.

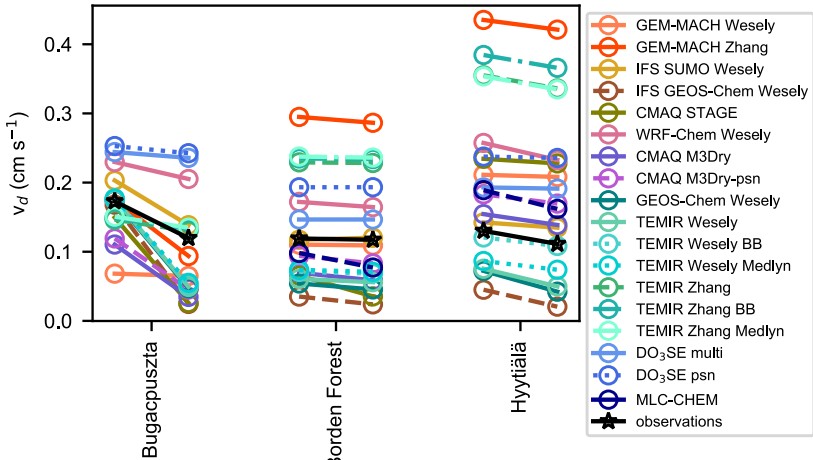

Figure 6 Multiyear mean ozone deposition velocity ($v_d$) for all conditions versus when snow depth greater than or equal to 1 cm
for sites with snow depth records and sufficient time with snow (25% averaged across hours per month). Months considered are
December-February for Bugacpuszta, December-February for Borden Forest, and November-March for Hyytiälä. Months are
given equal weight in averages.
Central models bracket observed $v_d$ during December–May but are too high during August and September (and only slightly too
high during October and November) (Fig. 2). Two clear model outliers during warm months are TEMIR Zhang models (Fig. 3),
which show relatively low soil and cuticular uptake (Fig. 5). TEMIR psn also show no stomatal uptake, following very low input
root-zone soil moisture (below prescribed wilting point). At the same time as TEMIR Zhang models are clear model outliers
during warm months, they allow the complete set of models to bracket observations during August-November, as others are
mostly too high (or in a few cases just right). Without June and July ozone fluxes, however, it is unclear how TEMIR Zhang
models alter summertime performance of the model spread.

Only eight models show substantial summertime stomatal uptake at Bugacpuszta (Fig. 5). There is no summertime stomatal
uptake simulated by TEMIR psn, IFS SUMO Wesely, and DO₃SE models, and very little by CMAQ M3Dry and CMAQ
M3Dry-psn. Only these models simulate dry deposition at this site and employ soil moisture dependencies on stomatal
conductance. They simulate little-to-no stomatal uptake at Bugacpuszta because input soil moisture is below prescribed wilting
point. We emphasize that wilting point, which is not measurable, is uncertain across sites. Models with substantial summertime
stomatal uptake show a large spread in stomatal fractions of $v_d$ – from 12.5% to 40% with one model simulating 60% (Fig. 12) –
and produce distinct stomatal uptake seasonal cycles (Fig. 10). Many models show similar $v_d$ seasonal cycle shapes (Fig. 3) but
dissimilar stomatal uptake seasonal cycle shapes, suggesting that nonstomatal uptake seasonality plays a role in normalizing
differences in $v_d$ seasonal cycles across models.






Bugacpuszta has the most similar summertime model spreads for the top three pathways as compared to other sites (except
Hyytiälä) (Fig. 11), suggesting a high degree of uncertainty in the magnitude of all pathways during warm months. Most models
show substantial summertime contributions from soil uptake, but the magnitude of soil uptake varies across models (Fig. 5). In
contrast, for summertime cuticular and stomatal pathways, models disagree as to whether contributions are substantial in addition
to the magnitude of uptake. For example, like how some models show very low stomatal uptake (as discussed above), some
models show negligible cuticular uptake. Establishing whether there should be summertime stomatal and/or cuticular uptake at
Bugacpuszta would be a first step towards further constraining models.

Multiyear monthly mean *LAI* shows a sharp summer peak, maximizing during June (~3.6 m$^2$ m$^{-2}$) (Fig. 10). Values are similar
during August to November, and then decreases from November to March, with a minimum during March. Observed $v_d$ is
missing for *LAI* greater than 2 m$^2$ m$^{-2}$ (corresponding to June and July). There is no discernable observed $v_d$-*LAI* relationship for
*LAI* below 1 m$^2$ m$^{-2}$, and models capture this (Fig. 7). Observations show a strong $v_d$ increase from 1 to 2 m$^2$ m$^{-2}$. Models show
an increase, but most do not capture the large observed slope. This is especially true for models with soil moisture dependencies
on stomatal conductance, implying that during at least some periods of high vegetation density, there should not be soil moisture
stress, or as strong of soil moisture stress as simulated by some models.

Models simulate that soil uptake dominates wintertime $v_d$ (Fig. 5). The exception is GEM-MACH Wesely, which
underestimates wintertime $v_d$. Wintertime stomatal fractions can be up to 10% due to low $v_d$ but are mostly within 0–5%.
Because central models capture wintertime $v_d$ (Fig. 2), and models agree that soil uptake dominates, some models may have
some skill during cooler months at Bugacpuszta. There is variability in soil uptake across models (Fig. 11), however. Models
largely capture observed wintertime $v_d$ decreases with snow, with most slightly overestimating the change but a few (DO₃SE models,
WRF-Chem Wesely, TEMIR Zhang, GEM-MACH Wesely) underestimating it (Fig. 8). Future attention to non-central models should
focus on capturing wintertime nonstomatal uptake generally, rather than changes with snow.

A key outstanding question at Bugacpuszta is: should models simulate low stomatal uptake throughout summer, or only during
late summer? Most models are too high against observations during August and September. This includes models employing soil
moisture dependencies on stomatal conductance (and thus simulate very-low-to-no stomatal uptake), implying too-high
simulated nonstomatal uptake. Continuous year-round ozone flux observations, especially during periods of the growing season
with and without moisture stress, are needed to better assess model performance at Bugacpuszta. Independent measures of
stomatal conductance during periods of missing ozone fluxes would be useful in constraining the absolute stomatal portion of dry
deposition, but further constraining nonstomatal uptake, which models indicate is an important fraction of summertime $v_d$
(despite disagreeing on the exact pathway), requires additional ozone flux measurements.

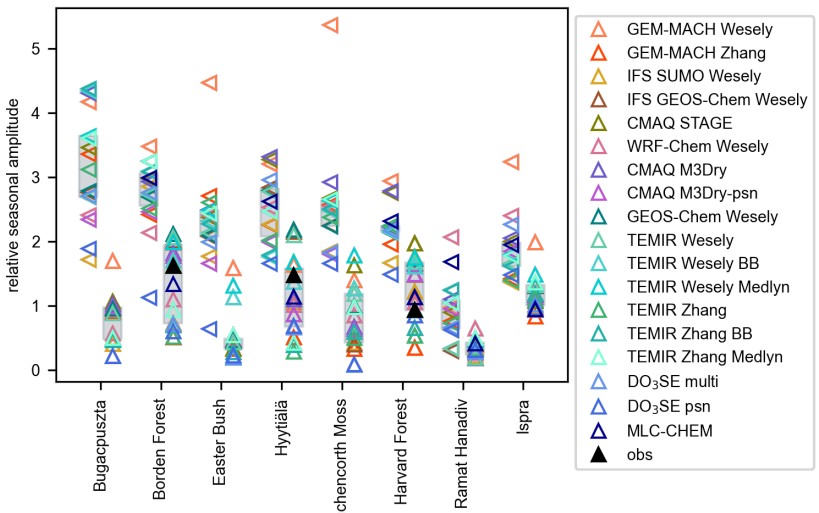


Figure 7 Relative seasonal amplitudes of multiyear monthly mean stomatal uptake (sideways triangles) and ozone deposition
velocities (upwards triangles) across models, defined as the maximum across months of multiyear monthly averages minus the
minimum, divided by the average. Black triangles denote the relative seasonal amplitude of observations for sites with
wintertime minima and summertime maxima. Grey shading denotes the interquartile range across models.
**5.2 Auchencorth Moss**
Auchencorth Moss is a peat bog covered with heather, moss, and grass in Scotland. The model spread in terms of the model with
the highest annual average $v_d$ divided by the model with the lowest is 5 (4.3 during summer and 9.1 during winter) but based on the
interquartile range is 1.6 (1.5 during summer and 3 during winter). Across sites, for the annual metrics, Auchencorth Moss has
the largest spread for the maximum/minimum metric and the second largest for the interquartile range.

There is no clear shape of the observed $v_d$ seasonal cycle at Auchencorth Moss (Fig. 1). Whether this is true on a climatological
basis is unclear due to data incompleteness – observed values during February–May have low data capture mostly because data
are missing during 2016 – as well as strong interannual variability and only two years of data. A longer and more complete ozone
flux data is needed to fully assess interannual variability as well as seasonality at Auchencorth Moss. We focus below on
'multiyear averages', acknowledging that only half the months of the year have two years of data.

A key finding for Auchencorth Moss is that models do not capture high observed $v_d$ year-round (Fig. 2). The exception is
TEMIR Zhang Medlyn during July. This is the only site examined with negative biases (> 30% of observed multiyear seasonal
averages) across seasons and models (except for TEMIR Zhang Medlyn during July) (Fig. 4). Biases tend to be smallest during
summer and largest during winter because many models simulate peak $v_d$ during warm months (Fig. 3). Notably, models differ



substantially in their relative seasonal amplitudes, with a very even and wide distribution across models (Fig. 9), especially
relative to other short vegetation sites.

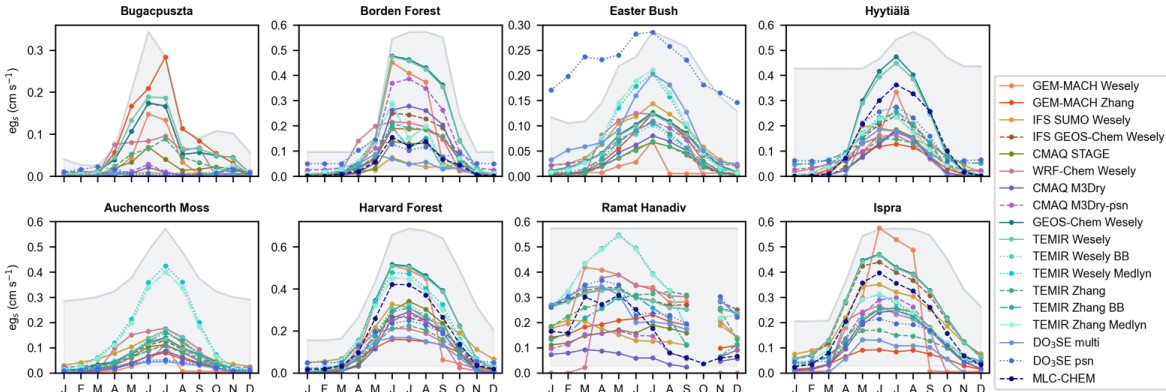


Figure 8 Multiyear monthly mean effective stomatal conductance ($eg_s$) from single-point models. Grey shading denotes
multiyear monthly mean leaf area index (used to emphasize seasonality in this variable; y-ranges not given). Note different y-
axis ranges for $eg_s$ among panels.
Simulated $v_d$ seasonality is mostly due to stomatal uptake (Fig. 6). Some models show that soil uptake plays a role, and all but
two models show moderate contributions from correlations between pathways. The seasonality shape and magnitude of stomatal
uptake is very similar across most models (Fig. 10). Major exceptions are TEMIR Medlyn models, which show peak values
around 0.4 cm s$^{-1}$ in contrast to the rest that average just under 0.1 cm s$^{-1}$. For the relative seasonal amplitudes in stomatal uptake,
the spread across central models is low (Fig. 9). The value for GEM-MACH Wesely is very high (> 5), with other models'
values spanning 1.75 to 3. Models deviating from the rest with respect to stomatal uptake's seasonality shape are GEM-MACH
Zhang (near-zero during August and after; strong peak during July) and DO$_3$SE (low during summer) as well as WRF-Chem
Wesely and IFS SUMO Wesely (the latter two are similar and higher than others especially during spring).

While high summertime stomatal uptake combined with moderately high year-round nonstomatal uptake distinguishes TEMIR
Zhang Medlyn from others (Fig. 5), we see the best agreement between this model and observations during warm months.
However, TEMIR Zhang Medlyn does not capture observed seasonality (or lack thereof). TEMIR Zhang Medlyn may have more skill
during summer than other models, but like other models, TEMIR Zhang Medlyn struggles with seasonality.  Future work should
establish whether there is strong seasonality in stomatal uptake coupled with offsetting seasonality in nonstomatal uptake at Auchencorth
Moss, or whether stomatal uptake should be higher year-round.

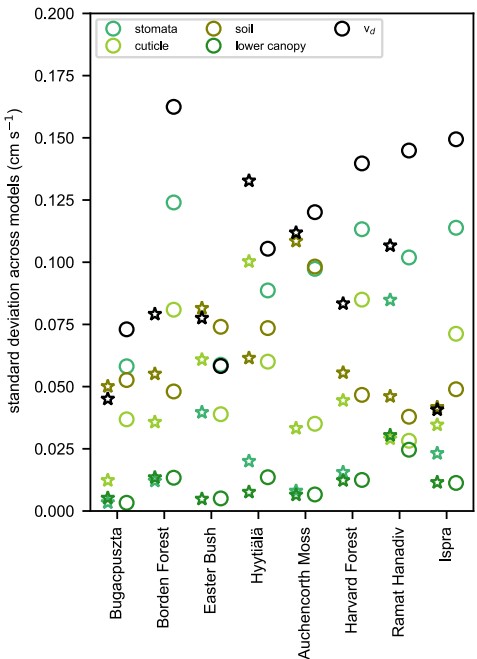


Figure 9 Model spread (standard deviation) across multiyear seasonal mean ozone deposition velocities ($v_d$) and effective
conductances for DJF (stars) and JJA (circles). DJF is December, January, and February. JJA is June, July, and August.

For soil uptake, the model spread is strong and similar during summer and winter (Fig. 11). During summer, the spread in

stomatal uptake is on par with soil uptake; spreads for stomatal and soil uptake are highest across pathways. During winter, the

spread in stomatal uptake is very low, and the spread in soil uptake is highest. Wintertime stomatal fractions vary from 0% to

20% across models (Fig. 12). Models except CMAQ STAGE simulate non-negligible soil uptake (Fig. 5). However, during

summer, models disagree on soil contribution to $v_d$ (0–80%) as well as the magnitude of soil uptake.  In contrast, during winter, models

agree that soil uptake contributes substantially (>60%) (apart from CMAQ STAGE and GEM-MACH Wesely) but disagree on

the magnitude of soil uptake. Snow depth is measured at Auchencorth Moss, but data are missing for half the ozone flux period,

and there is not a substantial amount of time with snow when there are measurements. We do not expect a large impact on

simulated values by accounting for snow throughout the ozone flux period.

196

Models estimate very-low-to-moderate cuticular uptake at Auchencorth Moss (Fig. 5), which is consistent across low vegetation

sites. Moderate values of cuticular uptake are simulated by GEM-MACH Zhang and TEMIR Zhang models, and values are

similar between summer and winter. Otherwise, models simulate very little cuticular uptake during winter and low cuticular

uptake during summer. Nonetheless, the model spread in cuticular uptake is similar between seasons. Summertime stomatal

fractions vary across central models from 25% to 55% (Fig. 12). Aside from one model simulating 80% and two models around



10%, half are around 20–30% and the other half are around 45–60%. There is a division across models in that no model
simulates stomatal fractions between 32.5% and 45%. The dichotomy seems to be due to variability in both stomatal and soil
uptake across models, consistent with high summertime model spreads for these pathways (Fig. 11).

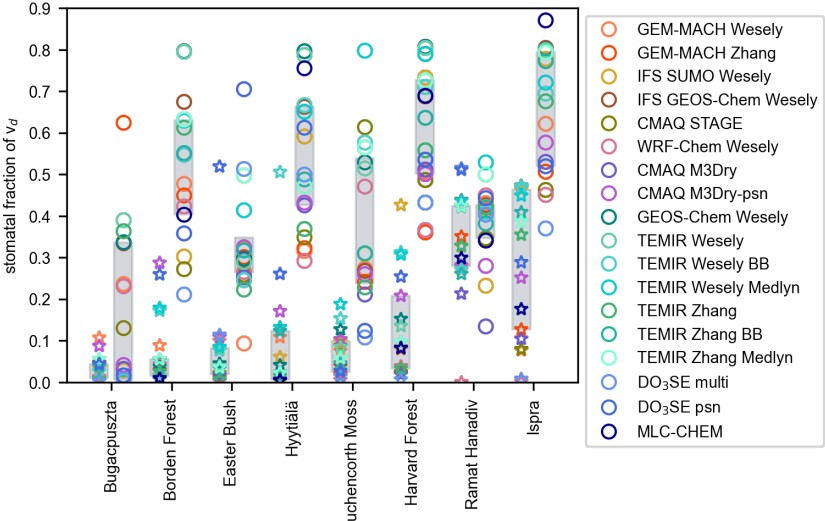


Figure 10 Multiyear seasonal mean stomatal fractions of ozone deposition velocities ($v_d$) across models during DJF (stars) and
JJA (circles). Grey shading denotes the interquartile range across models. DJF is December, January, and February. JJA is June,
July, and August.
Despite an unclear observed $v_d$ seasonal pattern, the relationship between monthly mean *LAI* and $v_d$ may provide insights into
model performance. With strong observed $v_d$ variations at low *LAI* (less than 0.6 m$^2$ m$^{-2}$), there is thus relationship, but there is a
positive relationship at moderate *LAI* (in the range of 0.6 to 0.9 m$^2$ m$^{-2}$) (Fig. 7). Observations then show that $v_d$ decreases with
*LAI* increases above 0.8 m$^2$ m$^{-2}$ but there is only one data point here. Most models seem to capture the observed relationship at
moderate *LAI* as well as that there should not be a relationship at low *LAI*. Some models (e.g., TEMIR models) overestimate the
increase's slope at moderate *LAI*, though. Thus, some models may have some skill at simulating seasonality in cuticular and/or
stomatal uptake. Nonetheless, strong observed $v_d$ variability at low *LAI* and changes with *LAI* during peak vegetation density need better
understanding. With observational constraints on stomatal uptake, we will be able to understand whether nonstomatal uptake should be
higher year-round and/or seasonality in nonstomatal uptake should act to offset seasonality in stomatal uptake.

We close by emphasizing that very high observed $v_d$ at Auchencorth Moss are uncertain – there is strong interannual and day-to-day
variability, but a lot of missing data. The peat/bog LULC type does not have many ozone flux measurements at other sites that
could be used to provide additional context to Auchencorth Moss measurements. Schaller et al. (2022) show that $v_d$ ranges from



0.05 cm s$^{-1}$ at night to 0.45 cm s$^{-1}$ during the day in July 2017 at a peatland in NW Germany. El Madany et al. (2017) look at
ozone fluxes at the same site during 2014 but does not present $v_d$. Fowler et al. (2001) present older measurements at
Auchencorth Moss, estimated with the gradient technique (eddy covariance is used for the data examined here), showing much
lower observed $v_d$ than examined here (e.g., winter and fall values here are twice what they are during 1995-1998, summer are
almost twice, and spring are higher but not twice). It is not clear what drives higher, more recent $v_d$ measurements at
Auchencorth Moss analyzed in this study and more detailed analysis is needed to figure it out. In general, building understanding
of ozone dry deposition at this LULC type provides a key test of understanding of soil uptake, and its dependence on its expected
drivers (soil organic carbon and water content), given peat/bog soils are organic rich and wet.
**5.3 Easter Bush**
Easter Bush is a managed grassland used for silage harvest and intensive grazing in Scotland. In terms of variability across
models, the spread based on the model with the highest annual average $v_d$ divided by the model with the lowest is 1.8 (1.8 during
summer and 3.0 during winter) but based on the interquartile range is 1.3 (1.3 during summer and 1.4 during winter). Model
spreads at Easter Bush are some of the lowest compared to other sites.

Easter Bush has one of the longest ozone flux records (Clifton et al., 2020a), and the longest record examined here as well as
strongest interannual variability. For example, the coefficient of variation across years is on average 60% across months. In
contrast, other sites show coefficients of variations across years from 10% to 30%. There is also strong interannual variability in
the observed seasonal cycle's shape at Easter Bush (Fig. 1). As for other sites with long term records, we focus on multiyear
averages but touch on summertime interannual variability. Some models capture some low summers, but models do not capture
high summers (except GEOS-Chem Wesely, IFS GEOS-Chem Wesely, and TEMIR Wesely, which capture one high year) and
underestimate interannual spread (Fig. 13). Future work should focus on understanding observed interannual variability, and
consider that interannual variability changes strongly by month, both in terms of the spread across years and ranking of years.

The central models' spread largely brackets observed multiyear monthly values across months. Specifically, observed values sit
mostly on the lower end of or just below the central models' spread, except during May, November, and December when
observed values are on the higher end (Fig. 2). Only CMAQ STAGE consistently shows lower $v_d$ than observed, but the relative
bias is low (-18% to -30%) (Fig. 4). During winter, GEM-MACH Wesely and TEMIR Wesely psn are too low, and the relative
biases are substantial (-51% to -70%). With a few exceptions (i.e., winter for GEM-MACH Wesely and TEMIR Wesely psn,
summer for WRF-Chem Wesely and TEMIR Wesely Medlyn), models are within ±50% of observed seasonal averages.

Overall, the below suggests that models may have skill at simulating climatological $v_d$ seasonality at Easter Bush, aside from a
clear set of outliers. There is a weak warm-season peak in observed $v_d$ (Fig. 3). Models show weak warm-season maxima and
relatively similar relative seasonal amplitudes (Fig. 9). Some models are clear outliers, however. For example, GEM-MACH
Wesely and TEMIR Wesely psn show particularly strong relative seasonal amplitudes (Fig. 9), in part due low wintertime $v_d$.



The absolute standard deviation across models for $v_d$ is higher during winter than summer (Fig. 11). This only happens at Easter
Bush and Hyytiälä; however, as noted above, the wintertime model spread reduces when considering the full vs. interquartile
range, suggesting that low outliers may drive the large standard deviation across models.

For most models, the primary driver of $v_d$ seasonality is stomatal uptake (Fig. 6). Individual contributions from stomatal uptake
barely contribute for GEM-MACH Wesely, TEMIR Wesely, and TEMIR Wesely BB. Several models, including GEM-MACH
Wesely, GEM-MACH Zhang, and TEMIR Wesely models, and to a lesser extent some TEMIR Zhang models, simulate large
contributions from soil uptake individually and/or via correlations with other pathways. Only two models, in contrast to seven at
the other grassland examined (Bugacpuszta), suggest that individual contributions from cuticular uptake matter for seasonality.

Most models are similar in terms of magnitude and seasonality shape of stomatal uptake (Fig. 10), as well as relative seasonal
amplitudes (Fig. 9). Exceptions are GEM-MACH Wesely (a very strong peak during July and is near zero after July; and thus
shows an anomalous seasonal amplitude), TEMIR Medlyn (much higher than other models during warm months), as well as IFS
SUMO Wesely and WRF-Chem Wesely (slightly higher than other models especially during spring). DO₃SE models are also an
exception – they show very different seasonal cycles from each other, despite both being high and seasonally distinctive relative
to other models. DO₃SE psn also shows an anomalous seasonal amplitude.

At Easter Bush, *LAI* peaks during July, with a broad maximum from May to November and low values during February and
March (Fig. 10). With some exceptions, models bound the observed relationship between $v_d$ and *LAI*, agreeing on a fairly weak
but positive dependence (Fig. 7). Outliers with respect to the $v_d$-*LAI* relationship (GEM-MACH Wesely and TEMIR Wesely psn)
also indicate that stomatal uptake does not strongly influence $v_d$ seasonality, suggesting the latter is incorrect.

During summer, model spreads for $v_d$ and deposition pathways are highest for soil uptake, then stomatal uptake, and then
cuticular uptake (Fig. 11). Most models simulate moderate or substantial stomatal uptake, but there is a division as to whether
models simulate very low, low, or moderate cuticular uptake (Fig. 5). Models simulate substantial soil uptake, both in terms of
absolute magnitudes and relative contributions. Exceptions are DO₃SE models, which have very low soil uptake. Stomatal
fractions range from 10% to 70%, with most models around 30% and only four models above 40% (Fig. 12). The range across
models for stomatal fractions is one of the largest across sites, but the interquartile range is one of the smallest. High agreement
in stomatal uptake magnitude, seasonality shape, and relative amplitude, as well as stomatal fractions, across most models
suggests that the next step should be to use observation-based estimates of stomatal uptake (e.g., from water vapor fluxes) to
evaluate whether models are accurate with respect to this pathway.

During winter, models simulate that $v_d$ is dominated by soil uptake, with some models simulating low-to-moderate contributions
from cuticular uptake (Fig. 5). Only DO₃SE models and GEM-MACH Wesely show little soil uptake; while soil uptake is still a





large fraction of $v_d$ for GEM-MACH Wesely, it is a small fraction for DO₃SE models. Stomatal uptake is very low except for
DO₃SE psn. Stomatal fractions are between 0% and 10% except DO₃SE psn (50%) (Fig. 12). Because models largely agree that
wintertime $v_d$ is dominated by soil uptake, and most models overestimate January–April $v_d$, but underestimate November–
December, future work should focus on changes in soil uptake on weekly to monthly timescales. We do not have snow depth
measurements at Easter Bush, but do not expect that accounting for snow would substantially impact on simulated values.
**5.4 Ramat Hanadiv**
Ramat Hanadiv is a shrubland is Israel near the Mediterranean coast. The spread based on the model with the highest annual
average $v_d$ divided by the model with the lowest is 2.2 (2.3 during summer and 2 during winter) but based on the interquartile range
is 1.4 (1.3 during summer and 1.5 during winter). Metrics are on the lower end of the cross-site range.

There are ozone flux observations at Ramat Hanadiv during January–September only, and only March, August, and September
have substantial data coverage. Three different years contribute to multiyear averages, with each year only having a few months
of data per year. For some months, years have overlapping data coverage. Some months with data for two years show interannual
variability while others do not. Like Bugacpuszta and Auchencorth Moss, more data is needed to assess interannual variability as
well as seasonality at Ramat Hanadiv. Below, we examine 'multiyear averages', acknowledging that only six months of the year
have two years of data, and three months have data from one year only.

Models show weak relative seasonal amplitudes for $v_d$ (Fig. 9). Values are very similar across models, more so than other sites.
Most models also show weak relative seasonal amplitudes for stomatal uptake, but there is a larger spread across central models
and some outliers. The lack of simulated seasonality for most models is likely due to constant $LAI$. Any simulated $v_d$ seasonality
is from stomatal uptake (Fig. 6), more so than (or in contrast to) the other short vegetation sites. GEM-MACH Wesely and WRF-
Chem Wesely, which are two of three models with input initial resistances (i.e., model parameters) varying by season, have very
distinct $v_d$ seasonal cycle shapes at this site, compared to the rest (Fig. 3).

The seasonal cycle shape of observed $v_d$ at Ramat Hanadiv is hard to discern with many months with low or no data coverage
(Fig. 1). The current set of observations indicates higher values during early spring and lower values during late summer.
Individual models do not to capture this, with models simulating near-constant values year-round or increases from winter to
early summer (Fig. 3). Exceptions are MLC-CHEM, DO₃SE models, and GEM-MACH Wesely, which at least somewhat
capture that the predominant seasonality feature should be lower late-summer values and higher early-spring values.

Across months with observations, models bracket observed $v_d$ (Fig. 2). In particular, models are within -35% to +55% of
observed seasonal averages (Fig. 4). Exceptions occur during summer and include GEM-MACH Wesely, IFS GEOS-Chem
Wesely, WRF-Chem Wesely, GEOS-Chem Wesely, TEMIR Wesely models, and TEMIR Zhang models (biases are higher than
+55%). The central models' spread only brackets observed values during January-April and June, and is too high during May



and July-September. The largest deviation happens during August. Thus, like Bugacpuszta, late summer is when the largest
model biases occur at Ramat Hanadiv.

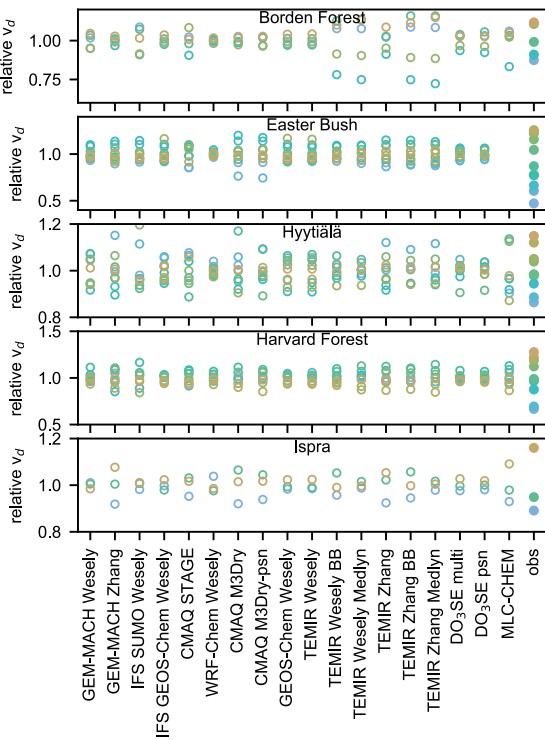


Figure 11 Simulated and observed yearly summertime mean ozone deposition velocities ($v_d$) for sites with records of at least
three summers. Values are normalized by the multiyear average of the respective model or observations to emphasize ranking
and spread across years. Colors rank yearly values from low (blue) to high (gold) for the observations. Model year when
observed year is missing is not shown. The highest year for Easter Bush is not shown because it is very high (2x the multiyear
mean observed value). Note that y-axis ranges vary among panels.
DO₃SE models, MLC-CHEM, and TEMIR psn show weak $v_d$ decreases from spring to fall. These models plus CMAQ models
consider stomatal conductance dependencies on soil moisture. CMAQ models show weaker $v_d$ declines from spring to fall,
compared to DO₃SE models, MLC-CHEM, and TEMIR psn. This behavior is consistent with their soil moisture dependencies.
For example, TEMIR psn and IFS SUMO Wesely models' stomatal conductance is set to zero when input soil moisture is less
than wilting point, but CMAQ models have more of a taper effect. Future work should aim to understand the role of soil moisture
on observed seasonal variation in $v_d$ and stomatal uptake.

Models with the highest biases during April-September are TEMIR models, GEM-MACH Wesely, WRF-Chem Wesely, GEOS-
Chem Wesely, and IFS GEOS-Chem Wesely (Fig. 3). These models simulate the highest stomatal uptake during this period,



apart from a few models with lower-than-average nonstomatal uptake (CMAQ STAGE, DO₃SE models, GEM-MACH Zhang)
(Fig. 5). Only CMAQ M3Dry models capture low observed $v_d$ during August. CMAQ M3Dry-psn captures July, but CMAQ
M3Dry does not, and they do not capture observed values during other months. Notably, CMAQ M3Dry models show much
lower summertime stomatal uptake than other models. CMAQ M3Dry models may have more skill during summer than other
models, but like the other models, they struggle with seasonality.

Lower canopy uptake is the highest for Ramat Hanadiv, both during summer and winter, across sites. However, relative and
absolute contributions of lower canopy uptake are still low compared to at least soil and stomatal uptake. Lower canopy uptake is
only simulated by Wesely models. Mostly Wesely models simulate low cuticular uptake compared to other models, so lower
canopy uptake does not necessarily contribute to the very high model biases of Wesely models.

Uptake by soil and stomata mostly comprises $v_d$ during winter and summer (Fig. 5). The model spread is highest for stomatal
uptake during winter and summer, compared to other pathways (Fig. 11). The spread for soil uptake is remarkably low given its
importance across models (less than 20% relative spread compared to mostly between 40–75% of $v_d$). Ramat Hanadiv is the
only site with a large wintertime spread across stomatal uptake estimates, and similar model ranges of stomatal fractions during
winter and summer. Models except WRF-Chem Wesely show substantial wintertime stomatal uptake. In general, stomatal uptake
is very high compared to other sites during winter, presumably due to the site's Mediterranean climate. Models also show
substantial summertime stomatal uptake except CMAQ M3Dry. Wintertime stomatal fractions range from 20% to 50% across
models (Fig. 12). The range is only slightly less across central models (25–40%), suggesting that wintertime stomatal uptake is a
key uncertainty at this site. Central models simulate a very small range of summertime stomatal fractions (similar to only Easter
Bush), centering on 40%, but the full range spans 12.5% to 50%.

At Ramat Hanadiv, most models should simulate lower stomatal and/or nonstomatal uptake during late summer, on par with
CMAQ M3Dry models, which have both lower stomatal and nonstomatal uptake than other models. However, stomatal and/or
nonstomatal uptake should be higher than simulated by CMAQ M3Dry during other times of year, and other models bracket
observations well at this time so they may provide insight here as to driving processes. Observational constraints on stomatal
uptake year-round will help to further narrow uncertainties as to whether and when models need improvement with respect to
stomatal vs. nonstomatal uptake, including when they capture the absolute magnitude of $v_d$ well.
**5.5 Ispra**
Ispra is a deciduous broadleaf forest in northern Italy. The model spread in terms of the model with the highest annual average
$v_d$ divided by the model with the lowest is 2.3 (3.1 during summer and 2.9 during winter) but based on the interquartile range is 1.5
(1.5 during summer and winter). These metrics are towards the higher end of other sites.





Observed multiyear monthly mean $v_d$ is similar year-round except during March and April when values are lower (Fig. 1). This
seasonal pattern is consistent across years except October–December. For example, observed $v_d$ is high during October 2013,
low during November 2015, and high during December 2014. As discussed below, causes of high year-round values are
uncertain; this, together with strong interannual variability during fall, indicates a need for more years of observations at Ispra,
coupled with complementary measurements targeting individual pathways. Below, we focus on multiyear averages, after briefly
evaluating summertime interannual variability.

Summertime observed $v_d$ is higher during 2014 than 2013 and 2015 (Fig. 1). Accordingly, model skill at interannual variability
should be determined by whether models capture much higher summertime average during 2014 vs. other years. Figure 13 shows
that some models suggest that $v_d$ should be highest during 2014, but hardly any models capture the large observed relative
difference between this year and other years. The exception is MLC-CHEM, and to a lesser extent GEM-MACH Zhang. Thus,
most models have little skill at simulating summertime interannual variability at Ispra.

The $v_d$ seasonality shape is a clear discrepancy between observations and models. In contrast to observations, central models' $v_d$
peaks during warm months (Fig. 2). Models show similar $v_d$ relative seasonal amplitudes, aside from GEM-MACH Wesely,
relative to other forests (Fig. 9). Central models bracket observations during April–September, but models show a low bias
during October–March. Relative summertime and springtime biases range from -33% to +32% except DO₃SE multi, TEMIR
Zhang, TEMIR Wesely BB, and GEM-MACH Zhang (lower) as well as GEM-MACH Wesely (higher) (Fig. 4). Relative
wintertime and fall biases range from -22% to -89% across models. Ispra is the only site besides Auchencorth Moss where
models are biased in the same direction for an extended period (i.e., longer than three months).

Models show that stomatal uptake largely drives $v_d$ seasonality (Fig. 6). Models simulate contributions from cuticular uptake,
mostly via positive correlations with the stomatal pathway. Models with non-zero individual contributions from cuticular uptake
(GEM-MACH Zhang, CMAQ models, and DO₃SE models) are the same as Harvard Forest and Borden Forest. Models show $v_d$
maxima during warm months because $v_d$ strongly depends on *LAI* (Fig. 7), which has a broad maximum during warm months
(Fig. 10). Specifically, simulated $v_d$ tends to increase with *LAI*, which contrasts with observed $v_d$.

A couple of models deviate from the majority in terms of $v_d$ seasonal cycles (Fig. 3). For example, GEM-MACH Zhang is low
during warm months and GEM-MACH Wesely is very high during warm months. WRF-Chem Wesely shows higher wintertime
$v_d$ than other models, especially January–March, due to high soil uptake, as well as high early-springtime uptake due to
combined high soil and stomatal uptake (Figs. 5, 10). GEM-MACH Wesely and WRF-Chem Wesely are two of three models
with input initial resistances (i.e., model parameters) varying by season, which likely causes these models to produce distinct
seasonal cycle shapes. GEM-MACH Zhang has low summertime stomatal and nonstomatal uptake, compared to the rest (Fig. 5).



Even though central models bracket observed multiyear monthly mean $v_d$ during April–September (Fig. 2), and many individual
models capture the increase from April to May, individual models fail to capture that July–September values should be roughly
constant, rather than decrease (Fig. 3). For example, some models (including DO₃SE psn, MLC-CHEM) simulate April-July
multiyear monthly mean $v_d$ very well but not August and September when they are low (because they simulate decreases from
early to late summer). Models may erroneously simulate decreases from early to late summer because they depend too strongly
on *LAI*, which weakly declines from July to September, or soil moisture.
During summer, the model spread is largest for stomatal uptake relative to other pathways (Fig. 11). Models simulate substantial
stomatal uptake, with DO₃SE multi and GEM-MACH Zhang simulating the lowest (but nonnegligible) values (Fig. 5). The
highest stomatal uptake is simulated by GEM-MACH Wesely, GEOS-Chem Wesely, IFS GEOS-Chem Wesely, IFS SUMO
Wesely, TEMIR Wesely, and MLC-CHEM. Central models show stomatal fractions of 50% to 77.5%, but the full model range is
37.5% to 87.5% (Fig. 12). The model spread across pathways is second largest for cuticular uptake. Soil uptake is very low
across models except WRF-Chem Wesely as well as CMAQ STAGE and GEM-MACH Wesely where it is higher. The ranking
and spread across pathways of pathways' standard deviations at Ispra is very similar to Borden Forest and Harvard Forest, but
not Hyytiälä. Given that central models capture the average magnitude of warm-season $v_d$ well but disagree mainly on stomatal
vs. cuticular fractions as well as monthly changes within the warm season (or lack thereof), future work should prioritize using
observational constraints on stomatal uptake to further evaluate model performance.
During winter, simulated $v_d$ tends not to be dominated by one pathway; instead, there are small contributions from 2–4 pathways
(Fig. 5). Exceptions are WRF-Chem Wesely where soil uptake dominates and a few models where cuticular uptake tends to
dominate (e.g., CMAQ STAGE, CMAQ M3Dry, DO₃SE multi). The model spread in soil uptake is largest across pathways (Fig.
11), and high WRF-Chem Wesely values play a role in this. Otherwise, soil uptake is low, or in a few cases moderately low (e.g.,
MLC-CHEM, IFS SUMO Wesely). Cuticular uptake is close behind soil uptake in terms of the spread. Stomatal fractions span
0% to 47.5%, with the largest range across central models (10–45%) across sites (Fig. 12). Eleven models show low-to-
moderately-low stomatal uptake, but others predict none (GEM-MACH Wesely, GEM-MACH Zhang, CMAQ STAGE, GEOS-
Chem Wesely, CMAQ M3Dry, TEMIR Wesely, DO₃SE multi). More models predict non-zero stomatal uptake at Ispra
compared to other sites, apart from Ramat Hanadiv. Whether simulated wintertime stomatal, cuticular, soil, and/or lower canopy
uptake should be higher at Ispra is uncertain. There may also be fast ambient losses of ozone. Ispra does not have snow depth
observations, but we anticipate that accounting for snow would not substantially change model results. Future attention should be
placed elsewhere with respect to better understanding of large wintertime model biases. A key first step is to understand whether
there is stomatal uptake during winter, and then what its magnitude is.
**5.6 Hyytiälä**
Hyytiälä is a boreal evergreen needleleaf forest in Finland. The model spread in terms of the model with the highest annual average
$v_d$ divided by the model with the lowest is 2.7 (1.9 during summer and 21 during winter) but based on the interquartile range is 1.6



(1.4 during summer and 2.4 during winter). The metrics of model spread at Hyytiälä are at the higher end of other sites' values,
especially for annual and winter values.

Observed multiyear monthly mean $v_d$ maximizes during warm months, and this is consistent across years (Fig. 1). Most models
simulate higher values during warm months relative to cool months (Fig. 3). Outliers with respect to the seasonality are TEMIR
Zhang (strong overestimate during cold months leading to near constant values year-round), GEM-MACH Wesely (strong
overestimate during warm months), GEOS-Chem Wesely and TEMIR Wesely (overestimate during summer), and WRF-Chem
Wesely (strongly overestimate during early spring). Here we examine observed relative seasonal amplitude for $v_d$ because
observed and (most) modeled values have warm-month maxima and cool-month minima as well full seasonal cycles, allowing
meaningful comparisons. The observed relative seasonal amplitude falls within the central models' range, but towards the upper
end, and most models predict too-low values (Fig. 9).

In general, the largest relative model $v_d$ biases at Hyytiälä occur during cool months (Fig. 4) and the wintertime $v_d$ model spread is
the highest relative to other sites (Fig. 11), implying that wintertime $v_d$ at this site is a key uncertainty. Wintertime relative biases range
from -81% to +87% except for a few models that have much higher positive biases: GEM-MACH Zhang (+307%), TEMIR Zhang
models (+211 to +245%), and DO₃SE psn (+104%). However, most models are biased high, apart from IFS SUMO Wesely (-5%), IFS
GEOS-Chem Wesely (-81%), GEOS-Chem Wesely (-62%), and TEMIR Wesely models (-15% to -57%). Models largely simulate that
cuticular and soil uptake are dominant contributors (Fig. 5). Most models simulate near-zero wintertime stomatal uptake, despite
relatively high *LAI* (Fig. 10), implying that models have at least rudimentary skill at capturing the seasonality of evergreen vegetation.
Central models show stomatal fractions between 0% and 12.5%, but a few models show contributions of 17.5% to 50% (Fig. 12). The
model with the 50% (TEMIR Wesely BB) in addition to very low stomatal uptake has very low nonstomatal uptake.

During winter, models also show differences in partitioning and magnitudes of cuticular vs. soil uptake (Fig. 5). The model spread in
cuticular uptake is larger than soil uptake (Fig. 11) – Hyytiälä is the only site where this happens – presumably because *LAI* remains
relatively high at this site year-round and models seem to suggest that cuticular uptake is more important than ground uptake at forests.
Ten models show substantial cuticular uptake, whereas only two models show low cuticular uptake, and the rest show none. Seven
models show substantial soil uptake, while ten show very little to none. Models showing high vs. low cuticular and soil uptake are
sometimes the same. For example, four simulate substantial cuticular uptake and soil uptake, and five simulate minimal cuticular uptake
and soil uptake. In the former case, models overestimate wintertime $v_d$; in the latter, models underestimate it. Most models capture small
observed decreases in wintertime $v_d$ with snow, but the spread across models during snow and snow-free periods is very large (Fig. 8).
Thus, attention should focus on constraining wintertime cuticular vs. soil uptake. Establishing whether there is cuticular and/or soil uptake
during winter is an important first step towards narrowing model uncertainties.





Within the warm season, whether models show pronounced $v_d$ seasonality varies (Fig. 3). Models also do not capture that
observations maximize during August and minimize during March (Fig. 2). Specifically, models tend to overestimate late-winter/spring
$v_d$ while underestimating fall/early-winter $v_d$, as indicated by comparing the interquartile range to observations. Multiyear monthly
mean *LAI* peaks during August (around 3.75 m² m⁻²), after an increase from May (Fig. 10). Then, *LAI* decreases to November,
and is constant from November to May (around 2.75 m² m⁻²). Models bound the observed $v_d$-*LAI* relationship, and largely
capture the increase from 3 to 3.5 m² m⁻² (Fig. 7). However, most models do not capture the $v_d$ change from 3.5 to 3.75 m² m⁻²
where observations suggest that the slope should be the same as for 3 to 3.5 m² m⁻² (instead models suggest decreases). Models also
overestimate the increase from 2.75 to 3 m² m⁻². Some effect overrides *LAI*'s influence on seasonality in stomatal uptake in models,
given that both observed *LAI* and $v_d$ peak during August, but simulated stomatal uptake and $v_d$ do not. Simulated declines with soil
moisture may play a role here.

Models simulate that stomatal uptake and co-variations between pathways are important seasonality drivers (Fig. 6). Only two models
suggest that there are not individual contributions by stomatal uptake (GEM-MACH Wesely, GEM-MACH Zhang), but a number of
models suggest that the sum of individual contributions from other pathways and co-variations are at least as important as stomatal
uptake. There are similarly evenly distributed spreads across models in terms of relative seasonal amplitudes for stomatal uptake and $v_d$
(Fig. 9). Most models' stomatal uptake seasonal cycles show a broad warm-season peak, apart from some models with more pronounced
seasonality during warm months (e.g., GEM-MACH Wesely, GEOS-Chem Wesely, TEMIR Wesely, CMAQ M3Dry models) (Fig. 10).
IFS SUMO Wesely peaks during May and then declines afterwards. Model outliers in terms of high magnitudes of summertime stomatal
uptake include GEOS-Chem Wesely, TEMIR Wesely, MLC-CHEM, and GEM-MACH Wesely.

During summer, relative model biases range from -14% to +20% except for GEM-MACH Wesely (+88%), IFS SUMO Wesely (-25%),
WRF-Chem Wesely (+32%), TEMIR Wesely (+34%), and GEOS-Chem Wesely (+40%) (Fig. 4). Models show substantial stomatal
uptake (Fig. 5) with stomatal fractions spanning 27.5% to 80% (Fig. 12). Central models show 42.5–65%. Models that simulate lower
canopy uptake show low uptake via this pathway, like other forests. The largest model spread is for soil and stomatal uptake, but closely
followed by cuticular uptake (Fig. 11), which is distinct from other forests. Soil uptakes' high model spread is due to large estimates from
WRF-Chem Wesely and GEM-MACH Wesely and zero soil uptake from DO₃SE models; other models simulate more similar estimates
of soil uptake, ranging from low to moderate values. Models show cuticular uptake but disagree as to whether it is low or moderate.
Observational constraints on stomatal uptake will help to further narrow uncertainties as to the magnitude and relative
contribution of summertime stomatal uptake, as well as changes on weekly to monthly timescales.

Key findings regarding seasonality at Hyytiälä include: models struggle to capture the exact timing of maximum and minimum values,
overestimate wintertime values and thus underestimate the relative seasonal amplitude, and disagree about seasonality within the warm
season, while generally capturing that there should higher values during warm months. Silva et al. (2019) use Hyytiälä observations to
train a machine learning model and apply the model to predict $v_d$ at Harvard Forest, finding that their model predicts a late summertime



peak in $v_d$, which is observed at Hyytiälä but not at Harvard Forest. Assuming that differences between these two sites are characteristic
of sites' broad LULC classifications, both our findings and theirs suggest a need for improved predictive ability of seasonality differences
between coniferous vs. deciduous forests.

Thus far we discuss multiyear averages at Hyytiälä. We turn to summertime interannual variability. Models do not capture the
summertime ranking across years (Fig. 13). Several models predict particularly low (high) $v_d$ during some summers, but these are not
low (high) summers in the observations. Some models are close to capturing the degree of summertime interannual variability, but
typically these models show a more uneven distribution across years than suggested by observations. Notably, models show more
variability in their year-to-year rankings at Hyytiälä compared to other sites with longer records. Nonetheless, we conclude that model
skill is poor at this site in terms of interannual variability.
**5.7 Harvard Forest**
Harvard Forest is a temperate mixed forest in the northeastern United States. The model spread in terms of the model with the highest
annual average $v_d$ divided by the model with the lowest is 1.9 (1.8 during summer and 4.8 during winter) but based on the
interquartile range is 1.2 (1.4 during summer and 2.6 during winter). Like other forests, the wintertime spread is largest. Aside
from winter values, the metrics of the spread at Harvard Forest are on the lower end of estimates across sites.

Observed multiyear monthly mean $v_d$ maximizes during May–September (Fig. 1). Observed seasonal cycles vary across years, but
values are generally higher during warmer vs. cooler months across years. We focus on multiyear averages until the subsection end,
where we touch on summertime interannual variability. Models capture that $v_d$ peaks during warm months (Fig. 2). The exception is
GEM-MACH Zhang, which has similar monthly averages year-round. Despite capturing seasonality shape, models overestimate the
relative seasonal amplitude (Fig. 9), apart from GEM-MACH Zhang, TEMIR Zhang, and TEMIR Zhang BB (substantial underestimate)
as well as DO$_3$SE psn (slight underestimate). Outliers show high wintertime $v_d$ relative to other models and observations, implying that
models bound the observed relative seasonal amplitude does not necessarily indicate ensemble skill.

Models are within $\pm65\%$ of observed values across seasons (Fig. 4). Exceptions occur during spring and summer for GEM-MACH
Wesely, winter and spring for GEM-MACH Zhang, and spring for WRF-CHEM Wesely and TEMIR Zhang Medlyn. Central models
bracket observations well. Specifically, observations fall in the lower end of the spread during warm months and the upper end during
November–January, but otherwise are in the middle of the spread. Across models, summertime biases are positive, ranging from +4 to
+144%, except IFS GEOS-CHEM Wesely (-4%) and TEMIR Zhang (-2%). Thus, overestimated relative seasonal amplitudes (Fig. 9) are
likely due to high summertime $v_d$. Previous work suggests that GEOS-Chem's overestimate at Harvard Forest is due to too-high model
*LAI* (Silva and Heald, 2018), but clearly there is another issue because models are forced with site-specific *LAI*. Most models tend to
underestimate $v_d$ at low *LAI* and overestimate $v_d$ at high LAI, overstating $v_d$ increases with *LAI* (Fig. 7).



During winter, model biases tend to be negative, ranging from -24% to -71%, with exceptions of GEM-MACH Wesely (+85%), TEMIR
Zhang models (+25% to +33%), and MLC-CHEM (+13%) as well as two models with very low negative biases (DO₃SE psn and WRC
Chem Wesely) (Fig. 4). The wintertime model spread is highest for soil uptake across pathways, with cuticular uptake close behind. Soil
uptake is always at least 37.5% (and up to 70%) of $v_d$ except for GEM-MACH Wesely (20%) (Fig. 5). Most models show little-to-no
stomatal uptake, but some models show nonnegligible values. Central models show stomatal fractions of 5–15% (Fig. 12). Estimates for
cuticular uptake vary – across models, there are substantial, small, and negligible contributions. Lower canopy uptake is low for models
that simulate this pathway but can be an important fraction of $v_d$. There are no snow depth observations at Harvard Forest. Assuming no
snow throughout may influence some models' ability to estimate wintertime $v_d$ well. However, based on our analysis at other sites, we
do not anticipate the lack of snow data to be the main driver of model-observation or model-to-model differences. Establishing whether
there should be stomatal or cuticular uptake during winter would be a useful first step in further constraining models. Otherwise, attention
should focus on narrowing uncertainties related to wintertime ground uptake.

Some models capture the broad observed $v_d$ maximum during the warm season while others show more seasonality within the warm
season (Fig. 3). A few models show pronounced declines after July (e.g., MLC-CHEM, TEMIR psn). Pronounced declines after July do
not occur in observed multiyear monthly averages but occur during several individual years (Fig. 1). Simulated pronounced declines may
follow these models' soil moisture dependencies (note that not all models have soil moisture dependencies, and there are differences
among models that do have them). That models with soil moisture dependencies are not capturing the observed multiyear mean
seasonality may be due to soil moisture dependencies themselves, and/or with uncertainty in soil moisture input. For example, soil
moisture was not measured during all years with ozone fluxes at Harvard Forest, and thus we use a climatological average during those
years. Future work should examine seasonality during individual years, paying attention to years with climatological average vs. year-
specific input soil moisture, to determine model strengths and limitations.

Models show stomatal uptake is an important driver of $v_d$ seasonality (Fig. 6). Six models estimate that stomatal uptake largely drives
seasonality, with some contributions from correlations (mainly positive correlations between stomatal and cuticular pathways). The rest
estimate moderate contributions from stomatal uptake, but at least as much of an influence from individual nonstomatal pathways or
correlations (positive or negative). Models show a clear seasonality to stomatal uptake, with a peak during warm months and zero or near
zero values during winter (Fig. 10). The spread for relative seasonal amplitude for stomatal uptake across central models is the smallest
across sites (Fig. 9). Six models deviate from the rest, however. CMAQ M3Dry, CMAQ STAGE, and GEM-MACH Wesely have high
relative seasonal amplitudes for stomatal uptake, GEM-MACH Zhang, IFS SUMO Wesely, and DO₃SE psn have low values. In contrast,
the spread for relative seasonal amplitude for $v_d$ has a more even distribution across models. Thus, while there is a fair amount of
agreement across models in terms of seasonality in stomatal uptake, models disagree as to nonstomatal uptake seasonality and its role on
$v_d$ seasonality. Together with findings that models exaggerate the $v_d$-*LAI* relationship and most models overestimate the relative
seasonal amplitude for $v_d$, this result implies future work should aim to better constrain nonstomatal influences on seasonality.



During summer, the model spread is highest for stomatal uptake, with cuticular uptake close behind (Fig. 11). Models show substantial
contributions from stomatal uptake -- the model range spans 30% to 80%, but the central models' range spans 50% to 70% (Fig. 12).
Estimates for cuticular uptake vary (Fig. 5) – across models, there are substantial, moderate, and low contributions. Soil uptake is low,
except for WRF-Chem Wesely and GEM-MACH Wesely. Lower canopy uptake is low for models that simulate this pathway, like
other forests. Observational constraints on stomatal uptake will help to further narrow model uncertainties as to magnitude and
relative contribution of summertime stomatal uptake.

Interannual variability is strong across months (Fig. 1). A series of papers pointed this out for daytime values and investigated
drivers during summer (Clifton et al., 2017, 2019). Models capture neither the large observed spread across years during summer
nor the ranking of years (Fig. 13). Most models simulate that some of the highest summers observed are low $v_d$ summers.
Previous work points to nonstomatal pathways driving summertime interannual variability (Clifton et al., 2017, 2019), and thus
models may be lacking in their ability to simulate the degree to which nonstomatal uptake varies from year to year, and likely
key process dependencies.
**5.8 Borden Forest**
Borden Forest is a mixed forest in the boreal-temperate transition zone in Canada. The model spread in terms of the model with the
highest annual average $v_d$ divided by the model with the lowest is 2.3 (3.4 during summer and 10 during winter) but based on the
interquartile range is 1.4 (1.8 during summer and 3 during winter). The metrics of model spread are towards the higher end of
other sites, except for winter and the summertime interquartile range when they are the highest.

Observed multiyear monthly mean $v_d$ shows a broad maximum during warm months at Borden Forest (Fig. 1), like Harvard
Forest and Hyytiälä. However, uniquely, observations at Borden Forest show particularly large winter vs. summer differences and steep
changes during spring and fall. Specifically, $v_d$ increases from March to June by 0.5 cm s$^{-1}$. Then, $v_d$ remains high from June to
September (0.6–0.65 cm s$^{-1}$) and declines steeply from September to November. Models simulate higher $v_d$ during warmer vs.
cooler months (Figs. 2, 3), and the observed relative seasonal amplitude lies close to the middle of the central models' spread
(Fig. 9). However, there is a clear discrepancy between models and observations in that models do not capture very high $v_d$
across warm months (Fig. 3). All models except GEM-MACH Wesely have low summertime biases, with a range from -15% to -
74% (Fig. 4). In general, high observed $v_d$ during warm months at Borden Forest needs better understanding, given uncertainty in ozone
flux measurements from the gradient technique (see discussion in Sect. 3.2).

The individual contribution from stomatal uptake is a key driver of $v_d$ seasonality, apart from IFS SUMO Wesely, CMAQ
STAGE, and DO₃SE models (Fig. 6). These four models do, however, show stomatal contributions to seasonality via correlations
with other pathways. Notably, there are more individual nonstomatal contributions to seasonality at Borden Forest than other
forests. There are also a variety of simulated $v_d$ seasonal cycle shapes at Borden Forest, in contrast to Harvard Forest and Ispra.
Some models simulate weak changes from cooler to warm months (DO₃SE models, TEMIR Zhang models, IFS SUMO Wesely,



GEM-MACH Zhang) while others simulate moderate changes (WRF-Chem Wesely, MLC-CHEM, CMAQ STAGE) or strong
changes (GEOS-Chem Wesely, TEMIR Wesely, IFS GEOS-Chem Wesely, GEM-MACH Wesely, CMAQ M3Dry models,
TEMIR Wesely psn). TEMIR psn simulate erratic monthly changes during June to October. Generally, models with the strongest
changes from cooler to warm months simulate that stomatal uptake predominately drives $v_d$ seasonality (Fig. 6). Conversely,
models with weak changes from cooler to warm months indicate that nonstomatal pathways contribute more predominantly.

With respect to the relationship between multiyear monthly mean $v_d$ and $LAI$, observed $v_d$ increases with $LAI$ but the slope varies
(Fig. 7). The observed slope is strongest for 0.5 to 1 $m^2$ $m^{-2}$, and models tend to underestimate this change, but do simulate increases.
Then, the observed slope weakens but remains positive for 1 to 2 $m^2$ $m^{-2}$ – most models suggest decreases instead. Then, the
observed slope weakens even further above 2 $m^2$ $m^{-2}$. Some models capture the slope of $LAI$ increases above 2 $m^2$ $m^{-2}$ but others
exaggerate it (e.g., GEM-MACH Wesely, GEOS-Chem Wesely, TEMIR Wesely, CMAQ M3Dry models). The main issue is that
individual models tend not to capture that there should be relatively high $v_d$ during May and October (Fig. 3). Specifically,
models simulate a later spring onset to higher $v_d$ as well as an earlier fall decline, and thus a shorter season of elevated $v_d$ than
observed. We thus suggest that models are too strongly tied to $LAI$, which strongly increases from May to June and strongly
decreases from September to October (Fig. 10).

Additionally, many models do not capture that multiyear monthly mean $v_d$ is similar during June–September (Fig. 3). Some
models simulate declines from August to September (e.g., CMAQ M3Dry-psn, GEOS-Chem Wesely, TEMIR Wesely, GEM-
MACH Wesely). A weak decline from August to September occurs in the observed multiyear average (the strong decline
happens from September to November); some models capture the August-to-September decline's magnitude while others
exaggerate it. Some models show low values during July (e.g., TEMIR psn), in addition to August-to-September declines.
Observations show low values during July not in multiyear monthly mean seasonal cycles, but during 2012 and perhaps 2008
(Fig. 1). Many models show peak $v_d$ during June. Again, this does not happen in observed multiyear monthly averages, but
occurs in 2010. Thus, models may exaggerate depositional responses (in particular, stomatal) to changes in environmental conditions
(e.g., soil moisture) on a climatological basis but have some skill in certain years.

During summer, the largest model spread across pathways occurs for stomatal uptake, followed by cuticular uptake and then soil
uptake (Fig. 11), similar to Harvard Forest and Ispra. Models show substantial stomatal uptake, apart from two with very low
values (IFS SUMO Wesely and DO₃SE multi). Stomatal fractions range from 20% to 80% across models, but 40% to 62.5%
across central models (Fig. 12). Eight models simulate lower cuticular uptake, while the rest simulate higher cuticular uptake
(Fig. 5). Models with lower canopy uptake show low cuticular uptake, with two exceptions: GEM-MACH Wesely, which has
high cuticular uptake, and MLC-CHEM, which does not archive lower canopy uptake diagnostic but has low cuticular uptake.
Most models simulate low soil uptake, but a few models simulate moderate-to-high soil uptake (GEM-MACH Wesely, GEM-



MACH Zhang, CMAQ STAGE, WRF-Chem Wesely, and MLC-CHEM). Observational constraints on stomatal uptake will help
to further narrow model uncertainties as to the magnitude and relative contribution of stomatal uptake.

During winter, models show a mixture of over- and under-estimates. Models with overestimates are TEMIR Zhang models (+68
to +73%), GEM-MACH Zhang (+124%), WRF-Chem Wesely (+13%), DO$_3$SE multi (+9%) and DO$_3$SE psn (+44%). Otherwise,
underestimates span -20% to -78%. Models with high $v_d$ simulate high cuticular uptake, generally high soil uptake, and in one
case nonnegligible stomatal uptake (DO$_3$SE psn) (Fig. 5). Soil and cuticular uptake show the highest spreads across models, with
soil uptake the highest, similar to Harvard Forest and Ispra (Fig. 11). Central models show very low stomatal fractions, but
outliers span 10% to 30% (Fig. 12). Apart from DOS$_3$E psn, high stomatal fractions are due to high nonstomatal uptake, rather
than high stomatal uptake. Many models largely capture that observations show no $v_d$ change with snow, although some slightly
overestimate the change. Thus, the primary issue with wintertime model biases is likely unrelated to responses to snow, and
rather related to mischaracterized magnitudes of pathways or responses to other environmental conditions.

In terms of summertime $v_d$ interannual variability, some models underestimate the relative spread across years (Fig. 13), but
some only slightly underestimate it (IFS SUMO Wesely, CMAQ STAGE, TEMIR Zhang, MLC-CHEM, DO$_3$SE models) and a
few exaggerate it (TEMIR psn). Models generally struggle to capture the observed relative distribution across summers (i.e., two
high years, two low years, and one middle year). No model captures the year-to-year ranking across summers but many can
simulate that one of the highest years is a high $v_d$ summer and in some cases that one of the lowest years is a low $v_d$ summer.
CMAQ STAGE captures that the other high year is a high year, whereas no other model captures this (or distinguish it from
other years). Figure 1 shows that one year has particularly low $v_d$ during August, and that there is a separation between some
years relative to others during June (three low years vs. two high years). Future work should examine interannual variability in
monthly averages to further establish model skill.
**6 Conclusion**
We introduce AQMEII4 Activity 2 for intercomparison and evaluation of eighteen dry deposition schemes configured as single-
point models at eight sites with ozone flux records, driven by the same set of meteorological and environmental conditions. We
provide our approach's rationale, document the single-point models, and describe the observational datasets used to drive and
evaluate the models. The design of Activity 2 allows us to focus on parametric and process uncertainty. We launch Activity 2
results by analyzing simulated multiyear mean ozone deposition velocities and effective conductances, as well as observed
multiyear mean ozone deposition velocities. Our focus is on monthly and seasonal averages across all hours of the day, apart
from one site for which we examine afternoon averages (Ramat Hanadiv). We evaluate simulated magnitudes and seasonal
cycles (e.g., shape, amplitude) of ozone deposition velocities against observations, and identify how differences and similarities
in relative and absolute contributions of individual deposition pathways and some dependencies on environmental conditions
influence the model spread and comparison with observations.




There are a variety of observed climatological seasonal patterns and magnitudes of ozone deposition velocities across sites. We
emphasize incomplete understanding of observed variations at several sites. Namely, there are unexpectedly high ozone
deposition velocities year-round at Auchencorth Moss, during the cool season at Ispra, and during the warm season at Borden
Forest; models do not capture these high values. Further model evaluation at these sites requires better understanding of the
observations. We emphasize that our measurement testbed is likely insufficient to generalize results to specific LULC types, so
we focus on site-specific results. We also cannot discount the fact that differences in ozone flux methods and instrumentation and
a lack of coordinated processing protocols across data sets limit meaningful synthesis of our results across sites. However, given
that key processes and parameters are strongly tied to LULC type in dry deposition parameterizations, a core question is whether
the magnitude and dependencies of ozone deposition velocities can be described from a LULC-type perspective. To address this
question, future work will need to better understand observed site-to-site differences in ozone deposition velocities, which likely
requires new multiscale ozone flux datasets.

Observed interannual variation in ozone deposition velocities is strong at most sites examined here, demonstrating the
importance of long-term ozone flux records for model evaluation. For example, even if a model captures values for a given year,
the model may not reproduce interannual variability or the multiyear average. Our focus is climatological evaluation, with the
caveat that three sites (Ramat Hanadiv, Auchencorth Moss, and Bugacpuszta) do not have multiple years of data for several
months and two are missing some months of data across all years. Of course, full annual records with several years of data are
required for confident constraints on climatological seasonality. Nonetheless, sites with short-term records have very similar
monthly averages between years when there is good data coverage, with only a few exceptions (October at Auchencorth Moss
and fall at Ispra), implying some utility of these datasets towards our aim.

For sites with more than three summers of data, we identify whether models capture the ranking and spread across summers. We
find that models do not capture observed summertime interannual variability, a finding that agrees with earlier work with one
model at Harvard Forest (Clifton et al., 2017). Our work here shows that the issue is widespread across models and sites.
Specifically, we show poor model skill in simulating the degree of the interannual spread as well as the ranking across years.

Individual model performance strongly varies by season and site. Throughout the manuscript, we examine individual models as
well as model ensembles including the full set of models as well as the interquartile range, which helps us to narrow our focus to
key common uncertainties across models. The interquartile range across simulated averages ranges from a factor of 1.2 to 1.9
annually across sites, and largely reasonably bounds multiyear monthly mean ozone deposition velocities. Exceptions to the
latter are times denoted as particularly uncertain at Auchencorth Moss, Ispra, and Borden Forest, in addition to late summer at
Bugacpuszta and Ramat Hanadiv. The latter finding, together with our finding that many models that include soil moisture
dependencies on stomatal conductance exaggerate late-summer decreases in ozone deposition velocities at forests, suggests a
need to focus on refining soil moisture dependencies. Such work should probe interannual variability and seasonality with



additional observational constraints on stomatal uptake in the context of uncertainty in soil moisture input data. In general, in
some cases, gaps in site-specific measurement data (e.g., soil moisture and characteristics) forced us to make assumptions or
derive estimates for key model variables and parameters. This may influence model performance, and points to a need for a
standard minimum set of observations at future field studies.
Even beyond differing effects of soil moisture across the ensemble of models, there are differences in simulated seasonal cycle
shapes of ozone deposition velocities. Models that rely strongly on seasonally dependent parameters are often identified as
outliers, so we recommend that related canopy resistance equations should be tied to variables like leaf area index instead of only
seasonally varying parameters. In principle, seasonally varying parameters are not problematic, but a challenge seems to be
indicating site-specific phenology accurately. At half the sites, the model spread is highest during cooler months, implying a
need to better understanding of wintertime deposition processes. Strong wintertime sensitivities of tropospheric ozone
abundances in regional-to-global chemical transport models (Helmig et al., 2007; Matichuk et al., 2017; Clifton et al., 2020b)
also point to this need. By compositing observed and simulated ozone deposition velocities for all vs. snowy conditions during
cool months at sites with snow depth observations, we show that models' inability to capture the magnitude of wintertime values
generally is a larger issue than models' inability to capturing responses to snow. While our analysis suggests that snow-induced
changes are not the main driver of observed seasonality in ozone deposition velocities, we also find models may too strongly rely
on leaf area index to determine seasonality.
Several papers illustrate challenges in determining which ozone dry deposition parameterization is best given observations
compiled from the literature (Wong et al., 2019; Cao et al., 2022; Sun et al., 2022) or comparing seasonal differences for ozone
and sulfur dioxide deposition velocities at Borden Forest (Wu et al., 2018). While we agree with these earlier findings with our
completer and more diverse testbed, we take the evaluation a step further by pinpointing how different pathways contribute to the
spread. In general, both stomatal and nonstomatal pathways are key drivers of variability in ozone deposition velocities across
models. Additionally, in some cases, ozone deposition velocities are similar across models when the partitioning among
deposition pathways is very different (i.e., similar results for different reasons).
For the most part, models simulate that stomatal uptake predominately drives seasonality in ozone deposition velocities. Like
large model differences in seasonality of ozone deposition velocities, there are large model differences in seasonality of stomatal
uptake. A few models show that seasonality in nonstomatal uptake terms is also important for seasonality in ozone deposition
velocities. Across sites, both stomatal and nonstomatal pathways are important contributors to ozone deposition velocities during
the growing season. For example, during summer, the median of the stomatal fraction of the ozone deposition velocity across
models ranges from 30% to 55% across most sites. Thus, like observationally based estimates of stomatal fraction over
physiologically active vegetation compiled by a recent review (Clifton et al., 2020a), models clearly indicate a codominant role
for dry deposition through nonstomatal pathways. Nonetheless, as stated in the previous paragraph, we emphasize large
differences in simulated nonstomatal uptake, in addition to stomatal uptake, across models.




In general, we confirm here with our unprecedented full documentation of eighteen dry deposition schemes that dry deposition
schemes, especially nonstomatal deposition pathways, are highly empirical. While some schemes can capture some of the salient
features of observations and schemes could be adjusted to better capture the magnitude of observed ozone deposition velocities
at the sites examined here, better mechanistic understanding of observed variability, and a firm grasp on how different deposition
pathways change in time and space on different scales, are needed to improve predictive ability of ozone dry deposition. We will
continue to chip away at this problem; next for Activity 2 will be to leverage observation-based constraints on stomatal
conductance, together with inferred stomatal fractions of ozone deposition velocities, and examine diel, seasonal, and interannual
variations to further evaluate single-point models.
**Data Availability**
The hourly or half hourly observed ozone flux and forcing datasets are available to individuals wishing to participate in this
effort on a password-protected site managed by the U.S. EPA, subject to the individual's agreement that the people who created
and maintained the observation datasets are included in publications as the people see fit. Some datasets are already available
publicly, and in these cases, we have included the references to the datasets in the text.
**Author Contributions**
O. E. C. lead the manuscript's direction and writing, data processing and analysis, and coordination among authors. D. S. and C.
H. contributed to the manuscript's direction, data processing, and coordination among authors. J. O. B. contributed CMAQ
STAGE results and documentation. S. B. contributed DO$_3$SE results and documentation. P. C. contributed GEM-MACH results
and documentation. M. C. contributed data from Easter Bush and Auchencorth Moss. L. E. contributed DO$_3$SE results and
documentation and assisted with direction. J. F. contributed IFS results and documentation and assisted with direction. E. F.
contributed data from Ramat Hanadiv. S. G. assisted with direction. L. G. contributed MLC-CHEM results and documentation.
O. G. contributed data from Ispra. C. D. H. assisted with direction and contributed GEOS-Chem results and documentation. I. G.
contributed data from Ispra. L. H. contributed data from Bugacpuszta. V. H. contributed model results and documentation from
IFS. Q. L. contributed data from Ramat Hanadiv. P. A. M. contributed model results and documentation from GEM-MACH and
assisted with direction. I. M. contributed data from Hyytiälä. G. M. contributed data from Ispra. J. W. M. contributed data from
Harvard Forest. J. L. P. C. contributed WRF-Chem results and documentation. J. P. contributed M3Dry results and
documentation. L. R. contributed M3Dry results and documentation. R. S. J. contributed WRF-Chem results and documentation.
R. S. contributed data from Borden Forest. S. J. S. assisted with data processing and assisted with direction. S. S. and A. P. K. T
contributed TEMIR results and documentation. E. T. contributed data from Ramat Hanadiv. T. V. contributed data from
Hyytiälä. T. W. contributed data from Bugacpuszta. Z. W. and L. Z. contributed data from Borden Forest. All authors
contributed to manuscript writing and useful discussions on data analysis and processing and results.



**Acknowledgements**
The views expressed in this article are those of the author(s) and do not necessarily represent the views or policies of the U.S.
Environmental Protection Agency. Borden Forest Research Station is funded and operated by Environment and Climate Change
Canada. Easter Bush measurements were funded by European Union projects GREENGRASS (EC EVK2-CT2001-00105),
NitroEurope Integrated Project (contract no. 017841) and CarboEurope (contract no. GOCE-CT-2003-505572), and by the UK
DEFRA 1/3/201 Effects of Ground Level Ozone on Vegetation in the UK and the UK NERC Core national capability. We thank
the field teams at Easter Bush and Auchencorth Moss and other UK CEH staff, as well as Ivan Simmons and Carole Helfter. For
Hyytiälä, we acknowledge Petri Keronen, Pasi Kolari, and Üllar Rannik. For Ispra, we acknowledge technical assistance from
Carsten Gruening and Olga Pokorska. For Ramat Hanadiv, E. T. and E. F. acknowledge the Israel Science Foundation, Grant No.
1787/15, the Joseph H. and Belle R. Braun Senior Lectureship in Agriculture to E. T., and the crew at Ramat Hanadiv. Harvard
Forest observations were supported in part by the U.S. Department of Energy, Office of Science (BER), and National Science
Foundation Long-Term Ecological Research. O. E. C. acknowledges support from an appointment to the NASA Postdoctoral
Program at the NASA Goddard Institute for Space Studies, administered by Oak Ridge Associated Universities under contract
with NASA. C. D. H. was supported by the National Science Foundation (grant no. 1848372). I. M. and T. V. thank the
Academy of Finland Flagship funding (grant no. 337549) and ICOS-Finland by University of Helsinki funding. L. H. and T. W.
was partly supported by the National Research, Development and Innovation Office Grant K138176. DO$_3$SE runs performed by
L. E. and S. B. were in part supported by a project grant (NE/V02020X/1) of the Future of UK Treescapes research program
funded by the UKRI.

**Competing Interests**
None



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
