# Peer review of "A single-point modeling approach for the intercomparison and evaluation of ozone dry deposition across chemical transport models (Activity 2 of AQMEII4)"

_EGUsphere, 2023_

## Author Comment (AC1)

We are grateful to the two reviewers for their careful reads of our manuscript and constructive comments. Our response to review is written in **bold blue type**. Text that appears in our manuscript is in double quotation marks, and underlines within quoted text emphasize changes to old text addressing the reviewer's specific comment. Line numbers refer to line numbers in the new version of our manuscript.

Anonymous Referee #1

This manuscript provides a comprehensive intercomparison between established ozone dry deposition parameterizations. The result is somewhat expected but still highly relevant and valuable for the community, established with robust result.

I have only a few minor questions and suggestions:

• The model description is nice and thorough. However, putting them in the main text obstruct the flow of the manuscript. I suggest the authors move the detailed model description to appendix/supplemental material. The authors could also consider using tables to make the model description more organized and readable.

We thank the reviewer for the suggestion to reconsider the organization of this information to improve readability. As a result of this reconsideration, we now include subsections for the different resistances to better organize the descriptions for each model. As recognized by the reviewer, the model descriptions are fundamental to our effort, especially given the lack of documentation of dry deposition schemes (and particularly their implementation in chemical transport models) in the peer reviewed literature, and the need to rely on these descriptions for the interpretation of model differences. We would therefore like to make it clear to the reader that this documentation exists, and thus think that the documentation – but reorganized to improve readability – should remain in the main manuscript file.

• L 79 – 80: I suggest "land carbon sink" instead of "carbon storage, and more example/elaboration about how ozone affect ecosystem service

We now say: L75-76: "alters terrestrial carbon and water cycles" instead of "carbon storage."

We removed the 'ecosystem services' term, especially considering recent developments in the US to standardize the meaning of this term.

• Table 1: What precisely is *B*? "Parameter related to soil moisture" sounds very vague.

We now say: L958-962: "A set of soil hydraulic properties (Table S20) are estimated for each site from soil texture and used across models employing these parameters. For example, the variable *B* is an empirical parameter, which is calculated as the slope of the water retention curve in log space (Cosby et al. 1984), that relates volumetric soil water content to soil matric potential and can be referred to as a bulk hydraulic property of the soil (Clapp and Hornberger, 1978; Letts et al., 2000)."

• L 228 – 230: More discussion about how the uncertainty in  $r_a$  (choice of MOST universal function, h and  $z_a$ ) may (or may not) affect the study can be helpful.

We now say in the Conclusion: L1754-1756: "We encourage future work to examine the roles of parameters, sensitivities, and transport related processes. For example, previous work shows that differences in deposition velocities among air quality models under stable conditions may at least in part be due to different empirical formulations of Monin-Obukhov Similarity Theory (Toyota et al., 2016)."

• L 284 – 285: Clarify what is "effective LAI". What is its physical/biological meaning? How is it calculated?

This is a quantity produced from an empirical function in GEOS Chem (we say: "The variable  $LAI_{eff}$  is calculated using function of LAI, solar zenith angle ( $\theta$ ) [ $^{o}$ ], and cloud fraction."). We rephrased to make the physical/biological meaning clearer: L296-7: " $LAI_{eff}$  [m2 m-2] is effective LAI, which is the surface area of actively transpiring leaves per ground surface area".

• L 841: What "other compounds" and why are they "challenging" to be measured in high frequency? Some examples, discussions and citations would be helpful.

We now say: L886-894: "A key reason is that obtaining high-frequency concentration measurements of some compounds (e.g., NO2, SO2, HNO3, H2O2) can be challenging due to the detection limits of fast response sensors, the demands of running research grade instruments in an eddy covariance configuration (e.g., consumables, dedicated staff, data storage), and potential flux divergences due to atmospheric chemical consumption or production on the same time scale as deposition processes (Ferrara et al., 2021; Fischer et al., 2021). Nonetheless, recent work further developing or creating new instruments for eddy covariance fluxes of black carbon, ozone, NO2, ammonia, and a large suite of organic gases (Philips et al., 2013; Nguyen et al., 2015; Emerson et al., 2018; Fulgham et al., 2019; Novak et al., 2020; Hannun et al., 2020; Ramsay et al., 2018; Schobesberger et al., 2023; Vermeuel et al., 2023) demonstrates the potential for more widespread measurements that would assist in assessing the accuracy of dry deposition schemes more broadly."

• L 1728: Could the authors provide how may we address the over-reliance on LAI to determine seasonality? E.g. Would other ecophysiological parameters (e.g. seasonally-varying leaf nitrogen content/leaf-level photosynthetic capacity) help? What factors other than phenology might contribute to the seasonality of *v*d, but

not yet considered in the parameterizations? Is the seasonality of non-stomatal ozone uptake under-represented?

This is a good question, and one that we are actively trying to address. Our next paper uses observation-based estimates of stomatal conductance (from water and CO2 fluxes) to better constrain the single-point models, especially to establish how we can better simulate seasonality in stomatal uptake, and thus nonstomatal uptake. The last line of the paper says: "We will continue to chip away at this problem; next for Activity 2 will be to leverage observation-based constraints on stomatal conductance, together with inferred stomatal fractions of ozone deposition velocities, and examine diel, seasonal, and interannual variations to further evaluate single-point models."

Other AQMEII Activity 2 work is investigating leaf level nitrogen content as a more mechanistic variable to explain differences among sites and the application of this variable to plant functional types in global and regional parameterizations.

Citation: https://doi.org/10.5194/egusphere-2023-465-RC1

**Anonymous Referee #2 Reviewer summary**

This work contributes to Activity 2 of the AQMEII4 framework. This manuscript uses 18 models and model variants to examine how differences in their parameterization of ozone dry deposition drives model bias with respect to observations (of ozone dry deposition). To isolate inter-model variability due to differences in their ozone dry deposition parameterization, the models are used in a single point configuration and driven by observed meteorology and environmental conditions at each of eight sites. The models' ability to capture seasonal patterns and interannual variability (where possible) in ozone dry deposition is evaluated at the eight sites. The contribution of variability in the simulated deposition pathways (stomatal, soil etc.) to the models' bias against observations is also evaluated. Overall, this study finds that, broadly, models' ozone dry deposition pathways are required and more detailed observations are required to unpick the drivers of model biases in ozone dry deposition. I have some general and technical comments (see below) that should be addressed prior to publication.

**General comments:**

The authors' efforts to provide comprehensive descriptions of the ozone dry deposition schemes used in the 18 models and model variants is very commendable. Similarly, their efforts to compile comprehensive multi-annual observational datasets. Both are valuable resources for the community. The manuscript is also clearly laid out and well written.

Given the multitude of sites, models and dry deposition pathways it is difficult to draw overarching conclusions on the drivers of model bias in ozone dry deposition here. While I agree with the authors recommendations for more detailed observational data, which may help with identifying the biases deposition, the time scales required to generate the type of long-term data sets are obviously quite long. I would therefore be keen to hear more about the author's plans or ideas to identify drivers of model biases using the data sets developed for this manuscript. For example, would more detailed statistical analysis or model sensitivity studies be useful? Or is it that real-world heterogeneity at the site level prevents over-arching conclusions on sources of model bias for ozone dry deposition?

We think that more detailed statistical analysis and model sensitivity studies will be helpful, and we have ongoing and planned studies doing just this. However, we also think that there is only so far that we can go, given real-world heterogeneity at the site level, in making general, overarching conclusions.

We discuss the real-world heterogeneity in the conclusion saying, "We emphasize that our measurement testbed is likely insufficient to generalize results to specific LULC types... We also cannot discount the fact that differences in ozone flux methods and instrumentation and a lack of coordinated processing protocols across data sets limit meaningful synthesis of our results across sites." Originally, we included another point of discussion (on how some observed features are uncertain and the models don't capture them) here, but we moved this discussion elsewhere to clarify our point here. We now also repeat this at the beginning of 4.2.

We reference some of our plans for future analysis in the last line of the paper. However, we prefer not to elaborate because plans are subject to change. We say: "We will continue to chip away at this problem; next for Activity 2 will be to leverage observation-based constraints on stomatal conductance, together with inferred stomatal fractions of ozone deposition velocities, and examine diel, seasonal, and interannual variations to further evaluate single-point models."

1. Section 2.1

Would it be possible to highlight the dry deposition scheme components by group? For example, bold or underlined headings for e.g. 'Stomatal resistances', 'Non-stomatal resistances', 'Environmental dependencies' for each of the models could help readers navigate the schemes for their future reference.

Done. We thank the reviewer for this suggestion to improve the readability of Section 2.1 and now do this for each model subsection.

1. Figure captions after 'Figure 3' need relabelling.

Done. We thank the reviewer for pointing this out.

**Technical comments:**

1. Introduction, L140: "...global chemical transport models and used always as standalone models..."

=> Perhaps remove 'and' in the above sentence.

We thank the reviewer for pointing out to us that this was confusing. We now say: L136-7: "we also include schemes from global chemical transport models and schemes that are used always as standalone models".

1. Section 2.1.2, L279: "...then the parameter's value in Table S6."

=> "...then the parameter's value **is** in Table S6."

**Done; we thank the review for finding this!**

1. Section 2.1.2, L287: "...so that nighttime rst values on the single point model more similar to GEOS-Chem."

=> "...so that nighttime rst values on the single point model **are** more similar to GEOS-Chem."

**Done; we thank the review for finding this!**

1. Section 2.1.7, L553: "van der Walls"

=> van der Waals

**Done; we thank the review for finding this!**

**5.Figures 2 (or 3):**

Would it be possible to indicate the inter-annual variability in the observations here (Fig. 2 might be better), possibly as vertical bars/whiskers? I'm aware that these plots are illustrating inter-model variability, rather than inter-annual model variability – although the latter looks to be encapsulated by the former in Fig. 2. However, in the context of the site specific discussions, I think it would be useful to illustrate the inter-annual variability in the observations.

Given that the model years and observations years are the same, and site-specific forcing is used to run the models, the climatological observed and simulated averages are comparable. We think that describing interannual variability in the observations and inter-model variability on a single figure would be misleading. We note that we do compare the range and ranking of summertime interannual variability between the observations and models in Figure 13, in addition to Figure 1 that illustrates the interannual variability in the observations for all months.

**Citation**: https://doi.org/10.5194/egusphere-2023-465-RC2